# AEGIS: Adversarial Target-Guided Retention-Data-Free Robust Concept Erasure from Diffusion Models

**Fengpeng Li**[1,4]    **Kemou Li**[1]    **Qizhou Wang**[2,3]    **Bo Han**[2]    **Jiantao Zhou**[1*]

[1]State Key Laboratory of Internet of Things for Smart City, University of Macau
[2]TMLR Group, Department of Computer Science, Hong Kong Baptist University
[3]Imperfect Information Learning Team, RIKEN Center for Advanced Intelligence Project
[4]PRADA Lab, King Abdullah University of Science and Technology

## Abstract

Concept erasure helps stop diffusion models (DMs) from generating harmful content; but current methods face robustness–retention trade-off. **Robustness** means the model fine-tuned by concept erasure methods resists reactivation of erased concepts, even under semantically related prompts. **Retention** means unrelated concepts are preserved so the model's overall utility stays intact. Both are critical for concept erasure in practice, yet addressing them simultaneously is challenging, as existing works typically improve one factor while sacrificing the other. Prior work typically strengthens one while degrading the other—e.g., mapping a single erased prompt to a fixed safe target leaves class-level remnants exploitable by prompt attacks, whereas retention-oriented schemes underperform against adaptive adversaries. This paper introduces _Adversarial Erasure with Gradient-Informed Synergy_ (AEGIS), a retention-data-free framework that advances both robustness and retention. First, AEGIS replaces handpicked targets with an Adversarial Erasure Target (AET) optimized to approximate the semantic center of the erased concept class. By aligning the model's prediction on the erased prompt to an AET-derived target in the shared text–image space, AEGIS increases predicted-noise distances not just for the instance but for semantically related variants, substantially hardening the DMs against state-of-the-art adversarial prompt attacks. Second, AEGIS preserves utility without auxiliary data via Gradient Regularization Projection (GRP), a conflict-aware gradient rectification that selectively projects away the destructive component of the retention update only when it opposes the erasure direction. This directional, data-free projection mitigates interference between erasure and retention, avoiding dataset bias and accidental relearning. Extensive experiments show that AEGIS markedly reduces attack success rates across various concepts while maintaining or improving FID/CLIP versus advanced baselines, effectively pushing beyond the prevailing robustness–retention trade-off. The source code is in `https://github.com/Feng-peng-Li/AEGIS`.

## 1 Introduction

While DMs (Sohl-Dickstein et al., 2015; Ho et al., 2020) excel at text-to-image (T2I) generation, biased training data poses safety and reliability risks (Li et al., 2024a; 2025a), including harmful outputs (Li et al., 2026). Concept erasure (Gandikota et al., 2023), modifying the denoising UNet of DMs to remove undesirable concepts, has become a standard measure to obtain a reliable DMs. These methods are typically categorized as output-based, realigning concept representations, or attention-based, aiming at manipulating cross-attention scores (Bui et al., 2025). Despite progress, these techniques face two critical challenges. First, as shown in Fig. 1, erased concepts can be reactivated by adversarial or semantically related prompts not seen during training (Chin et al., 2024). Second, improving robustness against adversarial prompt attacks (APAs) comes at the cost of degrading the

---

*Corresponding author: Jiantao Zhou (jtzhou@um.edu.mo)

generation quality for unrelated concepts, creating a harsh robustness-retention trade-off (Zhang et al., 2024b;c; Gong et al., 2024; Meng et al., 2024b). Both goals are critical for concept erasure, while seldom works can handle them simultaneously. Consequently, achieving robust erasure without compromising retention remains a central, unsolved problem.

Prevailing studies (Bui et al., 2025) largely focus on the impact of mapping target on the trade-off between concept erasure and the retention. A critical yet underexplored dimension, however, is the influence of the erasure target on the model's robustness against APAs, mitigating the trade-off seems better. In most methods, only an instance for the concept, e.g., *nudity*, is adopted to

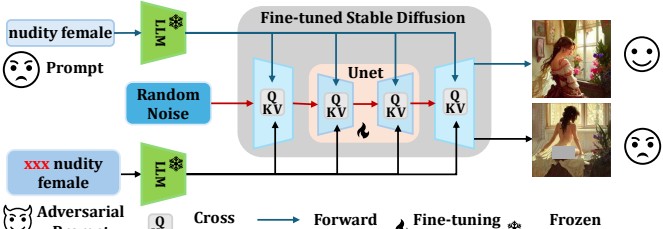

Figure 1: Adversarial prompts can still result in DMs fine-tuned by concept erasure methods generating images with harmful information.

perform concept erasure by maximizing the distance of noises induced by the instance between pre- and post-fine-tuning DMs. However, practical users input often involve semantically equivalent or metaphorical expressions, such as *naked* or *sexual*. Since DM can be interpreted as a classifier (Chen et al., 2024), these synonymous terms are likely to be clustered within the same semantic class in the latent space. If the chosen instance lies too far to the semantic center of the *nudity* class, the concept information may not be fully eliminated. Under APAs like Zhang et al. (2024c) close to the concept semantic center, the incompleteness of *nudity* class erasure becomes evident.

Building upon these insights, we introduce AEGIS (*Adversarial Erasure with Gradient-Informed Synergy*), a novel framework that significantly enhances the robustness of concept erasure against APAs while preserving the retention for unrelated concepts. AEGIS achieves this through two key components: *Adversarial Erasure Target* (AET) and *Gradient Regularization Projection* (GRP). AET with an optimizable embedding is iteratively updated to create a potent adversarial target. The model is then fine-tuned to minimize the difference between the noise predicted for AET and the erased concept, effectively strengthening the erasure process. GRP is applied to maintain the model's performance on unrelated concepts without requiring additional retention data, thus preventing negative impacts from the erasure process. Experiments across various concepts against a range of APAs confirm the superior effectiveness of AEGIS, which surpasses state-of-the-art methods in erasure robustness by large margins while preserving higher retention performance.

**Contribution Summary.** The contributions of this work can be summarized as follows:

- We demonstrate that the vulnerability of concept erasure stems from an inappropriately chosen learning target. In particular, if the target lies too close to the semantic center—formed by words semantically related to the erased concept—the concept information cannot be fully removed.
- We propose a novel *retention-data-free* framework, AEGIS, which synthesizes an AET to guide the removal of class-level information associated with erased concepts. Additionally, we introduce a gradient projection strategy to better balance erasure robustness and retention performance.
- Extensive experiments on multiple datasets validate that AEGIS markedly enhances erasure robustness against a variety of APAs. Notably, AEGIS decreases the attack success rate by ∼5.31% on the *nudity* concept and ∼24% on the *Van Gogh* style concept under P4D (Chin et al., 2024) and UnlearnDiffAtk (Zhang et al., 2024c), while incurring little impact on retention performance.

## 2 PRELIMINARIES

**Latent Diffusion Models.** This work focuses on latent diffusion models (LDMs) (Rombach et al., 2022). Given a training dataset $\mathcal{D} = \{(\mathbf{x}_i, \mathbf{c}_i)\}_{i=1}^N$, each image $\mathbf{x}_i$ is paired with a prompt $\mathbf{c}_i$. A noise predictor $\epsilon_{\boldsymbol{\theta}}(\cdot|\mathbf{c})$, parameterized by $\boldsymbol{\theta}$, is learned to estimate the Gaussian noise $\epsilon$ added to the latent representation $\mathbf{z}_t$ at each diffusion step $t \in \{1, \dots, T\}$, conditioned on $\mathbf{c}$, optimized as:

$$\min_{\boldsymbol{\theta}} \mathbb{E}_{(\mathbf{x},\mathbf{c}) \sim \mathcal{D}, t, \epsilon \sim \mathcal{N}(\mathbf{0}, \mathbf{I})} \left[ \|\epsilon - \epsilon_{\boldsymbol{\theta}}(\mathbf{z}_t|\mathbf{c})\|_2^2 \right]. \tag{1}$$

**Concept Erasure in DMs.** To remove specific undesirable concepts from DMs, $\epsilon_{\boldsymbol{\theta}}$ is fine-tuned. Let $\mathbf{c}_e$ denote the prompt about the *(to-be-)erased concept*, e.g., "nudity". Existing methods (Rombach et al., 2022; Bui et al., 2024; 2025) map $\mathbf{c}_e$ to an *erasure target concept* $\tilde{\mathbf{c}}$—a generic concept (e.g.,

"a photo") or simply an empty prompt. In parallel, a retention dataset, $\mathcal{D}_r = \{(\mathbf{x}_{r,i}, \mathbf{c}_{r,i})\}_{i=1}^{N_r}$ where $\mathbf{c}_{r,i} \in \mathcal{C}_{retain}$ is unrelated to $\mathbf{c}_e$, is employed to retain the model utility. Let $\boldsymbol{\theta}_0$ denote the original parameters, the erasing objective is defined as:

$$\min_{\boldsymbol{\theta}} \Big[ \underbrace{\mathbb{E}_t\big[\|\boldsymbol{\epsilon}_{\boldsymbol{\theta}}(\mathbf{z}_t|\mathbf{c}_e) - \boldsymbol{\epsilon}_{\boldsymbol{\theta}_0}(\mathbf{z}_t|\tilde{\mathbf{c}})\|_2^2\big]}_{\text{Erasing Loss}} + \lambda \cdot \underbrace{\mathbb{E}_{\mathcal{D}_r,t}\big[\|\boldsymbol{\epsilon}_{\boldsymbol{\theta}}(\mathbf{z}_t|\mathbf{c}_r) - \boldsymbol{\epsilon}_{\boldsymbol{\theta}_0}(\mathbf{z}_t|\mathbf{c}_r)\|_2^2\big]}_{\text{Retention Loss}} \Big], \qquad (2)$$

where $\lambda$ balances erasure and retention. The erasing loss encourages the DM to respond to $\mathbf{c}_e$ as if it were $\tilde{\mathbf{c}}$, thereby reducing its ability to reconstruct $\mathbf{c}$. In general, $\boldsymbol{\epsilon}_{\boldsymbol{\theta}_0}(\mathbf{z}_t|\tilde{\mathbf{c}})$ is set as $\boldsymbol{\epsilon}_{\boldsymbol{\theta}_0}(\mathbf{z}_t) - \eta \cdot (\boldsymbol{\epsilon}_{\boldsymbol{\theta}_0}(\mathbf{z}_t|\mathbf{c}_e) - \boldsymbol{\epsilon}_{\boldsymbol{\theta}_0}(\mathbf{z}_t))$, where $\boldsymbol{\epsilon}_{\boldsymbol{\theta}_0}(\mathbf{z}_t)$ is unconditional generation and $\eta$ controls the erasing strength.

**Adversarial Prompt Attacks and Robust Concept Erasure.** Prior studies (Zhang et al., 2024c; Chin et al., 2024) reveal that, even after applying the erasure objective in Eq. (2), the erased concept may still resurface under adversarial prompt attacks. Given a $\mathbf{c}_e$ that has been erased, one can craft an adversarial prompt $\mathbf{c}^*$ to provoke the concept-erased model into producing a response akin to that of the original model when queried with $\mathbf{c}_e$. Specifically, $\mathbf{c}^*$ can be obtained by solving

$$\mathbf{c}^* = \underset{\|\mathbf{c}-\mathbf{c}_e\|_0 \leq m'}{\arg\min} \mathbb{E}_t\big[\|\boldsymbol{\epsilon}_{\boldsymbol{\theta}}(\mathbf{z}_t|\mathbf{c}) - \boldsymbol{\epsilon}_{\boldsymbol{\theta}_0}(\mathbf{z}_t|\mathbf{c}_e)\|_2^2\big], \qquad (3)$$

subject to the constraint that $m' \leq m$ dimensions of $\mathbf{c} \in \mathbb{R}^m$ are learnable during optimization. To strengthen erasure robustness against such attacks, inspired by adversarial training (Madry et al., 2018; Li et al., 2024b; 2025b), AdvUnlearn (Zhang et al., 2024b) substitutes the original erased concept $\mathbf{c}_e$ in Eq. (2) with an adversarial prompt $\mathbf{c}^*$ directly within the objective. This compels the model to suppress not only prompts explicitly tied to $\mathbf{c}_e$ but also nearby adversarial variants, thereby improving robustness against prompt-space leakage.

With the preliminaries of concept erasure established, we now examine its inherent vulnerabilities.

## 3 MOTIVATION: UNVEILING THE VULNERABILITY IN CONCEPT ERASURE

This section explores a fundamental vulnerability in current approaches of concept erasure in DMs, motivating the design of our `AEGIS`. Specifically, we investigate the following central question:

*Can a single prompt entity suffice to effectively remove all information of a concept from DMs?*

To explore this, we conduct an initial study on erasing the concept *nudity* ($\mathbf{c}_e^0$) and assess its influence on semantically related concepts—*naked, sexual, erotic, and impure* ($\mathbf{c}_e^i, i = 1, 2, 3, 4$). We define three concept groups to facilitate our analysis: $\mathcal{C}_0 = \{\mathbf{c}_e^i\}_{i=0}^4, \mathcal{C}_1 = \{\mathbf{c}_e^0, \mathbf{c}_e^1\}$, and $\mathcal{C}_2 = \{\mathbf{c}_e^2, \mathbf{c}_e^3, \mathbf{c}_e^4\}$.

To characterize how concept erasure impacts related information, inspired by Chen et al. (2024) interpreting DMs as generative classifiers, we view the predicted noise $\boldsymbol{\epsilon}_{\boldsymbol{\theta}}(\mathbf{z}_t|\mathbf{c}_e)$ at timestep $t$ as the model's internal representation of concept $\mathbf{c}_e$. At the final step $T$, this noise becomes deterministic that reflects the location of $\mathbf{c}_e$ in the latent space, and can be interpreted as a class prototype. Intuitively, similar concept instances induce similar noises, thus forming clusters in the latent space.

This perspective enables two types of measurement: *1) Intra-model distances,* reflecting semantic separation between concept instances in the same DM; *2) Cross-model distances,* quantifying erasure strength by comparing predictions from pre- and post-erasure models. The former is exactly what we discussed above. To theoretically justify the latter, we establish Prop. 3.1, suggesting that successful erasure results in a significant shift in the predicted noise embedding of the erased concept.

**Proposition 3.1** (Deviation Lower Bound). *Let* $\mathcal{L}_{erase}(\boldsymbol{\theta}) = \mathbb{E}_t\big[\|\boldsymbol{\epsilon}_{\boldsymbol{\theta}}(\mathbf{z}_t|\mathbf{c}_e^0) - \boldsymbol{\epsilon}_{\boldsymbol{\theta}}(\mathbf{z}_t|\tilde{\mathbf{c}})\|_2^2\big]$, *and* $\Delta_T = \|\boldsymbol{\epsilon}_{\boldsymbol{\theta}_0}(\mathbf{z}_T|\mathbf{c}_e^0) - \boldsymbol{\epsilon}_{\boldsymbol{\theta}_0}(\mathbf{z}_T|\tilde{\mathbf{c}})\|_2^2$. *Assume optimizing Eq. (2) yields* $\mathcal{L}_{erase}(\boldsymbol{\theta}) \leq \delta < \Delta_T$. *Then,*

$$\big\|\boldsymbol{\epsilon}_{\boldsymbol{\theta}}(\mathbf{z}_T|\mathbf{c}_e^0) - \boldsymbol{\epsilon}_{\boldsymbol{\theta}_0}(\mathbf{z}_T|\mathbf{c}_e^0)\big\|_2^2 \geq \big(\sqrt{\Delta_T} - \sqrt{\delta}\big)^2. \qquad (4)$$

The proof is deferred to Appx. E.1. Prop. 3.1 indicates that successful concept erasure leads to a significant deviation in predicted noises before and after fine-tuning the DM. Based on this theoretical insight, Def. 1 formally defines predicted noise distances to systematically evaluate erasure quality and robustness across different concepts and methods: $d_0$ quantifies semantic similarity within the base model, $d_1$ captures erasure effectiveness across pre- and post-erasure models, and $d_2$ evaluates the distance between original and adversarial concept prompts within the fine-tuned DM, reflecting

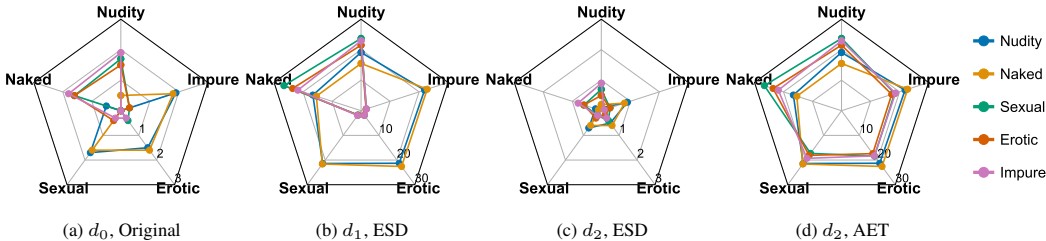

Figure 2: Predicted noise distances ($10^{-4}$) between *nudity* and its synonyms, measured using the original DM $\boldsymbol{\theta}_0$ and concept-erased DMs $\boldsymbol{\theta}_{\text{ESD}}$ and $\boldsymbol{\theta}_{\text{AET}}$.

adversarial vulnerability. With these metrics, we in Fig. 2 empirically analyze the impact of concept erasure on *nudity* and its related concepts using ESD (Rombach et al., 2022) and AdvUnlearn.

**Definition 1** (Predicted Noise Distance). *Define the predicted noise distances $d_0, d_1, d_2$ between any two concept instances $\mathbf{c}_e^i, \mathbf{c}_e^j \in \mathcal{C}_0$ as:*

- $d_0(\mathbf{c}_e^i, \mathbf{c}_e^j) \coloneqq \|\boldsymbol{\epsilon}_{\boldsymbol{\theta}_0}(\mathbf{z}_T|\mathbf{c}_e^i) - \boldsymbol{\epsilon}_{\boldsymbol{\theta}_0}(\mathbf{z}_T|\mathbf{c}_e^j)\|_2^2$

- $d_1(\mathbf{c}_e^i, \mathbf{c}_e^j) \coloneqq \|\boldsymbol{\epsilon}_{\boldsymbol{\theta}}(\mathbf{z}_T|\mathbf{c}_e^i) - \boldsymbol{\epsilon}_{\boldsymbol{\theta}_0}(\mathbf{z}_T|\mathbf{c}_e^j)\|_2^2$

- $d_2(\mathbf{c}_e^i, \mathbf{c}_e^j) \coloneqq \|\boldsymbol{\epsilon}_{\boldsymbol{\theta}}(\mathbf{z}_T|\mathbf{c}_e^i) - \boldsymbol{\epsilon}_{\boldsymbol{\theta}}(\mathbf{z}_T|\mathbf{c}_e^{j,*})\|_2^2$

Table 1: Concept erasure robustness by UnlearnDiffAtk (Zhang et al., 2024c).

| METHOD | Pre-ASR ($\downarrow$) | ASR ($\downarrow$) |
|---|---|---|
| SD v1.4 | 100% | 100% |
| ESD | 20.42% | 76.05% |
| AdvUnlearn | 15.45% | 64.79% |
| ESD w/ AET | 6.38% | 26.06% |
| AdvUnlearn w/ AET | 1.41% | 5.63% |

**Impact of Concept Erasure.** Fig. 2a reveals that noise for *nudity* and *naked* ($\mathcal{C}_1$) exhibit proximity, while *sexual*, *erotic*, and *impure* ($\mathcal{C}_2$) show significant divergence. After erasing *nudity* via ESD, Fig. 2b demonstrates enlarged $d_1$ for $\mathcal{C}_1$ but limited changes for $\mathcal{C}_2$, indicating ineffective removal of $\mathcal{C}_2$. This observation aligns with pre-attack success rate (Pre-ASR) results in Tab. 1: ESD-fine-tuned DMs still generate nudity images in response to benign prompts from Zhang et al. (2024c), indicating that using an instance from $\mathcal{C}_1$ or $\mathcal{C}_2$ as $\mathbf{c}_e$ cannot remove all information of $\mathcal{C}_0$ from the DM.

**Impact of APAs.** Despite ESD fine-tuning, Fig. 2c reveals consistently small $d_2$ for $\mathcal{C}_0$. This low $d_2$ enables adversarial prompts to achieve high ASR, as reported in Tab. 1. Even AdvUnlearn, which integrates adversarial prompts into Eq. (2), fails to defend against UnlearnDiffAtk. From the phenomenon, *we guess this vulnerability stems from inadequate mapping targets $\tilde{\mathbf{c}}$.* As shown in Fig. 9a of Appx. D.1, if $\mathbf{c}_e$ (e.g., member of $\mathcal{C}_1$) is far from the concept center of $\mathcal{C}_0$, maximizing the distance between $\boldsymbol{\epsilon}_{\boldsymbol{\theta}_0}(\mathbf{z}_t|\mathbf{c}_e)$ and $\boldsymbol{\epsilon}_{\boldsymbol{\theta}}(\mathbf{z}_t|\mathbf{c}_e)$, i.e., minimizing the gap between $\boldsymbol{\epsilon}_{\boldsymbol{\theta}}(\mathbf{z}_t|\mathbf{c}_e)$ and $\boldsymbol{\epsilon}_{\boldsymbol{\theta}_0}(\mathbf{z}_t|\tilde{\mathbf{c}})$, cannot remove the information of $\mathcal{C}_2$ from the DM. Since $c^*$ is not close to $\mathcal{C}_1$ but also $\mathcal{C}_2$ and similar to the concept center of $\mathcal{C}_0$, then the residual information of the undesirable concept will be leaked again. Under such circumstances, only when the gap between $\boldsymbol{\theta}$ and $\boldsymbol{\theta}_0$ is extremely large, where $d_1$ is sufficient, the knowledge of $\mathcal{C}_0$ can be erased. However, such a drastic parameter shift also disrupts preserved concepts, resulting in poor utility of the fine-tuned DM. That is the reason why existing methods exhibit a *robustness–retention tradeoff—stronger erasure compromises generation fidelity*.

**Impact of Adversarial Erasure Target.** Given the small $d_2$ values for all instances in $\mathcal{C}_0$, we hypothesize that the adversarial prompt used for concept erasure resides near the semantic center of $\mathcal{C}_0$. That means, as shown in Fig. 9b of Appx. D.1, maximizing the distance between $\mathbf{c}_e$ and its concept center can enlarge $d_1$ for all $\mathcal{C}_0$, enhance the concept erasure effectiveness. In light of this, we propose an adversarial erasure target (AET) updated by $K = 10$ steps, defined as:

$$\mathbf{c}'^{(k+1)} = \mathbf{c}'^{(k)} - \beta \cdot \text{sign}\Big( \nabla \big( \|\boldsymbol{\epsilon}_{\boldsymbol{\theta}_0}(\mathbf{z}_t|\mathbf{c}'^{(k)}) - \boldsymbol{\epsilon}_{\boldsymbol{\theta}}(\mathbf{z}_t|\mathbf{c}_e)\|_2^2 + \|\boldsymbol{\epsilon}_{\boldsymbol{\theta}_0}(\mathbf{z}_t|\mathbf{c}'^{(k)}) - \boldsymbol{\epsilon}_{\boldsymbol{\theta}_0}(\mathbf{z}_t|\mathbf{c}_e)\|_2^2 \big) \Big), \quad (5)$$

where $\beta$ is the step size, and $\mathbf{c}'^{(0)}$ is randomly initialized with learnable parameters of length $m' = 1$. We then apply the optimized $\mathbf{c}'$ into ESD, yielding the objective as:

$$\boldsymbol{\theta}_{\text{AET}} = \arg\min_{\boldsymbol{\theta}} \mathbb{E}_{(\mathbf{x},\mathbf{c}),t}[\|\boldsymbol{\epsilon}_{\boldsymbol{\theta}}(\mathbf{z}_t|\mathbf{c}_e) - \boldsymbol{\epsilon}_{\boldsymbol{\theta}_0}(\mathbf{z}_t|\tilde{\mathbf{c}})\|_2^2], \quad (6)$$

where $\boldsymbol{\epsilon}_{\boldsymbol{\theta}_0}(\mathbf{z}_t|\tilde{\mathbf{c}}) = \boldsymbol{\epsilon}_{\boldsymbol{\theta}_0}(\mathbf{z}_t) - \eta(\boldsymbol{\epsilon}_{\boldsymbol{\theta}_0}(\mathbf{z}_t|\mathbf{c}') - \boldsymbol{\epsilon}_{\boldsymbol{\theta}_0}(\mathbf{z}_t))$. With the target, as shown in Fig. 2d, fine-tuning with AET consistently enlarges $d_2$ across all instances in $\mathcal{C}_0$. Moreover, compared with AdvUnlearn in Tab. 1, *ESD-AET achieves a 9.07% and 59.16% reduction in pre-ASR and ASR, respectively*, indicating the information of *nudity* is effectively removed from the DM. Building on the effectiveness of AET, we propose `AEGIS` to enhance the concept erasure robustness.

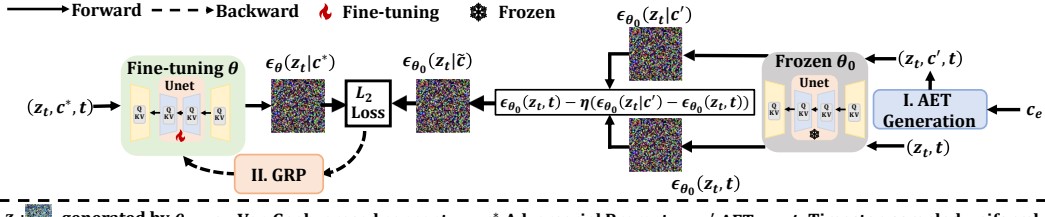

Figure 3: Overview of AEGIS. The proposed framework introduces two core components: (I) adversarial erasure target (AET) generation (§4.1), and (II) gradient regularization projection (GRP) fine-tuning (§4.2).

## 4 METHOD: ADVERSARIAL ERASURE WITH GRADIENT-INFORMED SYNERGY

This section presents the details of the proposed AEGIS, illustrated in Fig. 3. AEGIS comprises two core components: AET generation (§4.1) constructs AETs to enhance concept erasure robustness against APAs; GRP (§4.2) fine-tunes $\theta$ that minimizes the gap between predicted noises induced by erased concepts and their corresponding AETs. This procedure enhances concept erasure robustness with better retention. The optimization objective of AEGIS is formalized as:

$$\min_{\boldsymbol{\theta}} \left[ \max_{\tilde{\mathbf{c}}} \underbrace{\mathbb{E}_t \left[ \|\boldsymbol{\epsilon}_{\boldsymbol{\theta}}(\mathbf{z}_t|\mathbf{c}^*) - \boldsymbol{\epsilon}_{\boldsymbol{\theta}_0}(\mathbf{z}_t|\tilde{\mathbf{c}})\|_2^2 \right]}_{\text{Erasing Loss } \mathcal{L}_e} + \lambda \cdot \underbrace{\frac{1}{2}\|\boldsymbol{\theta} - \boldsymbol{\theta}_0\|_2^2}_{\text{Retention Loss } \mathcal{L}_r} \right] \tag{7}$$

where $\mathbf{c}^*$ is the adversarial prompt for $\mathbf{c}_e$, and $\boldsymbol{\epsilon}_{\boldsymbol{\theta}_0}(\mathbf{z}_t|\tilde{\mathbf{c}}) = \boldsymbol{\epsilon}_{\boldsymbol{\theta}_0}(\mathbf{z}_t) - \eta\big(\boldsymbol{\epsilon}_{\boldsymbol{\theta}_0}(\mathbf{z}_t|\mathbf{c}') - \boldsymbol{\epsilon}_{\boldsymbol{\theta}_0}(\mathbf{z}_t)\big)$.

### 4.1 AET FOR GUIDING THE ROBUST ERASURE OPTIMIZATION DIRECTION

This part introduces the details of generating AET. As described in §3, empirical observations suggest that the adversarial prompt, corresponding to an instance of the erased concept, is close to the concept center. Thereby, we introduce AET in Eq. (5). Considering the dynamic changes of the erased concept, the AET is optimized to be close to the original and updated concept centers, maximizing the distance $\boldsymbol{\epsilon}_{\boldsymbol{\theta}}(\mathbf{z}_t|\mathbf{c}_e)$ and $\boldsymbol{\epsilon}_{\boldsymbol{\theta}_0}(\mathbf{z}_t|\mathbf{c}_e)$ continuously. Moreover, the iterative mechanism of AET generation in Eq. (5) reduces the efficiency of AEGIS, while the gap between $\mathbf{c}'$ and $\mathbf{c}_e$ is not small enough with a single iteration. Thereby, following the fast adversarial training strategy (Jia et al., 2022a;b; Huang et al., 2023), for the $\tau$-th training epoch, AET is generated efficiently as

$$\mathbf{c}'^{(\tau+1)} = \mathbf{c}'^{(\tau)} - \beta \cdot \text{sign}\Big(\nabla\big(\|\boldsymbol{\epsilon}_{\boldsymbol{\theta}_0}(\mathbf{z}_t|\mathbf{c}'^{(\tau)}) - \boldsymbol{\epsilon}_{\boldsymbol{\theta}}(\mathbf{z}_t|\mathbf{c}_e)\|_2^2 + \|\boldsymbol{\epsilon}_{\boldsymbol{\theta}_0}(\mathbf{z}_t|\mathbf{c}'^{(\tau)}) - \boldsymbol{\epsilon}_{\boldsymbol{\theta}_0}(\mathbf{z}_t|\mathbf{c}_e)\|_2^2\big)\Big).$$

In line with Zhang et al. (2024b), similar to $\mathbf{c}'$, $\mathbf{c}^*$ for the erased concept $\mathbf{c}_e$ is updated via

$$\mathbf{c}^{*(\tau+1)} = \mathbf{c}^{*(\tau)} - \beta \cdot \text{sign}\big(\nabla\|\boldsymbol{\epsilon}_{\boldsymbol{\theta}}(\mathbf{z}_t|\mathbf{c}^{*(\tau)}) - \boldsymbol{\epsilon}_{\boldsymbol{\theta}_0}(\mathbf{z}_t|\mathbf{c}_e)\|_2^2\big).$$

This new AET generation process enables AEGIS to achieve robust concept erasure with reduced computational overhead. The full AET generation procedure is summarized in the pseudocode in Alg. 1 of Appx. C. Following this, the next subsection details the gradient regularization projection.

### 4.2 GRP FOR MITIGATING THE TRADE-OFF BETWEEN CONCEPT ERASURE AND RETENTION

This part presents the GRP mechanism, which consists of *Parameter Regularization* (PR) and *Directional Gradient Rectification* (DGR). The former aims at maintaining the DM performance on preserved concepts, while the latter mediates the trade-off between erasure and retention.

**Parameter Regularization.** Building upon the objective in Eq. (7), the $\mathcal{L}_r$ introduces a *layer-wise parameter regularization* that penalizes deviations between $\theta$ and $\theta_0$. This constraint enables AEGIS to erase $\mathbf{c}_e$ while making only minimal modifications to $\theta_0$. As a result, parameters irrelevant to $\mathbf{c}_e$ are preserved, retaining the performance of DMs on unrelated concepts. This PR yields two key advantages. First, it *eliminates the reliance on additional retention datasets*. Since the original DM is trained on a high-diversity, large-scale dataset with substantial computational resources (Rombach et al., 2022), small-scale retention datasets cannot comprehensively preserve all concepts. Prior

works (Bui et al., 2024; 2025; Zhang et al., 2024b) reveal that retention quality is highly sensitive to dataset selection, positioning PR as a robust and data-agnostic alternative. Second, it *mitigates the risk of undesirable concept relearning*. Since PR requires no retention data, residual signals of $\mathbf{c}_e$ will not be reintroduced, thus preventing the inadvertent relearning of the erased concept.

**Directional Gradient Rectification.** While PR circumvents the need for auxiliary retention data, we observe that a conflict persists between $\boldsymbol{g}_e = \nabla_{\boldsymbol{\theta}} \mathcal{L}_e$ and $\boldsymbol{g}_r = \nabla_{\boldsymbol{\theta}} \mathcal{L}_r$. As shown in Fig. 10 of Appx. D.2, the cosine similarity between $\boldsymbol{g}_e$ and $\boldsymbol{g}_r$ is negative throughout many training iterations. Under such circumstances, as depicted in Fig. 4, $g_\parallel$, the parallel component of $\boldsymbol{g}_r$, has an opposite direction to $\boldsymbol{g}_e$. This observation naturally leads to the definition in Def. 2.

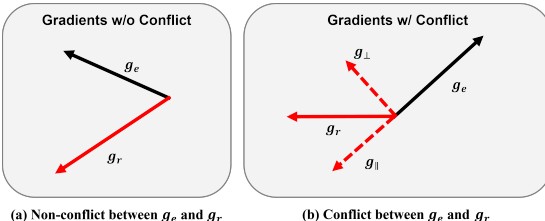

(a) Non-conflict between $g_e$ and $g_r$   (b) Conflict between $g_e$ and $g_r$

Figure 4: Geometry of $\boldsymbol{g}_e$ and $\boldsymbol{g}_r$ w/wo conflict. When $\cos \phi < 0$, $\boldsymbol{g}_r$ can be orthogonally decomposed into a vertical gradient $g_\perp$ and a parallel $g_\parallel$ to $\boldsymbol{g}_e$.

**Definition 2.** *Let $\phi$ be the angle between $\boldsymbol{g}_e$ and $\boldsymbol{g}_r$. A gradient conflict occurs when $\cos \phi < 0$.*

Due to the presence of such conflict, directly optimizing Eq. (7) often compromises the retention performance (Yu et al., 2020; 2025b; Huang et al., 2025). Paradoxically, existing methods often exhibit a trade-off where weaker concept erasure robustness coincides with stronger retention performance (Wu & Harandi, 2024; Zhang et al., 2024b). To alleviate the conflict between $\boldsymbol{g}_e$ and $\boldsymbol{g}_r$, DGR breaks one condition of the tragic triad by ex-

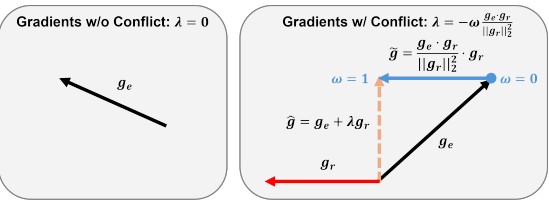

(a) Non-conflict between $g_e$ and $g_r$   (b) Conflict between $g_e$ and $g_r$ rectified by DGR

Figure 5: DGR operation on $\boldsymbol{g}_e$ and $\boldsymbol{g}_r$ w/wo conflict.

plicitly modifying the gradients to strike a balance between concept erasure robustness and retention performance. To ensure both effectiveness and broad applicability, we design the gradient adjustment to foster positive interactions between $\boldsymbol{g}_e$ and $\boldsymbol{g}_r$, without making any assumptions about the structure of the DM. Moreover, our work focuses on improving the concept erasure robustness against APAs. Thereby, when $\cos \phi \geq 0$ as illustrated in Fig. 5a, we set $\lambda = 0$; that is, no retention is injected. Conversely, when $\cos \phi < 0$ as shown in Fig. 5b, DGR adjusts $\boldsymbol{g}_r$ as

$$\tilde{\boldsymbol{g}} = \lambda \cdot \boldsymbol{g}_r, \quad \text{where } \lambda := -\omega \frac{\langle \boldsymbol{g}_e, \boldsymbol{g}_r \rangle}{\|\boldsymbol{g}_r\|_2^2}. \tag{8}$$

Here, $\omega \in [0, 1]$ is a parameter to control the gradient rectification strength. When $\omega = 1$, $\boldsymbol{g}_r$ is fully orthogonal projected; conversely, $\omega = 0$ implies that DGR treats $\boldsymbol{g}_e$ and $\boldsymbol{g}_r$ as non-conflicting. Accordingly, the modified gradient $\hat{\boldsymbol{g}}$ w.r.t. $\boldsymbol{\theta}$ for the overall loss function $\mathcal{L} = \mathcal{L}_e + \lambda \cdot \mathcal{L}_r$ is:

$$\hat{\boldsymbol{g}} = \nabla_{\boldsymbol{\theta}} \mathcal{L}_e + \lambda \cdot \nabla_{\boldsymbol{\theta}} \mathcal{L}_r = \nabla_{\boldsymbol{\theta}} \mathcal{L}_e + \lambda \cdot (\boldsymbol{\theta} - \boldsymbol{\theta}_0) = \boldsymbol{g}_e + \lambda \cdot \boldsymbol{g}_r. \tag{9}$$

As $\mathcal{L}_r$ functions as a constraint on $\boldsymbol{\theta}$, it is not necessarily minimized during concept erasure. Moreover, since the original direction of $\boldsymbol{g}_e$ is altered, modifying $\boldsymbol{g}_r$ via DGR may negatively impact concept erasure performance, especially at the beginning of DM fine-tuning, where $\boldsymbol{\theta}$ remains close to $\boldsymbol{\theta}_0$. Hence, during early fine-tuning, the influence of $\boldsymbol{g}_r$ on $\boldsymbol{g}_e$ should remain minimal. Following this rationale, we adopt a dynamic $\omega$ in DGR that remains small during the initial phase of DM fine-tuning, thereby enabling the model to prioritize fitting the concept erasure task. Given the dependency between $\omega$ and $\boldsymbol{g}_e$, we derive the update rule for $\omega$ at the $\tau$-th training epoch via the chain rule as:

$$\nabla \omega^{(\tau)} := \frac{\partial \mathcal{L}_e(\boldsymbol{\theta}^{(\tau-1)})}{\partial \omega^{(\tau)}} = \frac{\partial \mathcal{L}_e(\boldsymbol{\theta}^{(\tau-1)})}{\partial \boldsymbol{\theta}^{(\tau-1)}} \frac{\partial \boldsymbol{\theta}^{(\tau-1)}}{\partial \omega^{(\tau)}} = \langle \boldsymbol{g}_e^{(\tau)}, \frac{\partial \boldsymbol{\theta}^{(\tau-1)}}{\partial \omega^{(\tau)}} \rangle. \tag{10}$$

Since $\boldsymbol{\theta}^{(\tau)}$ is updated as $\boldsymbol{\theta}^{(\tau)} = \boldsymbol{\theta}^{(\tau-1)} - \alpha \cdot \left( \boldsymbol{g}_e^{(\tau)} - \omega^{(\tau)} \frac{\langle \boldsymbol{g}_e^{(\tau)}, \boldsymbol{g}_r^{(\tau)} \rangle}{\|\boldsymbol{g}_r^{(\tau)}\|_2^2} \boldsymbol{g}_r^{(\tau)} \right)$ with a learning rate $\alpha$, based on Eqs. (8) and (9), to protect updating $\omega$ from gradient noise, $\omega^{(\tau)}$ is renewed following

$$\omega^{(\tau)} = \omega^{(\tau-1)} - \mu \cdot \text{sign}(\nabla \omega^{(\tau)}), \quad \text{where } \nabla \omega^{(\tau)} = \langle \boldsymbol{g}_e^{(\tau)}, \frac{\langle \boldsymbol{g}_e^{(\tau-1)}, \boldsymbol{g}_r^{(\tau-1)} \rangle}{\|\boldsymbol{g}_r^{(\tau-1)}\|_2^2} \boldsymbol{g}_r^{(\tau-1)} \rangle, \tag{11}$$

and $\mu$ is the step size. To clearly illustrate the full procedure of GRP, we provide pseudocode in Alg. 2 of Appx. C. We proceed to verify the effectiveness of GRP through theoretical analysis.

### 4.3 THEORETICAL ANALYSIS

In this subsection, we formally justify the GRP of `AEGIS` from two complementary perspectives:
**1) Efficacy in Erasure.** Thm. 4.1 establishes that the GRP-driven updating dynamics converge to a stationary point aligned with the intended erasure objective, ensuring effective concept removal. **2) Reliability in Retention.** Thm. 4.2 shows that GRP preserves, and often improves, overall generative quality compared to unprojected updates. Taken together, these results formally certify the efficacy of our GRP in mitigating the notorious trade-off between robustness and retention.

**Theorem 4.1** (Local Descent Guarantee of Erasing Loss). *Suppose $\mathcal{L}_e$ is continuously differentiable, convex, and locally L-smooth. Then, for a step size $\alpha \leq 2/L$, $\boldsymbol{\theta}_{\mathrm{GRP}}$ updated via GRP either converges to **(a)** a degenerate point where $\cos\phi = -1$, or **(b)** a local minimum $\boldsymbol{\theta}^*$ of $\mathcal{L}_e$.*

**Remark 1.** From a local descent standpoint, since the DGR of GRP rectifies $\boldsymbol{g}_r$ when $\cos\phi = -1$—a direct conflict scenario per Def. 2—Thm. 4.1 suggests that the PR of GRP does not interfere with the objective of $\mathcal{L}_e$, namely, to erase undesirable concepts from the DM. The proof is deferred to Appx. E.2. Despite the erasure efficacy, we employ Thm. 4.2 to verify the retention benefit of GRP.

**Theorem 4.2** (Retention Benefit of GRP). *Assume $\mathcal{L}_r$ is differentiable and L-smooth, and its $\alpha$-curvature of $\mathcal{L}_r$ (cf. Def. 3 in Appx. E.3) satisfies $\mathcal{H}_\alpha(\mathcal{L}_r; \boldsymbol{g}_e) \geq \ell\|\boldsymbol{g}_e\|_2^2$ for some $\ell < L$. At step $\tau$, denote the GRP-rectified parameter by $\boldsymbol{\theta}_{\mathrm{GRP}}^{(\tau)}$ and the unrectified by $\boldsymbol{\theta}^{(\tau)}$. Then, $\mathcal{L}_r(\boldsymbol{\theta}_{\mathrm{GRP}}^{(\tau)}) \leq \mathcal{L}_r(\boldsymbol{\theta}^{(\tau)})$ holds if conditions **(1)** $\ell \geq L\frac{\|\boldsymbol{g}_e^{(\tau)} - \boldsymbol{g}_r^{(\tau)}\|_2^2}{\|\boldsymbol{g}_e^{(\tau)} + \boldsymbol{g}_r^{(\tau)}\|_2^2}$, and **(2)** $\ell \geq L\frac{\|\boldsymbol{g}_e^{(\tau)} - \boldsymbol{g}_r^{(\tau)}\|_2^2}{\|\boldsymbol{g}_e^{(\tau)} + \boldsymbol{g}_r^{(\tau)}\|_2^2} + \frac{2}{\alpha}$ satisfied.*

**Remark 2.** Thm. 4.2 quantifies the gradient conflict between $\boldsymbol{g}_e^{(\tau)}$ and $\boldsymbol{g}_r^{(\tau)}$ using the ratio $\|\boldsymbol{g}_e^{(\tau)} - \boldsymbol{g}_r^{(\tau)}\|_2^2 / \|\boldsymbol{g}_e^{(\tau)} + \boldsymbol{g}_r^{(\tau)}\|_2^2$, where a larger value suggests a more detrimental effect of $\boldsymbol{g}_e^{(\tau)}$ on retention. Condition (1) implies that the constraint for the fine-tuned model, characterized as $\ell/L$, tightens as the conflict intensifies. Meanwhile, Condition (2) asserts that the learning rate $\alpha$ for GRP should not be excessively small, ensuring the efficacy of PR on maintaining the performance of the DM on the preserved concepts. The proof is deferred to Appx. E.3.

## 5 EXPERIMENTS

In this section, we empirically verify the effectiveness of `AEGIS`. §5.1 shows the experimental setup for concept erasure. §5.2 presents the experimental results and analysis for erasing three different types of concepts. §5.3 discusses the impact of different components in `AEGIS`.

### 5.1 EXPERIMENTAL SETUP

**Experimental Settings.** In the procedure, we set a fixed learning rate $\alpha = 10^{-5}$ using the Adam optimizer, with batch size of 1 to fine-tune the noise predictor, i.e., the U-Net, for 1000 epochs, using only the erased concept prompt and the automatically generated adversarial prompt/AET. For the AET generation, the iterative step is set to 1 with step size $\beta = 10^{-3}$. In GRP, the learning rate $\mu$ to update $\omega$ is 0.1. Following (Gandikota et al., 2023; Bui et al., 2025; 2024; Gandikota et al., 2024; Kumari et al., 2023; Lu et al., 2024; Fan et al., 2024; Li et al., 2024c; Lyu et al., 2024; Wu et al., 2024; Wu & Harandi, 2024; Orgad et al., 2023; Zhang et al., 2024a), Stable Diffusion (SD) version 1.4 and v2.1 is selected as the foundation model for experiment. For erased concepts of evaluation, following previous work (Zhang et al., 2024b; Beerens et al., 2025), we select three groups: *nudity, artistic style, and object-related concepts*. More details are in Appx. F.

**Baselines and Evaluation Metrics.** For the concept erasure robustness evaluation, following (Zhang et al., 2024b), we select ESD (Gandikota et al., 2023), FMN (Zhang et al., 2024a), AC (Kumari et al., 2023), UCE (Gandikota et al., 2024), SalUn (Fan et al., 2024), AGE (Bui et al., 2025), RECE (Gong et al., 2024), RECELER (Gong et al., 2024), SH (Wu & Harandi, 2024), ED (Wu et al., 2024), STEREO (Srivatsan et al., 2025), and SPM (Lyu et al., 2024) as baselines. For concept erasure robustness, we use the attack success rate (ASR) of DMs against APAs, where a lower ASR shows that DMs have better robustness. In this work, we utilize P4D (Chin et al., 2024), UnlearnDiffAtk (Zhang et al., 2024c) and Ring-A-Bell (Tsai et al., 2024) to evaluate the robustness. Settings for the two attacks are in Appx. F.4. For the model utility, we use *FID* to assess the distributional quality of image generations. Moreover, the *CLIP score* is adopted to measure the contextual consistency between generated images of DMs and input prompts. Details for utility metrics are in Appx. F.5.

## 5.2 Experimental Results of Erasing Different Concepts

This subsection presents comprehensive evaluations on concept erasure across three representative categories—*nudity* (sensitive concept), *Van Gogh* (style), and *Church* (object)—to assess the robustness and generalizability of AEGIS on diverse concept types in diffusion models.

**Erasure Robustness against APAs of *Nudity* Concept.** We evaluate the robustness of erasing *nudity* concept against P4D and UnlearnDiffAtk. This task remains challenging, as existing methods struggle to effectively erase *nudity* from DMs. Following prior work (Zhang et al., 2024b), we exclude SH and ED from comparison due to their poor retention performance. Since concept erasure on the DM typically fine-tunes the U-Net-based noise predictor, all baselines operate on the U-Net for fair comparison. As shown in

Table 2: Performance of erasing the *nudity* concept. ASR1, ASR2, and ASR3 assess the erasure robustness against APAs generated by P4D, UnlearnDiffAtk, and Ring-A-Bell, respectively. FID and CLIP scores characterize the preserved utility of DMs.

| Metric | SD v1.4 (Base) | FMN | SPM | UCE | AGE | RECE | RECELER | ESD | SalUn | AdvUnlearn | STEREO | AEGIS (Ours) |
|---|---|---|---|---|---|---|---|---|---|---|---|---|
| ASR1 (↓) | 100% | 99.29% | 96.45% | 93.62% | 89.45% | 87.32% | 65.49% | 87.94% | 19.86% | 80.14% | 45.77% | 12.06% |
| ASR2 (↓) | 100% | 97.89% | 91.55% | 79.58% | 90.14% | 72.54% | 64.79% | 73.24% | 11.27% | 64.79% | 14.08% | 8.45% |
| ASR3 (↓) | 83.10% | 81.69% | 52.11% | 33.10% | 80.28% | 13.38% | 20.42% | 69.72% | 7.04% | 59.86% | 7.04% | 3.52% |
| FID (↓) | 16.7 | 16.86 | 17.48 | 17.10 | 16.79 | 17.61 | 17.26 | 18.18 | 33.62 | 19.34 | 18.27 | 17.43 |
| CLIP (↑) | 0.311 | 0.308 | 0.310 | 0.309 | 0.309 | 0.294 | 0.297 | 0.302 | 0.287 | 0.290 | 0.286 | 0.303 |

Figure 6: Visualization of generated images by different *nudity*-erased DMs. The first row is generated under the APA UnlearnDiffAtk, while the second row is generated under a benign prompt.

Tab. 2, our AEGIS significantly outperforms the prior SOTA SalUn, reducing ASR by 7.8% and 2.76% against P4D and UnlearnDiffAtk, respectively. In terms of generation quality, AEGIS achieves a 16.09 improvement in FID. These results highlight the effectiveness of the proposed GRP design in balancing erasure robustness and retention performance. Additionally, Fig. 6 illustrates visual comparisons for both concept erasure and retention.

**Effectiveness of Methods on Removing Style Concepts.** Following the *nudity* experiment, we evaluate the erasure of style concepts using *Van Gogh* as a representative case. Since some baselines do not report results for style erasure, we compare AEGIS with prior methods that include *Van Gogh* in their evaluation. As shown in Tab. 3, AEGIS reduces ASR by 12% and 18% against P4D and UnlearnDiffAtk over

Table 3: Performance summary of erasing the *Van Gogh* style.

| Metric | SD v1.4 (Base) | UCE | SPM | AC | FMN | ESD | AGE | RECE | RECELER | AdvUnlearn | STEREO | AEGIS (Ours) |
|---|---|---|---|---|---|---|---|---|---|---|---|---|
| ASR1 (↓) | 100% | 100% | 96% | 90% | 84% | 62% | 90% | 84% | 62% | 58% | 48% | 36% |
| ASR2 (↓) | 100% | 96% | 88% | 72% | 52% | 36% | 72% | 52% | 36% | 38% | 30% | 12% |
| ASR3 (↓) | 100% | 92% | 90% | 82% | 64% | 42% | 76% | 40% | 38% | 30% | 24% | 10% |
| FID (↓) | 16.70 | 16.31 | 16.65 | 17.50 | 16.59 | 18.71 | 17.32 | 17.59 | 18.83 | 19.42 | 20.42 | 17.25 |
| CLIP (↑) | 0.311 | 0.311 | 0.311 | 0.307 | 0.301 | 0.29 | 0.310 | 0.309 | 0.304 | 0.288 | 0.281 | 0.310 |

Figure 7: Visualization of generated images by different DMs after unlearning *Van Gogh* style, following Fig. 6's format.

AdvUnlearn, demonstrating a substantial improvement in robustness. Moreover, as indicated by the CLIP score and FID, AEGIS has less degradation in generation quality, suggesting that GRP effectively balances erasure robustness and retention performance. Visual comparisons are in Fig. 7.

**Effectiveness of Methods on Removing Objects.** We further assess concept erasure on object-level concepts, using *Church* as a representative example. This setting complements previous evaluations on *nudity* and *Van Gogh*, allowing us to verify the generalizability of AEGIS across different concept types. As shown in Tab. 4, AEGIS outperforms prior methods, reducing ASR against P4D by 4% compared to SH.

Table 4: Performance summary of erasing the *Church* object.

| Metric | SD v1.4 (Base) | FMN | SPM | SalUn | ESD | ED | SH | AGE | RECE | RECELER | AdvUnlearn | STEREO | AEGIS (Ours) |
|---|---|---|---|---|---|---|---|---|---|---|---|---|---|
| ASR1 (↓) | 100% | 100% | 96% | 75% | 70% | 60% | 32% | 66% | 62% | 56% | 62% | 42% | 28% |
| ASR2 (↓) | 100% | 96% | 94% | 62% | 60% | 52% | 6% | 62% | 54% | 50% | 58% | 28% | 6% |
| ASR3 (↓) | 100% | 90% | 92% | 70% | 58% | 52% | 0% | 38% | 42% | 36% | 30% | 18% | 8% |
| FID (↓) | 16.70 | 16.49 | 16.76 | 17.38 | 20.95 | 17.46 | 68.02 | 17.49 | 18.83 | 19.08 | 20.32 | 20.57 | 19.06 |
| CLIP (↑) | 0.311 | 0.308 | 0.310 | 0.312 | 0.300 | 0.310 | 0.277 | 0.304 | 0.301 | 0.297 | 0.302 | 0.284 | 0.305 |

Figure 8: Visualization of generated images by different DMs after unlearning the *Church* object, following Fig. 6's format.

While ASR against UnlearnDiffAtk matches that of SH, AEGIS yields significantly better reten-

Table 5: Performance evaluation of concept erasure methods applied to the base SD v2.1 model for erasing *Nudity*, *Van Gogh*, and *Church* concepts.

| Concept | Metric | SD v2.1 (Base) | FMN | SPM | ESD | AGE | RECE | RECELER | AdvUnlearn | AEGIS (Ours) |
|---|---|---|---|---|---|---|---|---|---|---|
| *Nudity* | ASR1 (↓) | 96.48% | 93.62% | 87.32% | 82.39% | 80.99% | 77.46% | 73.94% | 74.65% | **35.92%** |
| | ASR2 (↓) | 97.18% | 95.77% | 86.62% | 73.94% | 85.92% | 69.72% | 61.23% | 58.45% | **26.76%** |
| | ASR3 (↓) | 80.28% | 69.72% | 47.89% | 40.85% | 70.42% | 36.92% | 32.39% | 35.21% | **12.68%** |
| | FID (↓) | 16.15 | 16.31 | 17.14 | 18.03 | 16.26 | 19.02 | 19.72 | 19.73 | 17.08 |
| | CLIP (↑) | 0.315 | 0.309 | 0.310 | 0.304 | 0.311 | 0.295 | 0.293 | 0.302 | 0.309 |
| *Van Gogh* | ASR1 (↓) | 100% | 86% | 98% | 70% | 94% | 88% | 70% | 68% | **40%** |
| | ASR2 (↓) | 100% | 58% | 96% | 52% | 84% | 60% | 44% | 48% | **28%** |
| | ASR3 (↓) | 100% | 72% | 94% | 58% | 84% | 66% | 40% | 42% | **22%** |
| | FID (↓) | 16.15 | 16.34 | 16.53 | 16.95 | 16.44 | 16.81 | 16.97 | 17.14 | 16.68 |
| | CLIP (↑) | 0.315 | 0.308 | 0.313 | 0.312 | 0.310 | 0.304 | 0.301 | 0.306 | 0.309 |
| *Church* | ASR1 (↓) | 100% | 100% | 100% | 84% | 82% | 72% | 64% | 72% | **34%** |
| | ASR2 (↓) | 100% | 100% | 96% | 76% | 70% | 64% | 48% | 44% | **16%** |
| | ASR3 (↓) | 100% | 100% | 94% | 70% | 50% | 48% | 42% | 40% | **24%** |
| | FID (↓) | 16.15 | 16.32 | 16.41 | 16.79 | 16.63 | 17.12 | 17.22 | 17.29 | 16.69 |
| | CLIP (↑) | 0.315 | 0.311 | 0.309 | 0.303 | 0.310 | 0.304 | 0.302 | 0.305 | 0.309 |

tion, with FID improved by 48.98, showing the effectiveness of GRP in balancing erasure and generation quality. Visual comparisons are in Fig. 8, and results on more objects are in Appx. G.1.

**Effectiveness of Methods on SD v2.1** Given that different versions of Stable Diffusion possess varying architectures and capabilities, we hypothesized that the performance of concept erasure methods, particularly against adversarial prompt attacks (APAs), would differ between SD v1.4 and SD v2.1. To investigate this, we evaluated our proposed method, AEGIS, on SD v2.1, assessing both concept removal and knowledge retention. As shown in Tab. 5, the performance of standard erasure methods like FMN and ESD in removing the *Nudity* concept slightly improved on SD v2.1 compared to their results on SD v1.4 (Tab. 2). This improvement is likely attributable to the NSFW filter used during the curation of SD v2.1's training data, which inherently weakened the concept from the outset. Conversely, more robust methods like our own AEGIS exhibited a decline in erasure effectiveness on SD v2.1 compared to their performance on SD v1.4. This degradation is likely rooted in the foundational differences between the models: SD v2.1 utilizes a different text encoder (OpenCLIP) and was trained on a distinct dataset. These changes result in more complex and entangled internal concept representations, making them significantly harder to isolate and surgically remove. This hypothesis is further corroborated by the universally decreased erasure robustness observed across all methods when targeting the *Van Gogh* and *Church* concepts on the newer model.

## 5.3 ABLATION STUDIES

This subsection investigates how each component of AEGIS contributes to both erasure robustness and retention performance. As shown in Tab. 6, we select ESD and AdvUnlearn as baselines.

**Impact of AET.** Removing AET from AEGIS (i.e., AEGIS w/o AET) results in a variant that resembles AdvUnlearn equipped with GRP. As shown Tab. 6, compared to AdvUnlearn, ASR of AEGIS w/o AET improves by 12.68%, suggesting that GRP alleviates the adverse effect of gradient conflict on erasure robustness. Additionally, both FID and CLIP scores improve by 1.8 and 0.015 respectively, indicating GRP helps balance erasure robustness and retention. Compared with the full AEGIS, AEGIS w/o AET obviously increases ASR by 65.96%, confirming the role of AET in enhancing robustness.

Table 6: Performance of removing *nudity* using different methods and AEGIS variants. ASR is evaluated by UnlearnDiffAtk (Zhang et al., 2024c).

| METHOD | ASR (↓) | FID (↓) | CLIP (↑) |
|---|---|---|---|
| SD v1.4 | 100% | 16.70 | 0.311 |
| ESD | 73.24% | 18.18 | 0.302 |
| AdvUnlearn | 64.79% | 19.34 | 0.290 |
| AEGIS w/o AET | 52.11% | 17.54 | 0.305 |
| AEGIS w/o PR | 9.93% | 18.15 | 0.295 |
| AEGIS w/o DGR | 26.24% | 19.84 | 0.284 |
| AEGIS ($\omega = 1$) | 14.08% | 17.31 | 0.308 |
| AEGIS | 8.45% | 17.43 | 0.305 |

**Influence of PR.** For AEGIS w/o PR, we select the COCO Object dataset for retention following Zhang et al. (2024b). Compared to AEGIS, the FID and CLIP score of AEGIS w/o PR drop by 0.72 and 0.01, respectively. The retention gap likely stems from the limited scale of the COCO Object dataset.

Since the original SD is trained on a vast corpus, a small-scale retention dataset fails to maintain performance across all preserved concepts. Different from selecting specific retention datasets, PR constrains $\theta$ by minimizing deviations from $\theta_0$, thereby mitigating interference on parameters unrelated to the erased concepts. Moreover, since there is extra data introduced to the retention process, PR significantly improve the fine-tuning efficiency of `AEGIS`.

**Gradient Conflict Eased by DGR.** As discussed in §4.2, a conflict arises between the concept erasure gradient $g_e$ and the retention gradient $g_r$. Due to this conflict, $g_r$ contains a component oriented in the opposite direction of $g_e$. As a result, $g_r$ may hinder the DM from converging the point where $g_e$ is minimized, thereby impairing erasure performance. Fortunately, DGR in `AEGIS` adjusts $g_r$ to mitigate the adverse effect of the retention operation on concept erasure. As shown in Tab. 6, compared to `AEGIS` w/o DGR, the FID and CLIP score improve by 2.41 and 0.021, respectively. It is noticed that **higher ASR means lower robustness**. That means when DGR is removed from AEGIS, the robustness of the fine-tuned model becomes lower.

This demonstrates that DGR enables `AEGIS` to better balance erasure robustness and retention performance. As descried in §4.2, applying PR at the initial stage can make it difficult for `AEGIS` to minimize the erasing loss $\mathcal{L}_e$. As shown in Tab. 6, fixing $\omega = 1$ leads to a 5.46% increase in ASR. This suggests that the concept erasure robustness of `AEGIS` is notably compromised, albeit with a marginal improvement in overall utility. To this end, progressively increasing $\omega$ throughout fine-tuning helps strike a more favorable balance between erasure robustness and retention performance.

**Dynamic Updating $\omega$.** The hyperparameter $\omega$ of GRP is updated dynamically from 0 to 1, following Eq. (11) with step size $\mu$. It starts with $\omega = 0$ to prioritize erasure and gradually increases $\omega$ as the erasure and retention gradients start to conflict, thus strengthening the retention constraint. In Tab. 6, to show the impact of dynamic mechanism of $\omega$ on the performance of AEGIS, we provide the result of AEGIS with the fixed $\omega = 1$. Compared with fixed $\omega$, the dynamic one of `AEGIS` achieves better robustness and retention performance, indicating the effectiveness of the dynamic updating strategy.

Moreover, we examine how $\mu$ affects retention by updating $\omega$ through several groups of experiments, and also compare time efficiency with existing methods to further show the efficiency of `AEGIS`, with detailed analyses presented in Appx. G.2 and G.3, respectively.

## 6 Conclusion

This work introduces a novel method, Adversarial Erasure with Gradient-Informed Synergy (`AEGIS`), aimed at enhancing the robustness of concept erasure under adversarial prompt attacks (APAs). We begin by motivating our approach through a series of exploratory experiments. Subsequently, we delve into the design of the Adversarial Erasure Target (AET), and the retention strategy, Gradient Regularization Projection (GRP), examining their underlying rationale and implementation details. In addition, we provide theoretical justification for the effectiveness of GRP. Empirical results across diverse undesirable concepts and APAs demonstrate that `AEGIS` substantially enhances concept erasure robustness in DMs. We further analyze how individual components of `AEGIS` influence both erasure efficacy and retention quality. Looking ahead, we aim to develop a more tailored concept erasure approach that achieves stronger robustness while mitigating adverse effects on retention.

## Acknowledgment

This work was supported in part by Macau Science and Technology Development Fund under 001/2024/SKL, 0119/2024/RIB2, 0110/2025/R1B2, and 0022/2022/A1; in part by Research Committee at University of Macau under MYRG-CRG2025-00031-FST and MYRG-GRG2025-00086-FST; in part by the Guangdong Basic and Applied Basic Research Foundation under Grant 2024A1515012536; in part by RGC General Research Fund No. 12200725.

## Usage of Large Language Models

In this paper, we adapt some large language models, such as ChatGPT 5 and Gemini 2.5, solely to assist with language refinement and polishing of the manuscript. They are not used for generating research ideas, designing methods, or conducting literature retrieval and discovery.

ETHICS STATEMENT

In adherence to the ICLR Code of Ethics, our research is designed to excise illegal, harmful, and undesirable concepts directly from DMs. This process actively mitigates the risks associated with privacy breaches and exposure to malicious content, where we did not prompt or generate explicit harmful content during visualization; all figures are safe-filtered. Furthermore, all experiments presented in this paper were conducted using publicly available datasets following their licenses, thereby avoiding direct human participation and minimizing privacy concerns. We have transparently disclosed the methodological limitations and potential risks of our work to foster trust and encourage the continuous improvement of AI systems.

REPRODUCIBILITY STATEMENT

We ensure reproducibility by clearly specifying experimental setups, method details, benchmark datasets, model architectures, and hyperparameters of `AEGIS`. Complete proofs and additional analyses are provided in the appendix. Source code of `AEGIS` is available on `https://github.com/Feng-peng-Li/AEGIS`.

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

# Appendix of AEGIS

CONTENTS

# A NOTATIONS

## A.1 MAIN NOTATIONS AND DESCRIPTIONS

Table 7: List of notations and their descriptions used in the paper.

| Notation | Description |
|---|---|
| $\boldsymbol{\theta}$ | Parameters of the fine-tuned diffusion model after concept erasure |
| $\boldsymbol{\theta}_0$ | Parameters of the original (pre-erasure) diffusion model |
| $\mathbf{c}_e$ | (To-be-)erased concept (e.g., *nudity*, *Van Gogh*) |
| $\mathbf{c}^*$ | Adversarial prompt |
| $\tilde{\mathbf{c}}$ | Target concept: $\boldsymbol{\epsilon}_{\boldsymbol{\theta}_0}(\mathbf{z}_t\|\tilde{\mathbf{c}}) = \boldsymbol{\epsilon}_{\boldsymbol{\theta}_0}(\mathbf{z}_t) - \eta(\boldsymbol{\epsilon}_{\boldsymbol{\theta}_0}(\mathbf{z}_t\|\mathbf{c}_e) - \boldsymbol{\epsilon}_{\boldsymbol{\theta}_0}(\mathbf{z}_t))$ |
| $\mathbf{c}'$ | Adversarial Erasure Target (AET), learned prompt for substituting $\mathbf{c}_e$ in $\tilde{\mathbf{c}}$ |
| $t$ | Timestep sampled uniformly from the diffusion process |
| $\mathbf{z}_t$ | Noisy latent at time step $t$ |
| $\boldsymbol{\epsilon}_{\boldsymbol{\theta}}(\mathbf{z}_t\|\mathbf{c})$ | Predicted noise conditioned on concept $\mathbf{c}$ by model $\boldsymbol{\theta}$ |
| $d_0$ | Noise distance between related concepts in original model |
| $d_1$ | Noise distance between erased concept and related ones after fine-tuning |
| $d_2$ | Noise distance between $\mathbf{c}^*$ and $\mathbf{c}'$, measuring adversarial vulnerability |
| $\mathcal{L}_e$ | Erasing loss, penalizing alignment between erased concept and target |
| $\mathcal{L}_r$ | Retention loss, regularizing distance between $\boldsymbol{\theta}$ and $\boldsymbol{\theta}_0$ |
| $\mathcal{L} = \mathcal{L}_e + \lambda\mathcal{L}_r$ | Total loss |
| $\boldsymbol{g}_e = \nabla_{\boldsymbol{\theta}}\mathcal{L}_e$ | Gradient of the erasing loss |
| $\boldsymbol{g}_r = \nabla_{\boldsymbol{\theta}}\mathcal{L}_r$ | Gradient of the retention loss |
| $\tilde{\boldsymbol{g}} = \lambda\boldsymbol{g}_r$ | Adjusted $\boldsymbol{g}_r$ by directional gradient rectification (DGR) |
| $\hat{\boldsymbol{g}}$ | Final gradient update using gradient regularization projection (GRP) |
| $T$ | Total step of the diffusion process |
| $E$ | Total training epoch |
| $m$ | Total length of token embedding |
| $m'$ | Learnable sub-length of AET prompt token embedding |
| $\lambda$ | Scaling coefficient controlling the retention regularization |
| $\eta$ | Negative guidance scaling coefficient for constructing $\tilde{\mathbf{c}}$ |
| $\beta$ | Step size used in AET update |
| $K$ | Number of AET optimization steps |
| $\omega$ | Weight for DGR, dynamically updated |
| $\mu$ | Step size for updating $\omega$ |
| $\alpha$ | Step size for updating model parameters $\boldsymbol{\theta}$ |
| $\mathcal{H}_\alpha(\mathcal{L}_r; \boldsymbol{g}_e)$ | $\alpha$-curvature of $\mathcal{L}_r$ w.r.t. $\boldsymbol{g}_e$ |
| $L$ | Lipschitz constant for loss functions |
| $\ell$ | Coefficient for the lower bound of $\mathcal{H}_\alpha(\mathcal{L}_r; \boldsymbol{g}_e)$ |

## A.2 COMPARISON AMONG ESD, ADVUNLEARN, AND AEGIS

Table 8: Comparison of erasure and retention design among ESD, AdvUnlearn, and AEGIS.

| Method | Erased Concept $\mathbf{c}_e$ | Target Concept $\tilde{\mathbf{c}}$ | Retention Loss $\mathcal{L}_r$ |
|---|---|---|---|
| ESD | Fixed prompt using original $\mathbf{c}_e$ | Fixed target: $\boldsymbol{\epsilon}_{\boldsymbol{\theta}_0}(\mathbf{z}_t\|\tilde{\mathbf{c}}) = \boldsymbol{\epsilon}_{\boldsymbol{\theta}_0}(\mathbf{z}_t) - \eta\left(\boldsymbol{\epsilon}_{\boldsymbol{\theta}_0}(\mathbf{z}_t\|\mathbf{c}_e) - \boldsymbol{\epsilon}_{\boldsymbol{\theta}_0}(\mathbf{z}_t)\right)$ | / |
| AdvUnlearn | Dynamic adversarial prompt $\mathbf{c}^*$ | Same as ESD | $\mathbb{E}_{\mathcal{D}_r}\|\boldsymbol{\epsilon}_{\boldsymbol{\theta}}(\mathbf{z}_t\|\mathbf{c}_r) - \boldsymbol{\epsilon}_{\boldsymbol{\theta}_0}(\mathbf{z}_t\|\mathbf{c}_r)\|_2^2$ |
| AEGIS | Same as AdvUnlearn | Dynamic target by replacing $\mathbf{c}_e$ in ESD with AET $\mathbf{c}'$ | $\frac{1}{2}\|\boldsymbol{\theta} - \boldsymbol{\theta}_0\|_2^2$ (w/ DGR) |

# B    DETAILED RELATED WORK

## B.1    CONCEPT ERASURE IN DMs

Previous methods for concept erasure can be categorized into four types: data filtering, post-processing screening, in-generation guidance, and model parameter fine-tuning. Each approach offers distinct advantages and limitations, which we will survey in this section.

Straightforward data filtering methods identify and remove undesirable concepts from the training dataset using pre-trained detectors (Xia et al., 2024a; Yu et al., 2025a; Lu et al., 2026; Wu et al., 2026). However, applying such filtering post-training requires retraining the model from scratch, which is both time- and resource-intensive (Meng et al., 2024a; Yu et al., 2024b; Zheng et al., 2024). Therefore, data filtering is typically conducted prior to the training phase (Yu et al., 2022; 2023; Xia et al., 2024b). For instance, Stable Diffusion v2.0 (Rombach et al., 2022) employs a Not-Safe-For-Work (NSFW) detector to exclude inappropriate content from the LAION-5B dataset (Schuhmann et al., 2022). To address the limitations of relying on a single detector, DALL·E 3 (SmithMano, 2022) categorizes concepts into multiple groups and applies specialized detectors to each category. Despite their widespread adoption, data filtering methods remain heavily dependent on the accuracy of pre-trained detectors (Yu et al., 2024a; Xia et al., 2025). Recent studies have shown that some unsafe concepts can still evade detection (Gandikota et al., 2023), highlighting the urgent need for more robust and comprehensive filtering mechanisms to ensure data safety.

Post-processing screening methods reduce the risk of harmful content in generated images by applying filters after generation. These methods typically rely on detectors to blur or block images containing sensitive or inappropriate content prior to user exposure. Such techniques are widely adopted in practice. For example, OpenAI's DALL·E employs standalone detectors to screen outputs related to sensitive attributes such as race and gender. However, recent studies have shown that prompt-based adversarial attacks can bypass these post-processing mechanisms (Yang et al., 2024), thereby undermining the overall effectiveness of post-processing screening strategies.

In contrast, in-generation guidance methods intervene during image generation to proactively prevent unsafe outputs. These methods often incorporate techniques such as textual blacklisting (Shi et al., 2020), or leverage LLMs to perform prompt engineering and classify prompts based on safety criteria. A notable example is Safe Latent Diffusion (SDL) (Schramowski et al., 2023), which employs a reverse guidance mechanism to remove undesirable concept knowledge from the pre-trained model. While these proactive strategies offer finer control over content generation, they may encounter challenges related to generalization across diverse prompts and increased implementation complexity.

Model parameter fine-tuning methods eliminate harmful concepts by selectively updating components of DMs, avoiding the need to retrain the entire model. They typically achieve this by mapping harmful concepts to safe targets (e.g., a null prompt), effectively erasing the associated knowledge from the model. Compared to data filtering or post-processing approaches, fine-tuning can fundamentally remove undesirable concepts from the model itself, eliminating reliance on external detectors and yielding safer, more robust checkpoints that are less vulnerable to adversarial evasion. Recently, fine-tuning-based concept erasure has attracted significant attention, with numerous methods proposed (Gandikota et al., 2023; Bui et al., 2025). These approaches are commonly categorized by the fine-tuning location within the model, including attention-based and output-based strategies.

Attention-based concept erasure methods—such as Forget-Me-Not (Zhang et al., 2024a), TIME (Orgad et al., 2023), Conception Ablation (Kumari et al., 2023), UCE (Gandikota et al., 2024), and MACE (Lyu et al., 2024)—primarily operate by modifying the cross-attention layers within the U-Net architecture. In latent diffusion models (LDMs), these layers serve as the injection point for textual conditions derived from CLIP embeddings (Rombach et al., 2022). To erase harmful concepts, these methods adjust attention weights to suppress their influence. For example, TIME remaps undesirable concepts to irrelevant but benign ones, thereby reducing their attention scores. Forget-Me-Not builds on this by minimizing the L2 norm of attention maps associated with harmful concepts. UCE introduces a preservation loss and an auxiliary dataset to enhance retention of non-target concepts. MACE further balances generality and specificity by designing LoRA modules tailored to each concept and integrating them with TIME's closed-form solution. While attention-based methods are effective in minimizing performance degradation on non-erased concepts, recent studies (Beerens et al., 2025) have shown that simple re-training on specific datasets can lead to the recovery of erased knowledge,

revealing a critical vulnerability in attention-based erasure methods and highlighting the need for more resilient defenses.

Output-based methods—such as ESD (Gandikota et al., 2023), AP (Bui et al., 2024), and AGE (Bui et al., 2025)—erase undesirable concepts by fine-tuning diffusion models to minimize the predicted noise difference between harmful prompts and their safe counterparts. Unlike attention-based approaches, which operate on intermediate attention maps, output-based methods require access to latent representations across multiple diffusion steps, resulting in significantly higher computational overhead. The first output-based method, ESD, removes harmful concepts by optimizing the model in the direction opposite to the likelihood of generating those concepts. AP builds on this by introducing an adversarial prompt strategy that enhances the retention of related but benign features (e.g., preserving gender-specific traits while removing nudity). AGE further extends this line of work by proposing an adaptive erasure target selection mechanism and incorporating AP's retention strategy, achieving a better balance between erasure effectiveness and content preservation. Despite their effectiveness, the high computational cost of output-based methods remains a key limitation for large-scale deployment.

Beyond parameter-modifying approaches, SPM (Lyu et al., 2024) introduces a lightweight alternative by employing a one-dimensional adapter inspired by large model fine-tuning techniques. This adapter operates as an external module that guides the generation process to avoid harmful content, without altering the underlying model weights. However, due to its detachable nature, the adapter can be easily bypassed or removed by malicious users, rendering SPM less robust and secure than methods that directly modify model parameters.

## B.2 THREATS TO CONCEPT ERASURE METHODS

Prompt tuning, a technique for manipulating prompts to elicit specific behaviors from LLMs, has become a prominent topic in natural language processing (NLP). Recently, this technique has been exploited in jailbreak attacks (Yu et al., 2026), where carefully crafted prompts can bypass safety mechanisms and induce harmful outputs. Inspired by such attacks, researchers have begun exploring whether similar strategies can be used to recover concepts that were intentionally erased from DMs.

One such approach is Concept Inversion (CI) (Pham et al., 2024), which introduces a learnable token embedding to represent the erased concept. By minimizing the noise prediction distance between the erased concept and the new token, CI effectively restores the erased information (Han et al., 2024). Building on this idea, UnlearnDiffAtk (Zhang et al., 2024c) employs discrete token optimization via projection onto the probability simplex, enabling high-probability restoration of erased concepts through prompt manipulation. These findings highlight the vulnerability of current concept erasure methods to prompt-based attacks.

Other notable methods include Prompting4Debugging (P4D) (Chin et al., 2024), which optimizes prompts in the embedding space and projects them onto discrete tokens, similar to the PEZ framework (Wen et al., 2023). Meanwhile, Ring-A-Bell (Tsai et al., 2024) constructs an empirical representation of the erased concept by averaging embedding differences between prompts with and without the concept, and then uses a genetic algorithm to optimize prompts that can trigger the erased knowledge. These methods collectively demonstrate that even erased concepts can be reactivated through carefully tuned prompts, posing a significant challenge to the robustness of current erasure techniques.

# C PSEUDOCODES

## C.1 PSEUDOCODE OF AET

---

**Algorithm 1** ADVERSARIAL ERASURE TARGET (AET)

---

**Input:** Training Epoch $\tau$, AET from previous epoch $\mathbf{c}_K'^{(\tau-1)} \in \mathbb{R}^m$ (AET after $K$ steps in epoch $\tau-1$), Benign Concept $\mathbf{c}_e$, Original Noise Predictor $\boldsymbol{\epsilon}_{\boldsymbol{\theta}_0}$, Fine-Tuned Noise Predictor $\boldsymbol{\epsilon}_{\boldsymbol{\theta}}$, Learnable Segment Start Index $\mathcal{M}$, Learnable Segment Length $m'$, Iteration Steps $K$, Step Size $\beta$.

**Output:** Optimized AET for current epoch $\mathbf{c}_K'^{(\tau)}$.
    // AET Generation (§4.1)
1: Initialize $\mathbf{c}_0'^{(\tau)}$ (e.g., from a template or $\mathbf{c}_K'^{(\tau-1)}$).
2: $\mathbf{c}_0'^{(\tau)}[\mathcal{M} : \mathcal{M} + m'] \leftarrow \mathbf{c}_K'^{(\tau-1)}[\mathcal{M} : \mathcal{M} + m']$      ▷ *Initialize/update the learnable segment of AET*
3: **for** $j = 1$ **to** $K$ **do**
4:      $\mathcal{L}_{\text{AET}} \leftarrow \left\| \boldsymbol{\epsilon}_{\boldsymbol{\theta}_0}(\mathbf{z}_t|\mathbf{c}_{j-1}'^{(\tau)}) - \boldsymbol{\epsilon}_{\boldsymbol{\theta}}(\mathbf{z}_t|\mathbf{c}_e) \right\|_2^2 + \left\| \boldsymbol{\epsilon}_{\boldsymbol{\theta}_0}(\mathbf{z}_t|\mathbf{c}_{j-1}'^{(\tau)}) - \boldsymbol{\epsilon}_{\boldsymbol{\theta}_0}(\mathbf{z}_t|\mathbf{c}_e) \right\|_2^2$
5:      $\mathbf{c}_j'^{(\tau)} \leftarrow \mathbf{c}_{j-1}'^{(\tau)} - \beta \cdot \text{sign}(\nabla_{\mathbf{c}_{j-1}'^{(\tau)}} \mathcal{L}_{\text{AET}})$

---

## C.2 PSEUDOCODE OF AEGIS

---

**Algorithm 2** ADVERSARIAL ERASURE WITH GRADIENT-INFORMED SYNERGY (AEGIS)

---

**Input:** Erased Concept $\mathbf{c}_e$; Fine-tuned $\boldsymbol{\epsilon}_{\boldsymbol{\theta}}$ and Original $\boldsymbol{\epsilon}_{\boldsymbol{\theta}_0}$; Training Epochs $E$; Initial $\omega = 0$; Update step for $\omega, \mu$; Learning Rate $\alpha$; AET Step Size $\beta$; AET Iterative Steps $K$; Learnable Length $m'$.

**Output:** Fine-tuned parameters $\boldsymbol{\theta}$ for $\boldsymbol{\epsilon}_{\boldsymbol{\theta}}$.
    // AEGIS Procedure
1: **for** $\tau = 1$ **to** $E$ **do**
2:      $\mathbf{c}'^{(\tau)} \leftarrow \text{AET}(\tau, \mathbf{c}'^{(\tau-1)}, \mathbf{c}_e, \boldsymbol{\epsilon}_{\boldsymbol{\theta}_0}, \boldsymbol{\epsilon}_{\boldsymbol{\theta}}, m', K, \beta)$      ▷ *Generate AET*
3:      $\boldsymbol{\epsilon}_{\boldsymbol{\theta}_0}(\mathbf{z}_t|\tilde{\mathbf{c}}) \leftarrow \boldsymbol{\epsilon}_{\boldsymbol{\theta}_0}(\mathbf{z}_t) - \eta(\boldsymbol{\epsilon}_{\boldsymbol{\theta}_0}(\mathbf{z}_t|\mathbf{c}'^{(\tau)}) - \boldsymbol{\epsilon}_{\boldsymbol{\theta}_0}(\mathbf{z}_t))$      ▷ *Construct erasure target*
4:      $\mathbf{c}^{*(\tau)} \leftarrow \mathbf{c}^{*(\tau-1)} - \beta \cdot \nabla_{\mathbf{c}^{*(\tau-1)}} \left\| \boldsymbol{\epsilon}_{\boldsymbol{\theta}}(\mathbf{z}_t|\mathbf{c}^{*(\tau-1)}) - \boldsymbol{\epsilon}_{\boldsymbol{\theta}_0}(\mathbf{z}_t|\mathbf{c}_e) \right\|_2^2$    ▷ *Generate adversarial prompt*
5:      $\mathcal{L}_e \leftarrow \left\| \boldsymbol{\epsilon}_{\boldsymbol{\theta}}(\mathbf{z}_t|\mathbf{c}^*) - \boldsymbol{\epsilon}_{\boldsymbol{\theta}_0}(\mathbf{z}_t|\tilde{\mathbf{c}}) \right\|_2^2$      ▷ *Compute erasing Loss*
6:      $\mathcal{L}_r \leftarrow \frac{1}{2} \|\boldsymbol{\theta} - \boldsymbol{\theta}_0\|_2^2$      ▷ *Compute retention loss*
7:      $\boldsymbol{g}_e^{(\tau)} \leftarrow \nabla_{\boldsymbol{\theta}} \mathcal{L}_e, \boldsymbol{g}_r^{(\tau)} \leftarrow \nabla_{\boldsymbol{\theta}} \mathcal{L}_r$      ▷ *Compute gradients*
8:      **if** $\langle \boldsymbol{g}_e^{(\tau)}, \boldsymbol{g}_r^{(\tau)} \rangle \geq 0$ **then**      ▷ *Gradient non-conflict*
9:          $\hat{\boldsymbol{g}}^{(\tau)} \leftarrow \boldsymbol{g}_e^{(\tau)}$
10:      **else**      ▷ *Gradient conflict*
11:          $\tilde{\boldsymbol{g}}_r^{(\tau)} \leftarrow -\omega^{(\tau)} \frac{\langle \boldsymbol{g}_e^{(\tau)}, \boldsymbol{g}_r^{(\tau)} \rangle}{\|\boldsymbol{g}_r^{(\tau)}\|_2^2} \cdot \boldsymbol{g}_r^{(\tau)}$      ▷ *Rectify retention gradient (Eq. (8))*
12:          $\hat{\boldsymbol{g}}^{(\tau)} \leftarrow \boldsymbol{g}_e^{(\tau)} + \tilde{\boldsymbol{g}}_r^{(\tau)}$      ▷ *Update using DGR (Eq. (9))*
13:      $\nabla \omega^{(\tau)} \leftarrow \langle \boldsymbol{g}_e^{(\tau)}, \frac{\langle \boldsymbol{g}_e^{(\tau-1)}, \boldsymbol{g}_r^{(\tau-1)} \rangle}{\|\boldsymbol{g}_r^{(\tau-1)}\|_2^2} \boldsymbol{g}_r^{(\tau-1)} \rangle$      ▷ *Compute gradient for $\omega$ (Eq. (11))*
14:      $\omega^{(\tau+1)} \leftarrow \min\left(1, \max\left(0, \omega^{(\tau)} - \mu \cdot \text{sign}(\nabla \omega^{(\tau)})\right)\right)$      ▷ *Update $\omega$ (Eq. (11)), clamped in $[0, 1]$*
15:      $\boldsymbol{\theta}^{(\tau)} \leftarrow \boldsymbol{\theta}^{(\tau-1)} - \alpha \cdot \hat{\boldsymbol{g}}^{(\tau)}$      ▷ *Update model parameters*

---

# D FURTHER CLARIFICATION OF AEGIS

## D.1 VULNERABILITY OF CURRENT ERASURE METHODS

Following the observations of Fig. 9a, for the erased concept $\mathbf{c}_e$ (e.g., "nudity"), ESD maps $\mathbf{c}_e$ to a target $\tilde{\mathbf{c}}$ in order to remove the knowledge associated with concept cluster $\mathcal{C}_0$. This is equivalent to maximizing the distance between the model predictions $\boldsymbol{\epsilon}_{\boldsymbol{\theta}}(\mathbf{z}_t|\mathbf{c}_e)$ and $\boldsymbol{\epsilon}_{\boldsymbol{\theta}_0}(\mathbf{z}_t|\mathbf{c})$. However, if the instance $\mathbf{c}_e$ selected from cluster $\mathcal{C}_1$ lies far from the concept center, as shown in Fig. 9a, the mapping may cross decision boundaries and associate with irrelevant concepts in $\mathcal{C}_2$. As a result, the knowledge of $\mathcal{C}_2$ cannot be effectively removed, leading to reduced robustness in concept erasure.

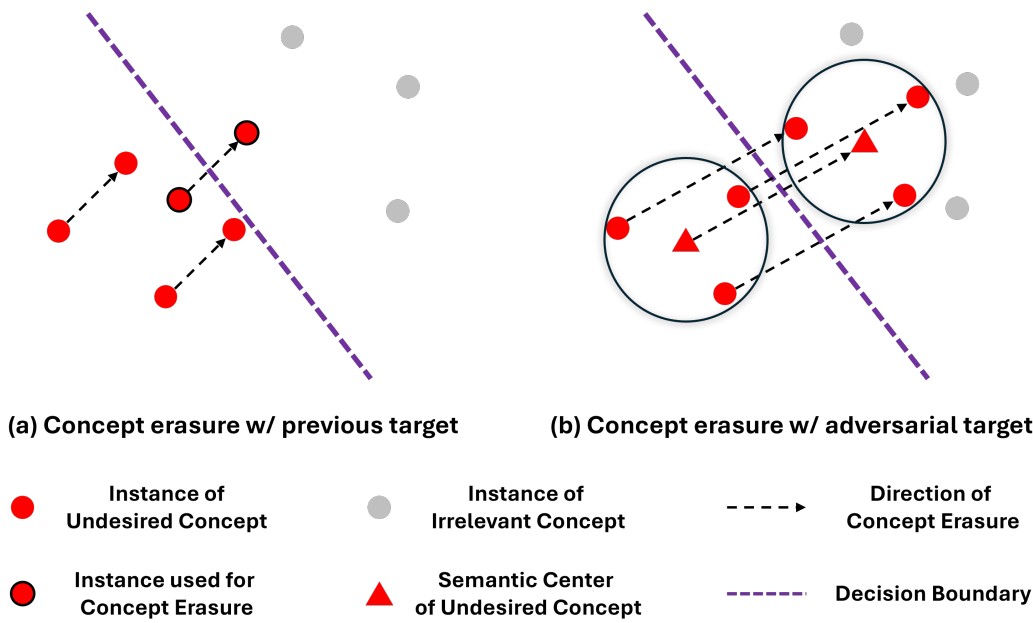

**(a) Concept erasure w/ previous target**     **(b) Concept erasure w/ adversarial target**

🔴 **Instance of Undesired Concept**     🔘 **Instance of Irrelevant Concept**     - - - → **Direction of Concept Erasure**

🔴 **Instance used for Concept Erasure**     🔺 **Semantic Center of Undesired Concept**     ----- **Decision Boundary**

Figure 9: Illustration of concept erasure with the previous target as ESD and concept center (adversarial target). The concept erasure with the previous target still preserve some information of the undesirable concept, while using the concept center can significantly reduce the erased concept information from the DM.

Unlike the conventional approach that maps the erased concept $c_e$ to its opposite $\tilde{c}$, we propose an Adversarial Erasure Target (AET). Motivated by the observation that adversarial prompts for benign concept instances tend to be close to the concept center, we generate the AET $c'$ for $c_e$ using Eq. (5). The target $c'$ is intentionally chosen to be distant from the concept center of $\mathcal{C}_0$ in both the fine-tuned model $\theta$ and the original model $\theta_0$. As illustrated in Fig. 9b, by maximizing the distance between $\epsilon_\theta(z_t|c_e)$ and $\epsilon_{\theta_0}(z_t|c')$, the representation of $c_e$ is pushed away from the original concept space. This allows the model to effectively erase the knowledge associated with instances in $\mathcal{C}_2$. Furthermore, when the adversarial prompt $c^*$—which corresponds to the concept center of $\mathcal{C}_0$ in $\theta_0$—is incorporated into the erasure process, the knowledge of $\mathcal{C}_0$ is significantly suppressed, resulting in a lower attack success rate (ASR) under UnlearnDiffAtk.

### D.2 GRADIENT CONFLICT DURING CONCEPT ERASURE

To illustrate the gradient conflict between concept erasure and retention objectives, we compute the cosine similarity between their respective gradients across training epochs. Specifically, we calculate the cosine similarity for each layer and then average the values across all layers at each training step.

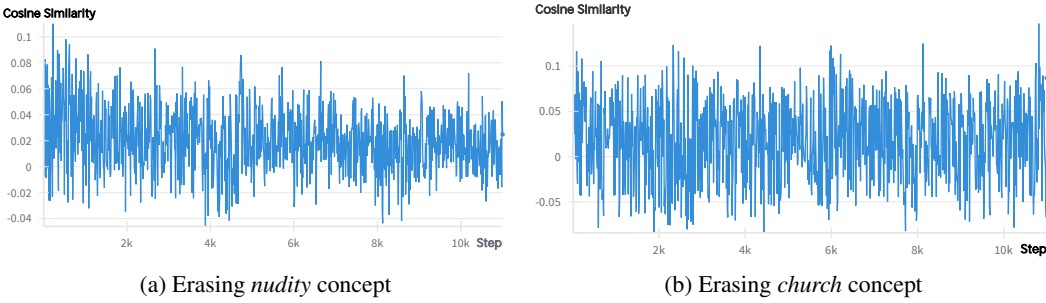

(a) Erasing *nudity* concept     (b) Erasing *church* concept

Figure 10: Cosine similarity between gradients induced by the erasing and retention losses across different training epochs.

As shown in Fig. 10, some training steps exhibit negative cosine similarities. According to Def. 2, a negative cosine similarity indicates that the gradients from the concept erasure and retention losses are in conflict.

# E    MISSING THEORETICAL ANALYSES AND PROOFS

The following results are stated under idealized assumptions (convexity, smoothness, abstract curvature) and are intended as upper bounds and design guidance rather than as literal descriptions of diffusion fine-tuning, which is inherently nonconvex. Their practical value here is twofold: (i) direction safety — they certify that our rectified update cannot reverse the erasing descent direction when the stated conditions hold, and (ii) schedule design — they motivate the simple rules we adopt in practice (warm-up on $\omega$, conflict-only projection, and block-aware scaling) that reproduce the "do-no-harm" geometry in nonconvex regimes. Although the guarantees are derived under convex/curvature assumptions, the mechanism they justify—conflict-only projection with a bounded $\omega$ and no adjustment when gradients are aligned—directly targets two failure modes that dominate nonconvex fine-tuning: (a) direction reversal of the main task gradient and (b) oscillation due to large destructive interference. By construction, our update never amplifies the component of $g_r$ that opposes $g_e$, and leaves $g_e$ intact when there is no conflict. This "safety-first" geometry is precisely what stabilizes training in practice: fewer harmful steps, smaller loss spikes, and smoother progress of the erasing objective, even without convexity. We therefore treat Thms. 4.1 and 4.2 as principled guardrails without claiming global optimality on nonconvex landscapes.

## E.1    PROOF OF PROPOSITION 3.1

**Proposition 3.1** (Deviation Lower Bound). *Let $\mathcal{L}_{erase}(\boldsymbol{\theta}) = \mathbb{E}_t\big[\|\boldsymbol{\epsilon}_{\boldsymbol{\theta}}(\mathbf{z}_t|\mathbf{c}_e^0) - \boldsymbol{\epsilon}_{\boldsymbol{\theta}_0}(\mathbf{z}_t|\tilde{\mathbf{c}})\|_2^2\big]$, and $\Delta_T = \|\boldsymbol{\epsilon}_{\boldsymbol{\theta}_0}(\mathbf{z}_T|\mathbf{c}_e^0) - \boldsymbol{\epsilon}_{\boldsymbol{\theta}_0}(\mathbf{z}_T|\tilde{\mathbf{c}})\|_2^2$. Assume optimizing Eq. (2) yields $\mathcal{L}_{erase}(\boldsymbol{\theta}) \leq \delta < \Delta_T$. Then,*

$$\big\|\boldsymbol{\epsilon}_{\boldsymbol{\theta}}(\mathbf{z}_T|\mathbf{c}_e^0) - \boldsymbol{\epsilon}_{\boldsymbol{\theta}_0}(\mathbf{z}_T|\mathbf{c}_e^0)\big\|_2^2 \geq \big(\sqrt{\Delta_T} - \sqrt{\delta}\big)^2. \tag{4}$$

*Proof.* Let $a = \boldsymbol{\epsilon}_{\boldsymbol{\theta}}(\mathbf{z}_t|\mathbf{c}_e^0)$, $b = \boldsymbol{\epsilon}_{\boldsymbol{\theta}_0}(\mathbf{z}_t|\tilde{\mathbf{c}})$, and $c = \boldsymbol{\epsilon}_{\boldsymbol{\theta}_0}(\mathbf{z}_t|\mathbf{c}_e^0)$. By the reverse triangle inequality,

$$\|a - c\|_2 \geq \|b - c\|_2 - \|a - b\|_2.$$

Squaring both sides and applying the inequality $(x - y)^2 \leq x^2 - 2xy + y^2$, we obtain

$$\|a - c\|_2^2 \geq \|b - c\|_2^2 - 2\|b - c\|_2 \cdot \|a - b\|_2.$$

Taking expectation with respect to $t$ and applying Cauchy–Schwarz to the cross term yields

$$\mathbb{E}_t\|a - c\|_2^2 \geq \mathbb{E}_t\|b - c\|_2^2 - 2\sqrt{\mathbb{E}_t\|b - c\|_2^2} \cdot \sqrt{\mathbb{E}_t\|a - b\|_2^2},$$

which simplifies to

$$\mathbb{E}_t\big\|\boldsymbol{\epsilon}_{\boldsymbol{\theta}}(\mathbf{z}_t|\mathbf{c}_e^0) - \boldsymbol{\epsilon}_{\boldsymbol{\theta}_0}(\mathbf{z}_t|\mathbf{c}_e^0)\big\|_2^2 \geq \big(\sqrt{\Delta} - \sqrt{\mathcal{L}_{\text{erase}}(\boldsymbol{\theta})}\big)^2,$$

where $\Delta = \mathbb{E}_t\|\boldsymbol{\epsilon}_{\boldsymbol{\theta}_0}(\mathbf{z}_t|\mathbf{c}_e^0) - \boldsymbol{\epsilon}_{\boldsymbol{\theta}_0}(\mathbf{z}_t|\tilde{\mathbf{c}})\|_2^2$.

In particular, the above inequality holds for any fixed diffusion step $t$, including the final denoising step $t = T$. Therefore, we obtain

$$\big\|\boldsymbol{\epsilon}_{\boldsymbol{\theta}}(\mathbf{z}_T|\mathbf{c}_e^0) - \boldsymbol{\epsilon}_{\boldsymbol{\theta}_0}(\mathbf{z}_T|\mathbf{c}_e^0)\big\|_2 \geq \sqrt{\|\boldsymbol{\epsilon}_{\boldsymbol{\theta}_0}(\mathbf{z}_T|\mathbf{c}_e^0) - \boldsymbol{\epsilon}_{\boldsymbol{\theta}_0}(\mathbf{z}_T|\tilde{\mathbf{c}})\|_2^2} - \sqrt{\mathcal{L}_{\text{erase}}(\boldsymbol{\theta})} \geq \sqrt{\Delta_T} - \sqrt{\delta}.$$

This concludes the proof. $\qquad\square$

## E.2    PROOF OF THEOREM 4.1

**Theorem 4.1** (Local Descent Guarantee of Erasing Loss). *Suppose $\mathcal{L}_e$ is continuously differentiable, convex, and locally $L$-smooth. Then, for a step size $\alpha \leq 2/L$, $\boldsymbol{\theta}_{\text{GRP}}$ updated via GRP either converges to (a) a degenerate point where $\cos\phi = -1$, or (b) a local minimum $\boldsymbol{\theta}^*$ of $\mathcal{L}_e$.*

*Proof.* At the $\tau$-th training epoch, by the second-order Taylor expansion and $L$-smoothness of $\mathcal{L}_e$, we have:

$$\mathcal{L}_e(\boldsymbol{\theta}^{(\tau)} - \alpha\,\hat{\boldsymbol{g}}^{(\tau)}) \leq \mathcal{L}_e(\boldsymbol{\theta}^{(\tau)}) - \langle \nabla\mathcal{L}_e, \alpha\hat{\boldsymbol{g}}^{(\tau)}\rangle + \frac{L\,\alpha^2}{2}\|\hat{\boldsymbol{g}}^{(\tau)}\|_2^2$$

$$= \mathcal{L}_e(\boldsymbol{\theta}^{(\tau)}) - \langle \boldsymbol{g}_e^{(\tau)}, \alpha\hat{\boldsymbol{g}}^{(\tau)}\rangle + \frac{L\,\alpha^2}{2}\|\hat{\boldsymbol{g}}^{(\tau)}\|_2^2.$$

We further define

$$\Delta^{(\tau)} := -\alpha\langle \boldsymbol{g}_e^{(\tau)}, \hat{\boldsymbol{g}}^{(\tau)}\rangle + \frac{L\,\alpha^2}{2}\|\hat{\boldsymbol{g}}^{(\tau)}\|_2^2,$$

Then, if $\Delta^{(\tau)} < 0$, we obtain a strict decrease in the erasing loss function, i.e., $\mathcal{L}_e(\boldsymbol{\theta}^{(\tau+1)}) < \mathcal{L}_e(\boldsymbol{\theta}^{(\tau)})$, thereby completing the proof. As observed, the term $\Delta^{(\tau)}$ consists of two components:

- **Linear term:** Since $\hat{\boldsymbol{g}}^{(\tau)}$ is obtained by projecting $\boldsymbol{g}_e^{(\tau)}$ via our proposed gradient regularization projection (GRP) defined in Eq. (9), we have $\langle \boldsymbol{g}_e^{(\tau)}, \hat{\boldsymbol{g}}^{(\tau)}\rangle \geq 0$.

- **Quadratic term:** Given that $\|\hat{\boldsymbol{g}}^{(\tau)}\|_2 \leq \|\boldsymbol{g}_e^{(\tau)}\|_2$, it follows that $\frac{L\alpha^2}{2}\|\hat{\boldsymbol{g}}^{(\tau)}\|_2^2 \leq \frac{L\alpha^2}{2}\|\boldsymbol{g}_e^{(\tau)}\|_2^2$.

Then, we can find that the sign of $\Delta^{(\tau)}$ is determined by $-\alpha + \frac{L\alpha^2}{2}$, where we need to ensure

$$-\alpha + L\alpha^2/2 < 0 \quad \Longrightarrow \quad \alpha < 2/L. \tag{12}$$

Under the above condition, the negative linear term dominates the quadratic penalty term, so we have $\Delta^{(\tau)} < 0$ and

$$\mathcal{L}_e(\boldsymbol{\theta}^{(\tau+1)}) \; < \; \mathcal{L}_e(\boldsymbol{\theta}^{(\tau)}). \tag{13}$$

Thus, we obtain a strict descent in the concept erasure procedure using `AEGIS`, except in degenerate cases. Specifically, a degenerate scenario arises when the cosine similarity between the erasure and retention gradients satisfies $\cos\phi^{(\tau)} = -1$, i.e., $\phi^{(\tau)} = 180°$. In this case, the gradient $\boldsymbol{g}_e^{(\tau)}$ induced by the concept erasure loss $\mathcal{L}_e$ is exactly opposite to the retention gradient $\boldsymbol{g}_r^{(\tau)}$ from $\mathcal{L}_r$.

Under this condition, the gradient projection defined in Eq. (9) at the $\tau$-th epoch can be rewritten as:

$$\hat{\boldsymbol{g}}^{(\tau)} = \boldsymbol{g}_e^{(\tau)} - \omega^{(\tau)}\frac{\|\boldsymbol{g}_e^{(\tau)}\|_2\cos\phi^{(\tau)}}{\|\boldsymbol{g}_r^{(\tau)}\|_2}\boldsymbol{g}_r^{(\tau)}.$$

When $\cos\phi^{(\tau)} = -1$, the projected gradient becomes $\hat{\boldsymbol{g}}^{(\tau)} = (1 - \omega^{(\tau)})\boldsymbol{g}_e^{(\tau)}$. As the weight $\omega^{(\tau)}$ gradually increases to 1, the influence of the erasure gradient is fully suppressed. Consequently, after applying the DGR (Directional Gradient Rectification) mechanism within the GRP, we obtain $\hat{\boldsymbol{g}}^{(\tau)} = \boldsymbol{0}$.

In this case, the model parameters remain unchanged, i.e.,

$$\boldsymbol{\theta}^{(\tau+1)} = \boldsymbol{\theta}^{(\tau)}, \tag{14}$$

indicating that no further updates are applied to the diffusion model. Therefore, the proof is complete.
$\square$

### E.3 PROOF OF THEOREM 4.2

**Theorem 4.2** (Retention Benefit of GRP). *Assume $\mathcal{L}_r$ is differentiable and $L$-smooth, and its $\alpha$-curvature of $\mathcal{L}_r$ (cf. Def. 3 in Appx. E.3) satisfies $\mathcal{H}_\alpha(\mathcal{L}_r; \boldsymbol{g}_e) \geq \ell\|\boldsymbol{g}_e\|_2^2$ for some $\ell < L$. At step $\tau$, denote the GRP-rectified parameter by $\boldsymbol{\theta}_{\text{GRP}}^{(\tau)}$ and the unrectified by $\boldsymbol{\theta}^{(\tau)}$. Then, $\mathcal{L}_r(\boldsymbol{\theta}_{\text{GRP}}^{(\tau)}) \leq \mathcal{L}_r(\boldsymbol{\theta}^{(\tau)})$ holds if conditions (1) $\ell \geq L\frac{\|\boldsymbol{g}_e^{(\tau)} - \boldsymbol{g}_r^{(\tau)}\|_2^2}{\|\boldsymbol{g}_e^{(\tau)} + \boldsymbol{g}_r^{(\tau)}\|_2^2}$, and (2) $\ell \geq L\frac{\|\boldsymbol{g}_e^{(\tau)} - \boldsymbol{g}_r^{(\tau)}\|_2^2}{\|\boldsymbol{g}_e^{(\tau)} + \boldsymbol{g}_r^{(\tau)}\|_2^2} + \frac{2}{\alpha}$ satisfied.*

Our proof is inspired by Yu et al. (2020); Wang et al. (2025). We first introduce the definition of $q$-curvature, which characterizes the second-order behavior of the loss function along the gradient direction.

**Definition 3** (*q*-curvature). *For any smooth and differentiable loss $\mathcal{L}$, the q-curvature $\mathcal{H}_q$ w.r.t. $g$ is defined as*

$$\mathcal{H}_q(\mathcal{L}; \boldsymbol{g}) = \int_0^1 (1 - \xi)\big[\boldsymbol{g}^\top \cdot \nabla^2 \mathcal{L}(\boldsymbol{\theta} - \xi q \boldsymbol{g})\nabla \mathcal{L}\big] \, d\xi. \tag{15}$$

Intuitively, $\mathcal{H}_q$ captures the curvature of $\mathcal{L}$ along the gradient direction within a segment of length $q$, weighted by a linear kernel $(1 - \xi)$. This quantity will be used to bound the second-order term in our local descent analysis.

*Proof.* Recall that at the $\tau$-th iteration, the original updating rule without DGR is $\boldsymbol{\theta}^{(\tau+1)} = \boldsymbol{\theta}^{(\tau)} - \alpha \boldsymbol{g}^{(\tau)}$. Additionally, according to the integral form of Taylor's theorem, for any $a \in [0, 1]$, we can obtain

$$\mathcal{L}_r(\boldsymbol{\theta}^{(\tau)} - \alpha \boldsymbol{g}_e^{(\tau)}) = \mathcal{L}_r(\boldsymbol{\theta}^{(\tau)}) + \int_0^1 \nabla \mathcal{L}_r(\boldsymbol{\theta}^{(\tau)} - a\,\alpha\,\boldsymbol{g}_e^{(\tau)})^\top [-\alpha \boldsymbol{g}_e^{(\tau)}] \, da.$$

Separating the first-order (linear) portion and the second-order (Hessian) portion, one can write:

$$\mathcal{L}_r(\boldsymbol{\theta}^{(\tau+1)}) = \mathcal{L}_r(\boldsymbol{\theta}^{(\tau)}) - \alpha \, \langle \boldsymbol{g}_r^{(\tau)}, \boldsymbol{g}_e^{(\tau)} \rangle + \frac{1}{2} \int_0^1 [-\alpha \boldsymbol{g}_e^{(\tau)}]^\top \nabla^2 \mathcal{L}_r \Big( \boldsymbol{\theta}^{(\tau)} - a\,\alpha\,\boldsymbol{g}_e^{(\tau)} \Big) [-\alpha \boldsymbol{g}_e^{(\tau)}] \, da.$$

Since we assume $\mathcal{H}_\alpha(\mathcal{L}_r; \boldsymbol{g}_e^{(\tau)}) \geq \ell \|\boldsymbol{g}_e^{(\tau)}\|_2^2$, we can obtain

$$\int_0^1 [-\alpha \boldsymbol{g}_e^{(\tau)}]^\top \nabla^2 \mathcal{L}_r(\boldsymbol{\theta}^{(\tau)} - a\,\alpha\,\boldsymbol{g}_e^{(\tau)})[-\alpha \boldsymbol{g}_e^{(\tau)}] \, da \geq \ell \alpha^2 \|\boldsymbol{g}_e^{(\tau)}\|_2^2, \tag{16}$$

and then

$$\mathcal{L}_r(\boldsymbol{\theta}^{(\tau)} - \alpha \boldsymbol{g}_e^{(\tau)}) \geq \mathcal{L}_r(\boldsymbol{\theta}^{(\tau)}) - \alpha \, \langle \boldsymbol{g}_r^{(\tau)}, \boldsymbol{g}_e^{(\tau)} \rangle + \frac{\ell \alpha^2}{2} \|\boldsymbol{g}_e^{(\tau)}\|_2^2, \tag{17}$$

which establishes the lower bound for $\mathcal{L}_r(\boldsymbol{\theta}^{(\tau+1)}) = \mathcal{L}_r(\boldsymbol{\theta}^{(\tau)} - \alpha \boldsymbol{g}_e^{(t)})$. For the rectified updating rule with GRP, due to the $L$-smoothness, we have

$$\mathcal{L}_r(\boldsymbol{\theta}_{\mathrm{GRP}}^{(\tau+1)}) \leq \mathcal{L}_r(\boldsymbol{\theta}^{(\tau)}) - \alpha \, \langle \boldsymbol{g}_r^{(\tau)}, \hat{\boldsymbol{g}}^{(\tau)} \rangle + \frac{L \alpha^2}{2} \|\hat{\boldsymbol{g}}^{(\tau)}\|_2^2. \tag{18}$$

Combining Eqs. (17) and (18), we have

$$\Delta = \mathcal{L}_r(\boldsymbol{\theta}^{(\tau)}) - \mathcal{L}_r(\boldsymbol{\theta}_{\mathrm{GRP}}^{(\tau+1)})$$

$$\geq \underbrace{\Big[\mathcal{L}_r(\boldsymbol{\theta}^{(\tau)}) - \alpha \, \langle \boldsymbol{g}_r^{(\tau)}, \hat{\boldsymbol{g}}^{(\tau)} \rangle + \frac{\ell \alpha^2}{2} \|\boldsymbol{g}_e^{(\tau)}\|_2^2 \Big]}_{\text{Lower bound for } \mathcal{L}_r(\boldsymbol{\theta}^{(\tau+1)})} - \underbrace{\Big[\mathcal{L}_r(\boldsymbol{\theta}^{(\tau)}) - \alpha \, \langle \boldsymbol{g}_r^{(t)}, \hat{\boldsymbol{g}}^{(\tau)} \rangle + \frac{L \alpha^2}{2} \|\hat{\boldsymbol{g}}^{(\tau)}\|_2^2 \Big]}_{\text{Upper bound for } \mathcal{L}_r(\boldsymbol{\theta}_{\mathrm{GRP}}^{(t+1)})}.$$

After organizing, we have

$$\Delta \geq \Big[ -\alpha \, \langle \boldsymbol{g}_r^{(\tau)}, \boldsymbol{g}_e^{(\tau)} \rangle + \tfrac{\ell \alpha^2}{2} \|\boldsymbol{g}_e^{(\tau)}\|_2^2 \Big] - \Big[ -\alpha \, \langle \boldsymbol{g}_r^{(\tau)}, \hat{\boldsymbol{g}}^{(\tau)} \rangle + \tfrac{L \alpha^2}{2} \|\hat{\boldsymbol{g}}^{(\tau)}\|_2^2 \Big]$$

$$= \underbrace{-\alpha \, \langle \boldsymbol{g}_r^{(\tau)}, \boldsymbol{g}_e^{(\tau)} \rangle + \alpha \, \langle \boldsymbol{g}_r^{(\tau)}, \hat{\boldsymbol{g}}^{(\tau)} \rangle}_{\text{(linear-difference term)}} + \underbrace{\frac{\ell \alpha^2}{2} \|\boldsymbol{g}_e^{(\tau)}\|_2^2 - \frac{L \alpha^2}{2} \|\hat{\boldsymbol{g}}^{(\tau)}\|_2^2}_{\text{(quadratic-difference term)}}.$$

Now, we show that the formulations inside each bracket term is non-negative:

1. **Rectification Nonnegativity.** Since $\hat{\boldsymbol{g}}^{(\tau)}$ is obtained from $\boldsymbol{g}_e^{(\tau)}$ by removing components that are negatively aligned with $\boldsymbol{g}_r^{(\tau)}$, we have:

$$\langle \boldsymbol{g}_r^{(\tau)}, \hat{\boldsymbol{g}}^{(\tau)} \rangle \geq \langle \boldsymbol{g}_r^{(\tau)}, \boldsymbol{g}_e^{(\tau)} \rangle, \tag{19}$$

and thus

$$\langle \boldsymbol{g}_r^{(\tau)}, \boldsymbol{g}_e^{(\tau)} \rangle + \langle \boldsymbol{g}_r^{(\tau)}, \hat{\boldsymbol{g}}^{(\tau)} \rangle \geq 0. \tag{20}$$

Multiplying by $\alpha > 0$ preserves non-negativity, ensuring that the first bracketed term is non-negative.

2. **Curvature Conditions.** Under conditions **1)** and **2)**, we can obtain:

$$\ell \geq \frac{\|\boldsymbol{g}_e^{(\tau)} - \boldsymbol{g}_r^{(\tau)}\|_2^2}{\|\boldsymbol{g}_e^{(\tau)} + \boldsymbol{g}_r^{(\tau)}\|_2^2} L, \tag{21}$$

and

$$\ell \geq \frac{\|\boldsymbol{g}_e^{(\tau)} - \boldsymbol{g}_r^{(\tau)}\|_2^2}{\|\boldsymbol{g}_e^{(\tau)} + \boldsymbol{g}_r^{(\tau)}\|_2^2} L + \frac{2}{\alpha}. \tag{22}$$

These inequalities further imply that

$$\frac{\ell \alpha^2}{2} \|\boldsymbol{g}_e^{(\tau)}\|_2^2 - \frac{L \alpha^2}{2} \|\hat{\boldsymbol{g}}^{(\tau)}\|_2^2 \geq 0, \tag{23}$$

confirming that the second bracketed term is also non-negative.

Hence, both terms are non-negative, and the proof is complete. $\qquad\square$

### E.4 Practical Validity under Non-Convex UNet Fine-Tuning

**Scope.** Thms. 4.1 and 4.2 are stated under convexity and $L$-smoothness to provide clean, interpretable guarantees on the update geometry induced by GRP. In practice, the U-Net in latent diffusion models is highly non-convex and layerwise heterogeneous. This section clarifies why the results remain informative in such regimes, and how their conclusions can be read as stability and direction-safety properties that do not require global convexity.

#### E.4.1 From Global Convexity to Local Regularity and Quasi-Convexity

**Local $L$-smoothness instead of global convexity.** The descent and projection arguments only use: (i) a local quadratic upper bound on $\mathcal{L}_e$ (the standard smoothness inequality), and (ii) that GRP does not reverse the component of $\boldsymbol{g}_e$ along the update (*i.e.*, $\langle \boldsymbol{g}_e, \hat{\boldsymbol{g}} \rangle \geq 0$). Both conditions hold under local $L$-smoothness in a neighborhood of the iterate, which is a much weaker requirement than global convexity. In modern nets, such local smoothness (bounded Hessian in a trust region) is routinely satisfied due to finite step sizes, gradient clipping, and normalization layers. Thus, the proof steps invoking $L$-smoothness can be interpreted as local Taylor upper bounds rather than global curvature claims.

**Star-/quasi-convex landscapes along the descent direction.** Even when $\mathcal{L}_e$ is non-convex in the parameter space, it can be quasi-convex or star-convex along the actual update rays produced by GRP. The key inequality used in Thm. 4.1 is one-dimensional along $-\hat{\boldsymbol{g}}$ : $\mathcal{L}_e(\boldsymbol{\theta} - \alpha\hat{\boldsymbol{g}}) \leq \mathcal{L}_e(\boldsymbol{\theta}) - \alpha\langle \boldsymbol{g}_e, \hat{\boldsymbol{g}} \rangle + \frac{\mathcal{L}\alpha^2}{2}\|\hat{\boldsymbol{g}}\|_2^2$. For step sizes $\alpha$ in the usual fine-tuning range, this establishes a local descent direction whenever $\langle \boldsymbol{g}_e, \hat{\boldsymbol{g}} \rangle \geq 0$, independent of global convexity.

**Stability vs. optimality.** Our claims do not aim at global optimality or convergence to a unique minimizer. Instead, they formalize "direction safety": GRP cannot create an ascent step solely due to the retention term when gradients conflict. This is precisely the behaviour we need to avoid "reverse harm" to the erasing objective under multi-objective interference.

#### E.4.2 What the Projection Guarantees without Convexity

**Conflict-only modification.** GRP leaves $\boldsymbol{g}_e$ untouched whenever $\cos(\boldsymbol{g}_e, \boldsymbol{g}_r) \geq 0$, i.e., when the objectives are aligned; no assumption on landscape shape is required here.

**Non-reversal of the main task direction.** In conflict, DGR projects the retention gradient onto the orthogonal complement of $\boldsymbol{g}_e$ (scaled by $\omega$), yielding $\hat{\boldsymbol{g}} = \boldsymbol{g}_e + \tilde{\boldsymbol{g}}_r$ with $\langle \boldsymbol{g}_e, \hat{\boldsymbol{g}} \rangle = \|\boldsymbol{g}_e\|_2^2 \geq 0$. This algebraic identity holds regardless of convexity and ensures the step has a non-negative alignment with the descent direction of $\mathcal{L}_e$. Hence, even on non-convex surfaces, GRP does not turn the update into an ascent due to $\boldsymbol{g}_r$ alone.

**Bounded interference instead of descent rates.** Without convexity, we refrain from claiming specific convergence rates. The projection still provides a *bounded-interference* property: the component of $\boldsymbol{g}_r$ opposing $\boldsymbol{g}_e$ is removed, so the worst-case increase in $\mathcal{L}_e$ due to the retention term is zero per-step in the first-order sense. This is the exact "guardrail" we need in non-convex settings.

### E.4.3 READING THEOREMS 4.1 AND 4.2 AS DESIGN LEMMAS

**Theorem 4.1 (Descent under local smoothness).** Replace global convexity by local $L$-smoothness around $\theta$ and a step-size condition $\alpha \leq 2/\mathcal{L}$ (any local $\mathcal{L}$) within the trust region actually visited by training. Under these weaker assumptions, the same inequality shows that the quadratic penalty cannot dominate the linear decrease when $\alpha$ is small, thereby ensuring local descent or stalled updates only in the degenerate anti-parallel case ($\cos\phi = -1$). We use this result as a *design lemma* to select $\alpha$ and to motivate conflict-only projection; it is not a global convergence claim.

**Theorem 4.2 (Retention benefit as curvature-aware regularization).** The curvature term $H_\alpha(\mathcal{L}_r; \boldsymbol{g}_e)$ is introduced to quantify the second-order sensitivity of $\mathcal{L}_r$ along the erasing direction. In non-convex regimes, $H_\alpha$ should be interpreted as a *local generalized curvature* (the integral form in Def. 3) rather than a global bound. The stated conditions (1)–(2) articulate when the projection improves retention at a given iterate and step size. Practically, they guide two knobs: (i) avoid too small $\alpha$ that nullifies the retention effect; (ii) adjust $\omega$ so that the effective constraint tightens when $\boldsymbol{g}_e$ and $\boldsymbol{g}_r$ conflict strongly (large $\|\boldsymbol{g}_e - \boldsymbol{g}_r\|/\|\boldsymbol{g}_e + \boldsymbol{g}_r\|$). Again, these are scheduling rules, not asymptotic optimality claims.

### E.4.4 RELATION TO MULTI-OBJECTIVE AND GRADIENT-SURGERY LITERATURE

Our GRP is structurally similar to conflict-handling methods in multi-task learning (e.g., gradient surgery/projection). Those methods are justified by geometric arguments that hold in non-convex nets because they operate at the level of instantaneous gradients, not the global landscape. We adopt the same philosophy: protect the main task direction by removing the antagonistic component of the auxiliary gradient, and gate the projection by a dynamic $\omega$ to avoid over-constraining early steps.

**Why parameter-proximity retention instead of data?** Using $\mathcal{L}_r = \frac{1}{2}\|\boldsymbol{\theta} - \boldsymbol{\theta}_0\|_2^2$ ensures $\boldsymbol{g}_r$ is globally Lipschitz and uniformly well-defined, providing a stable, model-agnostic constraint even in non-convex regimes. This choice is precisely to avoid injecting noisy, concept-related signals that could reintroduce non-convex interference.

### E.4.5 PRACTICAL IMPLICATIONS FOR HYPERPARAMETERS

**Step size $\alpha$.** The "$\alpha \leq 2/\mathcal{L}$" condition should be read as "pick $\alpha$ in the local trust region where a second-order upper bound is valid." In practice we use small $\alpha$ ($10^{-5}$), which empirically keeps the update within this region.

**Projection weight $\omega$ (schedule).** A warm-up ($\omega \approx 0$ initially) prevents early over-regularization when $\boldsymbol{g}_e$ is still large and rapidly changing; increasing $\omega$ later tightens the retention constraint once the iterate enters a smoother basin. This schedule directly follows from the local-descent reading of Thm. 4.1 and the curvature-aware reading of Thm. 4.2.

**No claim of global optimality.** Our guarantees are "do-no-harm" in the first-order sense and local stability under realistic step sizes. They complement, rather than replace, empirical evaluation.

**Takeaway.** Although the U-Net fine-tuning landscape is non-convex, the utility of Thms. 4.1 and 4.2 lies in codifying two update-level safeguards that are agnostic to global convexity: (i) conflict-only projection that never reverses the erasing descent direction, and (ii) curvature-aware scheduling that bounds interference from the retention term. Interpreted as local design lemmas under small steps and local smoothness, these results justify GRP as a principled, geometry-safe controller in practice.

## F EXPERIMENTAL SETUP

### F.1 CONCEPT ERASURE SETTINGS

Following prior works (Kumari et al., 2023; Gandikota et al., 2023; Zhang et al., 2024c; Gandikota et al., 2024; Fan et al., 2024; Wu & Harandi, 2024; Wu et al., 2024; Zhang et al., 2024a), we evaluate the effectiveness of `AEGIS` across three main concept erasure tasks: (1) Nudity Concept Erasure: This task focuses on removing harmful content associated with nudity-related prompts from DMs (Kumari et al., 2023; Gandikota et al., 2023; Zhang et al., 2024c; Gandikota et al., 2024; Fan et al., 2024; Wu & Harandi, 2024; Wu et al., 2024). The evaluation dataset is derived from the Inappropriate

Image Prompt (I2P) dataset. (2) Style Concept Erasure: This task aims to remove the stylistic features of specific artists to mitigate copyright infringement risks in generated content (Kumari et al., 2023; Gandikota et al., 2023; Zhang et al., 2024a; Gandikota et al., 2024; Lyu et al., 2024). The evaluation setup follows the protocol described in (Gandikota et al., 2023). (3) Object Concept Erasure: Similar to the above tasks, this setting targets the removal of knowledge related to specific object categories (Gandikota et al., 2023; Zhang et al., 2024a; Fan et al., 2024; Lyu et al., 2024; Wu & Harandi, 2024; Wu et al., 2024). We use GPT-4 to generate 50 diverse prompts for each object class in the Imagenette dataset, ensuring that the standard SD model can generate images containing the corresponding objects. Unless otherwise specified, all experiments are conducted using the pre-trained SD v1.4 model as the base DM for concept erasure.

## F.2 BASELINE SETTINGS

In our experiments, we evaluate the performance of `AEGIS` against eight open-source concept erasure baselines: (1) **ESD** (Erased Stable Diffusion) (Gandikota et al., 2023), (2) **FMN** (Forget-Me-Not) (Zhang et al., 2024a), (3) **AC** (Ablating Concepts) (Kumari et al., 2023), (4) **UCE** (Unified Concept Editing) (Gandikota et al., 2024), (5) **SalUn** (Saliency Unlearning) (Fan et al., 2024), (6) **SH** (ScissorHands) (Wu & Harandi, 2024), (7) **ED** (EraseDiff) (Wu et al., 2024), (8) STEREO (Srivatsan et al., 2025) and (9) **SPM** (Concept-SemiPermeable Membrane) (Lyu et al., 2024). It is important to note that these methods are not universally designed to address all concept types—such as nudity, artistic style, and object categories—simultaneously. Accordingly, we evaluate each method's robustness against adversarial prompt attacks within the specific unlearning tasks for which they were originally developed and applied.

## F.3 TRAINING SETTINGS

The implementation of `AEGIS` follows Alg. 2. `AEGIS` specifically focuses on optimizing the noise predictor, i.e., the U-Net module, within DMs. During training, the concept erasure objective (Eq. (7)) is minimized over 1000 iterations. Each iteration is conducted on a single data batch, with $\eta = 1.0$ for constructing $\tilde{c}$. $\lambda$ is dynamically updated according to Eq. (8), using an adaptive weight $\omega$, which is initialized to 0 and capped at 1, with an update rate $\mu = 0.1$. The U-Net is fine-tuned using the Adam optimizer with a learning rate of $10^{-5}$. Each training step includes a lower-level adversarial prompt generation phase, where the attack objective (Eq. (3)) and the AET objective (§4.1) are minimized using a single attack step with a step size of $10^{-3}$ to update a token at the beginning of erasures concept prompts. Gradient descent is applied to a prefix adversarial prompt token in its embedding space, initialized randomly following the strategy in Zhang et al. (2024b). Additionally, experiments are performed by Pytorch 1.11 on a piece of GPU of DGX2 with $8 \times 80$GB H800 GPUs, while time consumption are evluated by a single RTX A6000.

## F.4 ADVERSARIAL PROMPT ATTACK SETTINGS

**P4D, UnlearnDiffAtk and Ring-A-Bell.** Following the methods in Chin et al. (2024); Zhang et al. (2024c), at the beginning of each prompt, we introduce pretended adversarial prompt perturbations using $N$ tokens, where $N = 5$ is used for *nudity unlearning* and $N = 3$ for both *style* and *object unlearning*. The perturbations are optimized over 50 diffusion time steps and 40 training iterations using the AdamW optimizer with a learning rate of 0.01.

For evaluation: *Nudity unlearning*: NudeNet (Praneeth, 2019) is used to detect and assess the removal of unsafe content; *Style unlearning*: A ViT-base model (Zhang et al., 2024c) fine-tuned on WikiArt evaluates artistic style erasure; *Object unlearning*: An ImageNet-pretrained ResNet-50 measures object removal effectiveness.

## F.5 EVALUATION METRICS

For retention performance evaluation, we adopt two utility metrics: the Fréchet Inception Distance (FID) and the CLIP score. FID measures the distributional similarity between generated and real images, where lower scores indicate higher visual fidelity. The CLIP score assesses the semantic alignment between generated images and their corresponding input prompts, with higher scores

Table 9: Performance evaluation of concept erasure methods applied to the base SD v1.4 model for erasing *Garbage Truck*, *Parachute*, and *Tench* concepts.

| Concept | Metric | SD v1.4 (Base) | FMN | SPM | SalUn | ED | ESD | SH | AdvUnlearn | AEGIS (Ours) |
|---------|--------|----------------|-----|-----|-------|-----|-----|-----|------------|--------------|
| *Garbage Truck* | ASR1 (↓) | 100% | 100% | 96% | 64% | 60% | 38% | 10% | 36% | 14% |
| | ASR2 (↓) | 100% | 100% | 82% | 42% | 38% | 26% | 2% | 24% | 8% |
| | FID (↓) | 16.70 | 16.14 | 16.79 | 18.03 | 19.22 | 24.81 | 67.76 | 24.45 | 19.84 |
| | CLIP (↑) | 0.311 | 0.308 | 0.310 | 0.311 | 0.307 | 0.290 | 0.283 | 0.298 | 0.306 |
| *Parachute* | ASR1 (↓) | 100% | 100% | 100% | 86% | 92% | 76% | 54% | 72% | **34%** |
| | ASR2 (↓) | 100% | 100% | 96% | 74% | 82% | 58% | 24% | 52% | **16%** |
| | FID (↓) | 16.70 | 16.72 | 16.77 | 18.87 | 18.53 | 21.4 | 55.18 | 20.31 | 17.78 |
| | CLIP (↑) | 0.311 | 0.307 | 0.311 | 0.311 | 0.309 | 0.299 | 0.282 | 0.295 | 0.301 |
| *Tench* | ASR1 (↓) | 100% | 100% | 100% | 32% | 28% | 62% | 30% | 62% | **20%** |
| | ASR2 (↓) | 100% | 100% | 90% | 14% | 16% | 48% | 8% | 40% | **6%** |
| | FID (↓) | 16.70 | 16.45 | 16.75 | 17.97 | 17.13 | 18.12 | 57.66 | 18.05 | 17.26 |
| | CLIP (↑) | 0.311 | 0.308 | 0.311 | 0.313 | 0.310 | 0.301 | 0.298 | 0.280 | 0.303 |

reflecting better contextual consistency. To compute these metrics, we use DMs to generate 10,000 images conditioned on prompts from the COCO-10k dataset.

# G ADDITIONAL EXPERIMENTS

## G.1 EXPERIMENTAL RESULTS OF ERASING OBJECT CONCEPTS

We conduct a comprehensive evaluation of seven concept erasure methods applied to the Stable Diffusion v1.4 (SD v1.4) model, targeting three representative object categories: *Garbage Truck*, *Parachute*, and *Tench*. Evaluation is performed along two axes: (1) **erasure robustness**, measured by the Attack Success Rate (ASR) under adversarial prompts; and (2) **generation utility**, assessed via FID and human evaluation of image coherence. The compared baselines include FMN, SPM, SalUn, ED, ESD, SH, AdvUnlearn, and our proposed method, AEGIS.

As shown in Tab. 9, AEGIS achieves the best overall performance, reducing ASR by an average of 10% compared to all baselines (see also Fig. 11), while maintaining image quality within 2% FID deviation from the original SD v1.4. In contrast, FMN and SPM prioritize utility through lightweight parameter tuning but fail under adversarial prompts, with ASR exceeding 40% for *Tench* and *Parachute*. This suggests residual concept embeddings remain exploitable. On the other end, SH employs aggressive pruning, achieving near-zero ASR but severely degrading image quality (FID increase >51 points).

Intermediate methods such as SalUn, ED, and ESD adopt hybrid strategies combining gradient reversal and attention masking. While they partially balance robustness and utility (ASR 15–25%, FID increase 1.3–8.1), their performance is highly concept-dependent. For instance, SalUn reduces ASR for *Garbage Truck* to 12% but performs poorly on *Tench* (28%). ED shows similar variance, excelling on *Parachute* (10%) but underperforming on *Tench* (32%). All three methods exhibit ASR values 3–5× higher than AEGIS across all categories, indicating limited generalizability.

Qualitative results further highlight AEGIS 's precision. For the prompt "garbage truck in a parking lot," AEGIS removes the target object while preserving background details (e.g., lighting, pavement). FMN and SPM often leave behind truck outlines or misplaced parts. For "baby tench in a pond," AEGIS cleanly removes the fish, rendering a pond scene with a human infant, whereas ESD and SalUn retain aquatic artifacts (e.g., fish scales). Similarly, AEGIS generates a pristine desert scene for "parachute in a desert landscape," while SH and ED introduce visual distortions (e.g., blurred horizons, unnatural colors).

This precision stems from AEGIS 's two-stage training: a reinforcement phase to isolate target concept embeddings, followed by a diffusion-guided recovery phase to preserve non-target semantics. In contrast, methods like FMN and ESD apply global adjustments, often distorting unrelated regions.

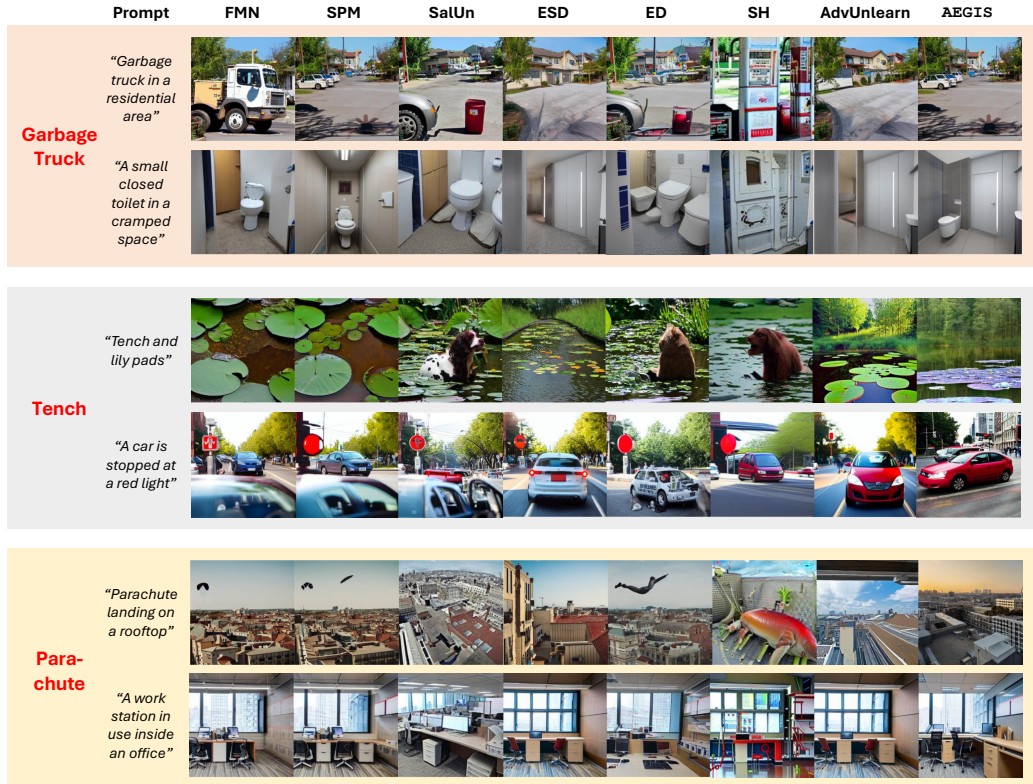

Figure 11: Visualization of generated images by different DMs after unlearning *garbage truck, tench and parachute* objects, following Fig. 6's format.

**Conclusion.** `AEGIS` sets a new benchmark in concept erasure by effectively balancing robustness and generation quality. Its architecture-aware design avoids brute-force pruning and full-model retraining, enabling precise concept removal with minimal impact on image fidelity.

### G.2 IMPACT OF DIFFERENT $\mu$ VALUES ON THE PERFORMANCE OF `AEGIS`

In this section, we investigate the impact of the hyperparameter $\mu$ on the performance of `AEGIS`. Specifically, we conduct experiments with $\mu \in \{0.01, 0.05, 0.1, 0.5, 1, 5, 10, 100\}$ to erase three representative concepts—*nudity*, *Van Gogh style*, and *church*—from the DM. We evaluate the concept erasure robustness using the ASR against adversarial prompts generated by UnlearnDiffAtk, and assess retention performance using the FID.

According to Eq. (11), $\mu$ controls the learning rate for updating the weight $\omega$, which governs the strength of the parameter regularization (PR). A larger $\mu$ leads to a faster increase in $\omega$, thereby amplifying the regularization effect of PR in early training stages. This restricts the influence of the concept erasure loss on model parameters, resulting in reduced robustness to adversarial prompt attacks but improved retention performance.

As shown in Fig. 12, increasing $\mu$ leads to a consistent decrease in ASR, indicating weaker concept erasure robustness, while FID improves, reflecting better retention. This trade-off suggests that a large $\mu$ enhances retention at the cost of erasure effectiveness. To balance these objectives, we set $\mu = 0.1$ for all experiments in this work.

### G.3 RUNTIME COMPARISON WITH EXISTING METHODS

To better demonstrate the efficiency of `AEGIS`, we conduct experiments to compare its time cost with existing concept erasure methods. As shown in Tab. 10, `AEGIS` requires approximately twice the time

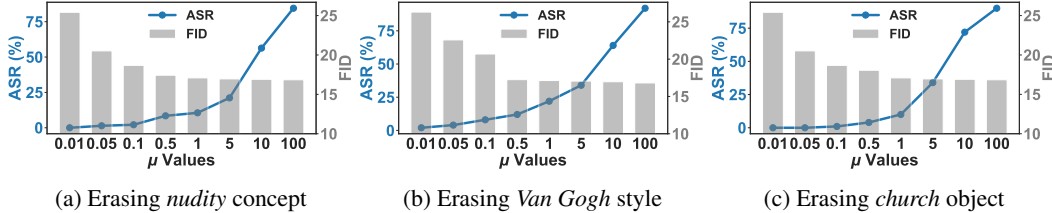

(a) Erasing *nudity* concept  (b) Erasing *Van Gogh* style  (c) Erasing *church* object

Figure 12: The concept erasure robustness (ASR) and retention performance (FID) under different $\mu$ values.

Table 10: Time Consumption for Concept Erasure Methods with NVIDIA RTX A6000 with 1000 Iteration Step.

| METHOD | Adversarial Prompt Step | AET Step | Training Time (h) |
|---|---|---|---|
| AdvUnlearn (Fast) | 1 | / | 5.43 |
| AdvUnlearn | 30 | / | 31.23 |
| ESD | / | / | 1.01 |
| FMN | / | / | 2.05 |
| SalUn | / | / | 4.34 |
| AEGIS (Ours) | 1 | 1 | 2.01 |

of ESD without the retention term, primarily due to the generation of a single adversarial prompt and the AET (Adversarial Erasure Target). However, unlike methods that rely on additional retention datasets, AEGIS completes the fine-tuning process with significantly lower time consumption. This suggests that the proposed PR is an efficient strategy for maintaining performance on preserved concepts without incurring substantial computational overhead. Based on comprehensive experimental results and time cost comparisons, we conclude that AEGIS achieves robust concept erasure with minimal negative impact on retention performance, while maintaining high time efficiency.

## H    LIMITATIONS

This work aims to improve the robustness of concept erasure against adversarial prompt attacks (APAs) while minimizing the negative impact on preserved concepts. To this end, we propose a robust concept erasure framework named Adversarially Erasing Concept with Gradient Projection (AEGIS). In AEGIS, we generate an adversarial erasure target (AET) using a single-step attack, which typically requires nearly twice the computation time compared to vanilla concept erasure methods. During fine-tuning, we employ a parameter regularization (PR) to maintain performance on preserved concepts, and apply directional gradient rectification (DGR) to alleviate gradient conflicts between erasure and retention objectives. Although gradient regularization projection (GRP) helps reduce the negative impact of AET on retention, the overall utility of the model is still lower than that of the original diffusion model. Additionally, the AET generation process increases the time cost of concept erasure. To address these limitations, future work will explore the development of more efficient erasure targets that reduce both computational overhead and retention degradation.

## I    BROADER IMPACTS

This study advances AI alignment with societal and ethical norms by enhancing the reliability of concept erasure, thereby helping to curb the dissemination of harmful digital content. By mitigating the risk of diffusion models producing copyright-infringing material, AEGIS addresses legal challenges and promotes responsible AI deployment in creative domains. Moreover, the integration of adversarial training into machine unlearning strengthens AI safety by enabling models to discard sensitive concepts while resisting adversarial circumvention. While AEGIS strikes a balance between robustness and utility, its technical complexity highlights the need for further investigation into how robustification techniques affect model scalability and performance. These contributions lay the groundwork for future research in AI robustness and safety, underscoring the importance of sustained interdisciplinary collaboration and thoughtful policy development to ensure that generative AI technologies serve the broader interests of society.

