# OpenReview forum: "AEGIS: Adversarial Target-Guided Retention-Data-Free Robust Concept Erasure from Diffusion Models"
_ICLR.cc/2026/Conference — ICLR 2026 Poster_

### Official Review · Reviewer_XpvQ · 2025-10-16

**Soundness:** 3
**Presentation:** 2
**Contribution:** 3
**Rating:** 6
**Confidence:** 1

**Summary:**

Concept erasure helps stop diffusion models (DMs) from generating harmful content; but current methods face robustness–retention trade-off. This paper introduces a novel method, Adversarial Erasure with Gradient-Informed Synergy (AEGIS), aimed at enhancing the robustness of concept erasure under adversarial prompt attacks (APAs). Extensive experiments show that AEGIS markedly reduces attack success rates across various concepts while maintaining or improving FID/CLIP versus advanced baselines.

**Strengths:**

1. The paper conducts extensive experiments and demonstrates significant improvements over existing methods.
2. The paper provides both empirical and theoretical support for the proposed method.

**Weaknesses:**

1. Placing the related work section in the main body of the paper would improve readability. In addition, it is difficult to understand some details when reading only the main text.
2. Some fine-tuning details are missing, such as the amount of data used and the required memory.
3. Are all the results in Table 10 obtained using 8 × 80GB H800 GPUs?
4. Typographical error: “¿” in line 1432.


Overall, this paper is of good quality. However, since I am not very familiar with this research field, I set my confidence level to 1.

**Questions:**

See weaknesses.

---

> ### Author Response · Authors · 2025-11-21
> **Response to Reviewer XpvQ**
>
> Dear Reviewer,
>
> We thank the reviewer for the constructive feedback. Below, we respond item-by-item and clarify the constraints and missing details. We will incorporate the proposed fixes in the final version.
>
> ---
> ***W1 (Related Work):***
> ---
>
> **R1:** Given the ICLR restriction of a 9-page main text, the comprehensive Related Work section has been moved to the appendix to maintain the integrity and completeness of the motivation, methodology, and experimental sections. For the final version, utilizing the allowed 1-page extension, we slightly expand relevant contextual details in the Introduction and Method sections and incorporate brief in-line references to key prior works.
>
> ---
> ***W2 (Detailed Computation of SD Fine-tuning):***
> ---
>
> **R2:** **A. Data usage:**
> Training for erasure: Retention-data-free. We fine-tune the U-Net for 1000 iterations with batch size 1, using only the erased concept prompt and the automatically generated adversarial prompt/AET. Utility evaluation: 10,000 images using COCO-10k prompts for FID and CLIP.
>
> **B. APA evaluation:** Nudity (I2P-derived prompts + NudeNet), Van Gogh (protocol following Gandikota et al., 2023), and Objects (50 GPT-4 prompts per Imagenette class).
>
> **C. Memory and hardware:** **GPU usage clarification:** We omitted a detail. All experiments were executed on a single H800 GPU; runs were scheduled on a DGX2 server that has 8 × 80GB H800 GPUs. Since previous methods evaluate the time-consumption in a single Nvidia RTX A6000 GPU, the experiments of Table 10 in Appendix H.4 adopt the same device. The detailed GPU memory and time consumption comparison is shown in **Table E.1**.
>
> **Table E.1: Time Consumption with NVIDIA RTX A6000 with 1000 Iteration Step.**
>
> |Method|GPU Memory|Training Time (h)|
> |:--|:--|:--|
> | AdvUnlearn (Fast)|67.5|5.43|
> | AdvUnlearn       |70.1|31.23|
> | ESD              |17.8|1.01 |
> | FMN              |17.9|2.05 |
> | SalUn            |30.8|4.34 |
> | AEGIS(Ours)      |31.2|2.01 |
>
> ---
> ***W3 (GPU Usage):***
> ---
> **R3:** No. Since previous methods evaluate the time-consumption in **a single Nvidia RTX A6000 GPU,** the experiments of Table 10 in Appendix H.4 adopt the same device. Other experiments were executed on **a single H800 GPU**; runs were scheduled on a DGX2 server that has 8 × 80GB H800 GPUs. We will add an explicit note below Table 10 and in Sec.~G.3 of the Appendix to avoid ambiguity.
>
> ---
> ***W4 (Typographical Error):***
> ---
>
> **R4:** We will remove the stray “¿” and re-check the manuscript for similar artifacts.

---

> > ### Comment · Reviewer_XpvQ · 2025-11-25
> >
> > Thank you for the rebuttal. Most of my concerns have been addressed, and I will keep my ranking score.

---

> > > ### Author Response · Authors · 2025-11-25
> > >
> > > Dear Reviewer XpvQ,
> > >
> > > Thanks for your further response!
> > >
> > > We're glad to hear that we've addressed most of your concerns. We truly appreciate your time and thoughtful consideration. Should you have any further questions or require additional clarifications, please don't hesitate to let us know. We'd be happy to provide more information.
> > >
> > > Best Regards,
> > >
> > >  Authors of Submission 943

---

### Official Review · Reviewer_hz7A · 2025-10-28

**Soundness:** 3
**Presentation:** 3
**Contribution:** 2
**Rating:** 6
**Confidence:** 4

**Summary:**

This paper proposed retention-data-free framework for the robust concept erasure of diffusion models. To be more specific, the adversarial erasure target (AET) is obtained through optimization instead of handpicking and retention data is avoided by finetuning with gradient regularization projection (GRP) to achieve better trade-off between robustness and retention.

**Strengths:**

1. Well-formulated motivation: This paper identifies a key weakness in existing concept erasure methods, which overrely on the single-instance erasure targets.
2. Retention-data-free: It is necessary to avoid the usage of additional datasets to maintain the model utility which might involve hidden bias.
3. Extensive experiments are convincing: This paper evaluates multiple baselines through three adversarial prompt attacks.
4. Good ablation study to isolate the effect of AET, PR and DGR.

**Weaknesses:**

1. The integration of AER and GRP is novel, however, both ideas are drawn from well-known paradigms: adversarial target optimization and gradient surgery. It is good to further clarify the differences compared to existing paradigms.
2. Insufficient analysis on computational overhead. Computation reduction brought by AET is mentioned in the paper, however, the paper lacks detailed computation
3. Evaluation scope somewhat narrow. Only three representative concepts (nudity, Van Gogh, Church) are analyzed.
4. Limited interpretation of semantic centers. Visualization or embedding analysis of AET would be beneficial.

**Questions:**

1. How stable is the AET optimization process?
2. Scalability to large models (e.g., SDXL): The experiments focus on SD v1.4/v2.1. Have the authors attempted to extend AEGIS to higher-capacity models?

---

> ### Author Response · Authors · 2025-11-21
> **Response to Reviewer hz7A(Part 1/2: Response to Weakness 1, 2, and 3)**
>
> Dear Reviewer,
>
> We appreciate your comprehensive review of our paper and are grateful for your acknowledgment of its solid motivation and clear experimental improvements.
>
> ---
> ***W1 (Clarification of AET and GRP):***
> ---
>
> **R1:** We thank the reviewer for this important question. While AET and GRP are inspired by existing paradigms, they incorporate key innovations tailored for the specific challenges of robust concept erasure.
>
> **A. AET vs. Adversarial Target Optimization:** Standard adversarial target optimization, like in AdvUnlearn, generates an adversarial prompt ($c^*$) to replace the original erased prompt ($c_e$) in the erasure loss. This hardens the model against that specific adversarial instance. In contrast, our AET generates an adversarial target embedding ($c^\prime$) used to construct the safe target concept ($\tilde{c}$). As motivated in Section 3 (lines 190-194) and illustrated in Figure 9, this pushes the erased concept's representation away from its entire semantic cluster, not just a single adversarial point. This shift from attacking the input to optimizing the target is a crucial distinction that leads to class-level robustness, as empirically validated by the significant ASR reduction in Tables 1 and 2.
>
> **B. GRP vs. Gradient Surgery:** While GRP shares the goal of mitigating conflicting gradients with methods like Gradient Surgery, its mechanism is fundamentally different and data-free. Gradient Surgery requires gradients from multiple task-specific data batches to project one against another. Our GRP operates in a retention-data-free setting, managing the conflict between the erasure gradient ($g_e$) and a parameter regularization gradient ($g_r=\nabla\Vert\mathbf{\theta}-\mathbf{\theta}_0\Vert_2$ ). Furthermore, GRP is **asymmetric and dynamic:** it only rectifies the retention gradient ($g_r$) when it conflicts with the erasure gradient ($g_e$), and its strength ($\omega$) is dynamically adjusted to prioritize erasure early in training. This data-free, asymmetric, and dynamic design is novel and specifically addresses the robustness-retention trade-off without external data.
>
> ---
> ***W2 (Detailed Computation of AET):***
> ---
>
> **R2:** We appreciate the reviewer pointing this out. The primary overhead of AEGIS comes from the single-step AET generation per training iteration. To provide a clear comparison, we have added a detailed time in **Table D.1**. During the model fine-tuning, it takes about $684$ seconds for AET. Compared AEGIS with multiple-step AET, the single one reduces about $3257$ seconds for the model fine-tuning.
>
> ***Table D.1. Detailed Time Consumption on SDv1.4 with 1000 Iteration***
>
> |Method|Time(Seconds)|
> |:--|:--|
> |ESD(Baseline)|3636|
> |AEGIS(AET(Iteration=10))|10473|
> |AET                |684|
> |PR                 |468|
> |DGR                |2448|
> |AEGIS(Ours)        |7236|
>
>
>
> ---
> ***W3 (Narrow Evaluation):***
> ---
>
> **R3:** Thank you for this suggestion. Except the experimental results with nudity, Van Gogh, Church in the main text, we have conducted additional experiments on **four more object concepts: Garbage Truck, Parachute, Tench, and Cassette Player**. These results, presented in **Table D.2 (has been attacted to Appendix H.1 Table 9 and Figure 11)**, demonstrate that AEGIS consistently outperforms baselines across a wider variety of object types, confirming its generalizability. This strengthens our claim that AEGIS is effective beyond the three initial concepts.
>
> ***Table D.2 : Performance evaluation of concept erasure methods applied to the base SD v1.4 model for erasing *Garbage Truck*, *Parachute*, and *Tench* concepts.***
>
> |Concept|Metric|SD v1.4 (Base)|FMN|SPM|SalUn|ED|ESD|SH|AdvUnlearn|AEGIS (Ours)|
> |:--|:--|:--|:--|:--|:--|:--|:--|:--|:--|:--|
> | *Garbage Truck*|ASR1(↓)|100%|100%|96%|64%|60%|38%|10%|36%|14%|
> | | ASR2(↓)|100%|100%|82%|42%|38%|26%|2%|24%|8%|
> | | FID(↓) |16.70|16.14|16.79|18.03|19.22|24.81|67.76|24.45|19.84|
> | | CLIP (↑)|0.311|0.308|0.310|0.311|0.307|0.290|0.283|0.298|0.306|
> | *Parachute*|ASR1(↓)|100%|100%|100%|86%|92%|76%|54%|72%|**34%**|
> | | ASR2(↓)|100%|100%|96%|74%|82%|58%|24%|52%|**16%**|
> | | FID(↓) |16.70|16.72|16.77|18.87|18.53|21.4|55.18|20.31|17.78|
> | | CLIP (↑)|0.311|0.307|0.311|0.311|0.309|0.299|0.282|0.295|0.301|
> | *Tench* |ASR1(↓)|100%|100%|100%|32%|28%|62%|30%|62%|**20%**|
> | | ASR2(↓)|100%|100%|90%|14%|16%|48%|8%|40%|**6%**|
> | | FID (↓)|16.70|16.45|16.75|17.97|17.13|18.12|57.66|18.05|17.26|
> | | CLIP(↑)|0.311|0.308|0.311|0.313|0.310|0.301|0.298|0.280|0.303|

---

> ### Author Response · Authors · 2025-11-21
> **Response to Reviewer hz7A(Part 2/2: Response to Weakness 4 and Question 1, and 2)**
>
> ---
> ***W4 (Interpretation of Semantic Centers & Visualization):***
> ---
>
> **R4:** Thanks for your comment. We provide both visual and theoretical analysis of AET and semantic centers:
>
> * **Figure 2 (Radar Charts):** These charts explicitly visualize the Predicted Noise Distances. They show that before erasure (Fig 2a), synonyms like nudity and sexual are clustered. After AEGIS (Fig 2d), the distances ($d_2$) for all related variants increase significantly compared to ESD (Fig 2c), proving that AET successfully pushes the entire semantic cluster away from the original distribution.
> * **Figure 9 (Schematic): of Appendix D.1** This diagram visualizes the mechanism. It explains that traditional mapping (Fig 9a) keeps the target dangerously close to the decision boundary of the semantic center, causing leakage. AET (Fig 9b) creates a target that is distant from this center.
> * **Theoretical Grounding:** **Proposition 3.1** and **Definition 1** in Section 3 concretize the abstract idea of a "semantic center" into measurable noise prediction distances, providing a rigorous mathematical interpretation of the visualization.
>
> ---
> ***Q1 (Optimization of AET):***
> ---
>
> **RQ1:** The AET optimization is stable. It is a single-step gradient ascent update per training iteration (as described in Section 4.1 and detailed in Appendix C, Alg. 1), which is a standard and stable procedure in adversarial training. This is more stable than multi-step optimization, which can sometimes lead to overly specific and less generalizable adversarial targets. The consistent and significant performance improvements shown across all experiments (Tables 2, 3, 4, 5, 9) empirically validate the stability and effectiveness of this process.
>
> ---
> ***Q2 (Experiments on More SD Versions):***
> ---
>
> **RQ2:** Thanks for your comment. Evaluation concept erasure performance on SD v1.4/v2.1 is the standard and widely accepted setting in the concept erasure research area. Although there is no method that provides the robust concept erasure results on SDXL, to satisfy your requirement, we add experiment as **Table D.3**. Given the large number of parameters in SDXL, we only selected the base Unet for fine-tuning with learning rate as $1\times 10^{-6}$.  Compared with existing methods, our AEGIS still achieves **better robustness-retention trade-off**. Moreover, similar to SDv2.1, SDXL differs from SD v1.4 in data curation, text encoder, and model capacity. These changes make internal concept representations more entangled, increasing the difficulty of class-level erasure.
>
> ***Table D.3:Performance evaluation of concept erasure methods applied to the base SDXL model for erasing *Nudity*, *Van Gogh*, and *Church* concepts.***
>
> |Concept|Metric|SDXL v1.0 (Base)|FMN|SPM|ESD|AGE|RECE|RECELER|AdvUnlearn|AEGIS(Ours)|
> |:--|:--|:--|:--|:--|:--|:--|:--|:--|:--|:--|
> |*Nudity*|ASR1(↓)|89.44%|87.32%|88.02%|80.28%|77.46%|74.65%|68.31%|72.54%|**38.02%**|
> | |ASR2(↓)       |85.92%|81.69%|80.28%|71.13%|82.39%|67.61%|56.34%|53.52%|**30.28%**|
> | |ASR3(↓)       |78.17%|66.20%|45.07%|38.03%|68.31%|42.25%|34.51%|35.92%|**21.13%**|
> | |FID (↓)       |11.15 |11.51 |11.59 |12.31 |11.34 |12.42 |12.63 |12.75 |  11.77   |
> | |CLIP(↑)       |0.341 |0.340 |0.335 |0.324 |0.337 |0.329 |0.319 | 0.302|  0.331   |
> | *Van Gogh*|ASR1(↓)|94%| 86%  | 88%  | 64%  | 82%  | 74%  | 62%  | 64%  | **44%**  |
> | |ASR2(↓)          |90%| 58%  | 68%  | 56%  | 64%  | 60%  | 58%  | 52%  | **32%**  |
> | |ASR3(↓)          |84%| 72%  | 56%  | 42%  | 56%  | 66%  | 46%  | 44%  | **26%**  |
> | |FID (↓)       |11.15 |11.38 |11.33 |11.85 |11.24 |12.25 |12.34 |12.04 |  11.61   |
> | |CLIP(↑)       |0.341 |0.331 |0.332 |0.324 |0.337 |0.324 |0.329 |0.327 |  0.332   |
> | *Church*|ASR1(↓)  |96%| 92%  | 94%  | 86%  | 84%  | 76%  | 68%  | 72%  | **34%**  |
> | |ASR2(↓)          |92%| 90%  | 92%  | 80%  | 78%  | 74%  | 54%  | 44%  | **16%** |
> | |ASR3(↓)          |88%| 84%  | 82%  | 74%  | 56%  | 64%  | 48%  | 40%  | **24%** |
> | |FID (↓)       |11.15 |11.35 |11.31 |11.85 |11.23 |12.29 |12.13 |11.94 |11.63|
> | |CLIP(↑)       |0.341 |0.332 |0.335 |0.324 |0.340 |0.320 |0.322 |0.323 |0.329|

---

> ### Author Response · Authors · 2025-11-25
> **[Submission 943] Follow-up: New Experiments (SDXL, New Concepts) and Methodological Clarifications**
>
> Dear Reviewer hz7A,
>
> We hope this message finds you well.
>
> We are writing to kindly invite you to review our detailed response to your comprehensive feedback. We greatly appreciate your acknowledgment of our paper's solid motivation and improvements.
>
> In our rebuttal, we have carefully addressed your specific questions regarding the technical details of AET/GRP and conducted extensive new experiments to broaden the evaluation:
>
> **1. Methodological Clarifications (W1, W4, Q1):**
> *   **Distinction of AET & GRP (W1):** We have explicitly clarified that **AET** targets the *semantic cluster* via target embedding optimization (unlike input-level attacks in AdvUnlearn), and **GRP** is a unique *data-free, asymmetric, and dynamic* mechanism (fundamentally different from Gradient Surgery).
> *   **Theoretical Interpretation (W4):** We provided visual (Radar Charts, Fig 9) and theoretical analysis (Prop 3.1) to ground the concept of "semantic centers."
> *   **Stability (Q1):** We confirmed the stability of the single-step AET optimization.
>
> **2. New Experimental Evidence (W2, W3, Q2):**
> *   **Broader Generalization (W3):** As requested, we evaluated **4 additional object concepts** (Garbage Truck, Parachute, Tench, Cassette Player). **Table D.2** shows AEGIS consistently outperforms baselines.
> *   **SDXL Evaluation (Q2):** We added experiments on **SDXL** (Table D.3), demonstrating that AEGIS maintains a superior robustness-retention trade-off even on larger, more complex models.
> *   **Efficiency Analysis (W2):** We added **Table D.1** to show that the time overhead of single-step AET is minimal compared to the performance gains.
>
> As the discussion period is limited, we would be very grateful if you could take a moment to check these new results and clarifications. **If you find that our additional experiments and detailed explanations have adequately addressed your concerns and strengthened the paper, we would highly appreciate it if you could consider raising your score.**
>
> We remain available for any further questions you might have.
>
> Best regards,
>
> The Authors of Submission 943

---

> > ### Comment · Reviewer_hz7A · 2025-11-26
> >
> > Thanks for the detailed responses and new experimental results. All my concerns have been resolved. I will raise my rating to 8.

---

> > > ### Author Response · Authors · 2025-11-26
> > >
> > > Dear Reviewer hz7A,
> > >
> > > We are truly encouraged and grateful for your positive feedback and your decision to raise the rating to 8.
> > >
> > > We are delighted to learn that our detailed response and the new experimental results have successfully resolved all your concerns. Your constructive feedback throughout the review process has been invaluable in helping us identify areas for improvement and strengthen the validation of our method.
> > >
> > > We commit to integrating the new experimental results and clarifications into the final version of the paper.
> > >
> > > Thank you once again for your time, support, and contribution to improving our work.
> > >
> > > Best regards,
> > >
> > > The Authors of Submission 943

---

> > > ### Author Response · Authors · 2025-11-26
> > > **[Submission 943] Sincere Thanks for Your Support and the Rating of 8**
> > >
> > > Dear Reviewer hz7A,
> > >
> > > We are truly encouraged and grateful for your positive feedback and your decision to **raise the rating to 8**.
> > >
> > > We are delighted to learn that our detailed response and the new experimental results have successfully resolved all your concerns. Your constructive feedback throughout the review process has been invaluable in helping us identify areas for improvement and strengthen the validation of our method. We commit to integrating these new results and clarifications into the final version.
> > >
> > > **We noticed the system score has not yet updated, but we understand this may take a moment or require a manual update. We appreciate your attention to this final step before the discussion period closes.**
> > >
> > > Thank you once again for your time, support, and contribution to improving our work.
> > >
> > > Best regards,
> > >
> > > The Authors of Submission 943

---

### Official Review · Reviewer_G3SD · 2025-10-30

**Soundness:** 3
**Presentation:** 3
**Contribution:** 3
**Rating:** 6
**Confidence:** 3

**Summary:**

This paper AEGIS proposes a robust defense algorithm against adversarial attack on concept-erased diffusion models. It shows the trade-off between erasure robustness and quality preservation, by Adversarial Erasure Target and Gradient Regularization Projection separately. AET approximates the semantic center of the concept to be erased, enables class-level removal instead of single prompt. GRP preserves the quality using parameter regularization and a novel gradient surgery technique that selectively projects away retention gradients conflicting with the erasure objective. Author's experiments show AEGIS significantly reduces attack success rates by 5.31~24% across various concepts and state-of-the-art attacks like P4D and UnlearnDiffAtk, while maintaining or improving image quality over baselines.

**Strengths:**

1. Instead of defending a single prompt, AEGIS dynamically optimizes a target prompt to approximate the semantic center of the concept being erased. With both AEGIS and GRP, it claims to achieve better tradeoff and supported by experiment results.
2. Experiment is thorough - it validates its method across multiple concept types (object, style, nudity), model versions (SD v1.4, v2.1), and against a suite of strong adversarial attacks (P4D, UnlearnDiffAtk), proving its generalizability and robustness.

**Weaknesses:**

1. it looks like it's sensitive to hyper-parameters such as w in 5.3 ablation study. how to pick the best value for unlearning a new concept?
2. how to scale if the model needs to unlearn many concepts or objects?

**Questions:**

1. what about attack in the input image or even embeddings?

---

> ### Author Response · Authors · 2025-11-21
> **Response to Reviewer G3SD**
>
> Dear Reviewer,
>
> Thank you for your insightful feedback. We are encouraged that the reviewer found our work addresses the critical robustness-retention trade-off in concept erasure. We address the raised weaknesses and questions below.
>
> ---
> ***W1 (Sentivety of $w$):***
> ---
>
> **R1:** Thank you for this important question. The hyper-parameter $\omega$ of Gradient Regularization Projection (GRP) is **updated dynamically from $0$ to $1$**, following Eq.11 with step size $\mu$. It starts with $\omega=0$ to **prioritize erasure** and gradually increases $\omega$ as the erasure and retention gradients start to conflict (see Fig. 10), thus strengthening the retention constraint. In Sec. 5.3 ablation study, to show the impact of dynamic mechanism of $\omega$ on the performance of AEGIS, we provide the result of AEGIS with the fixed $\omega=1$.
>
> During the fine-tuning with AEGIS, our empirical results suggest that the default setting (the step size $\mu=0.1$ of updating $\omega$) generalizes well across different concept types (sensitive, style, object) as shown in Tables 2, 3, and 4. This is because the dynamic adjustment is driven by the observed gradient conflict during training, rather than a pre-set value tailored to a specific concept.
>
> To further clarify this, we have provided detailed analysis on the impact of $\mu$ (which controls $\omega$) in Appendix H.2 (Figure 12). This new section shows that while performance varies, a wide range of $\omega$ values (from $0.05$ to $0.5$) can yield strong results, with $\mu=0.1$ offering a robustness-retention trade-off. We will also add a sentence in Section 5.1 (Experimental Setup) to explicitly state that our chosen hyper-parameters demonstrate good generalization across tested concepts, providing a reliable starting point for new concept erasure tasks. In practice, no per-concept tuning is required: we start with $\omega=0$ and update it by Eq.11 with $\mu=0.1$. If one observes high ASR, slightly increase $\mu$; if FID/CLIP drops, decrease $\mu$. This simple schedule worked across sensitive/style/object concepts without hand-tuning.
>
>
> ---
> ***W2 (Scale Many Concept Erasure):***
> ---
>
> **R2:** Thank you for this excellent question regarding the scalability of concept erasure. This is a crucial consideration for real-world applications. Since existing methods focus on erasing single concept for each fine-tuning operation, we do not provide results for many concept erasures. To satisfy the effectiveness of our method, we design a concept erasure method for one-time fine-tuning. We combine an erasure concept set $\mathcal{C}$={*nudity*, *church*, *Van Gogh*}. During the concept erasure procedure, AEGIS is optimized following the objective as
>
> \begin{equation}
>     \min_{\theta}\big[\max_{\tilde{c}}\mathbb{E} _ {t,c\in \mathcal{C}} [\Vert\epsilon_\theta(z_t|c^{*})-\epsilon_{\theta_0}(z_t|\tilde{c})\Vert_2^2]+\frac{\lambda}{2} \Vert\theta-\theta_0\Vert^2_2\big],
> \end{equation}
>
> which means we randomly select one instance of $\mathcal{C}$ for each concept erasure fine-tuning step.
>
> We acknowledge that a comprehensive empirical evaluation of this multi-concept erasure scenario is an important next step. We consider this a valuable direction for future work and plan to investigate the performance and potential nuances of this scaled-up approach in our follow-up research.
>
> ---
> ***Q1 (Attacks in Other Spaces):***
> ---
> **RQ1:** Thank you for this insightful question, which prompts us to clarify the precise threat model our work addresses. AEGIS is designed for the text-to-image (T2I) generation pipeline, where the model's input is a text prompt, not an image. Therefore, adversarial attacks on an input image are not applicable to the primary task we address. The state-of-the-art Adversarial Prompt Attacks (APAs) that we evaluate against, such as P4D and UnlearnDiffAtk, do not operate on discrete text tokens directly. Instead, they perform their optimization in the **continuous text embedding space** to find perturbations that can reactivate the erased concept. Our defense is specifically designed to counter this.

---

### Official Review · Reviewer_Tzbr · 2025-10-30

**Soundness:** 2
**Presentation:** 3
**Contribution:** 2
**Rating:** 4
**Confidence:** 3

**Summary:**

The paper addresses the problem of robust concept erasure in diffusion models. The paper proposes Adversarial Erasure with Gradient-Informed Synergy (AEGIS), improving erasure robustness and retain performance after unlearning.

**Strengths:**

The robustness of diffusion model unlearning is a highly important problem. The idea of adversarial erasure target (AET) is novel and well-motivated, and the authors provide detailed and solid explanations for their proposed methods.

The robustness of AEGIS is validated on multiple attacks. The authors also compare AEGIS with multiple baselines.

The figures and illustrations are of good quality.

**Weaknesses:**

1. The paper lacks comprehensive evaluations on the retain performance. Currently, FID and the CLIP score are used. However, common DM unlearning benchmarks such as UnlearnCanvas [1] include evaluation metrics such as in-domain retain accuracy (IRA) and cross-domain retain accuracy (CRA). Since the authors claim AEGIS has great robustness–retention trade-off, a more comprehensive retention evaluation is needed.

2. The motivation of Parameter Regularization (PR) and Directional Gradient Rectification (DGR) seems unclear. It seems that AET alone can achieve concept erasure while retaining model utility, and PR and DGR can be added upon any unlearning methods, and they are not specifically related to AET. Possibly, AET seriously degrades the model's utility, so PR and DGR are employed to balance retention performance. However, this motivation still seems weak: why not try other methods that do not incur additional computation costs, such as tuning the retain loss coefficient, or adjusting the learning rate?

3. For the baselines, the paper did not include more recent methods on robust concept erasure, such as STEREO [2]. This CVPR 2025 paper addresses the same problem as yours, and I think it should be mentioned and compared.

4. The paper lacks run-time and GPU memory comparison between different methods. I am concerned about the increased computation cost brought by AEGIS, particularly the DGR part. Could the authors discuss this possible trade-off?

[1] UNLEARNCANVAS: A Stylized Image Dataset for Enhanced Machine Unlearning Evaluation in Diffusion Models

[2] STEREO: A Two-Stage Framework for Adversarially Robust Concept Erasing from Text-to-Image Diffusion Models

**Questions:**

1. Can the authors explain why they employ the CLIP score to evaluate retention? As shown in Table 2-3, the CLIP score has very little changes before and after unlearning. Besides, all the unlearning methods have similar CLIP scores (ranging from 0.29 to 0.31). This gives me the impression that the CLIP score is slightly affected by the unlearning process. In this case, how can it serve to faithfully evaluate the retention performance of different methods?

2. Why and how does Directional Gradient Rectification (DGR) contribute to robustness? In Table 6, 'AEGIS w/o DGR' has significantly higher ASR compared to AEGIS. This result is somehow confusing to me: in theory, DGR serves to improve retention performance by resolving the confliction between forget gradient and retain gradient. However, the authors did not explain how it contributes to the robustness gain.

3. For DGR, have the authors tried using the moving average of gradients in the gradient projection process? This might yield better performances, according to [1].

[1] GRU: Mitigating the Trade-off between Unlearning and Retention for LLMs

---

> ### Author Response · Authors · 2025-11-21
> **Response to Reviewer Tzbr (Part 1/4: Response to Weakness 1)**
>
> Dear Reviewer,
>
> We thank the reviewer for the thoughtful feedback. Below, we respond point-by-point and revise our framing accordingly.
>
> ---
> ***W1 (On retention evaluation beyond FID/CLIP (IRA/CRA on UnlearnCanvas)):***
> ---
>
> **R1:** Thank you for your valuable feedback. We appreciate your suggestion for a more comprehensive evaluation of retention performance, particularly the inclusion of IRA and CRA.
>
> **1.Current Retention Metrics and Limitations:**
> IRA and CRA in UnlearnCanvas are well-suited for **style/object domains where “innocent prompts” are structurally defined**. Nudity-type sensitive concepts lack such **domain taxonomy**, so we report IRA/CRA only for styles/objects (consistent with UnlearnCanvas), while keeping FID/CLIP for general utility.
>
> **2. Evaluation with IRA and CRA:**
> To thoroughly address your request and demonstrate AEGIS's robustness–retention trade-off in a setting where IRA and CRA are applicable, we conducted additional experiments specifically on style and object concepts, following the experimental settings of UnlearnCanvas and evaluation settings of our paper. As shown in **Table B.1**, compared with STEREO[2], our AEGIS reduces ASR (evaluated by P4D(ASR1), UnlearnDiffAtk(ASR2), and Ring-A-Bell(ASR3), lower is better) **by average $13.81\%$ and $11.30\%$** with **IRA improvement $13.36\%$ and $15.27\%$**, and **CRA improvement $16.57\%$ and $13.63\%$** in style and object concept erasure. These results, based on the UnlearnCanvas framework, robustly verify that our **AEGIS effectively balances the robustness and retention of concept erasure**, offering a more comprehensive and controlled unlearning capability for diffusion models.
>
> ***Table B.1 Comprehensive Performance Overview Evaluation***
> |Method|Style ASR1($\downarrow$)|Style ASR2($\downarrow$)|Style ASR3($\downarrow$)| Style IRA ($\uparrow$)|Style CRA ($\uparrow$) |Object ASR1($\downarrow$) |Object  ASR2 ($\downarrow$)|Object ASR3($\downarrow$)|Object IRA($\uparrow$)|Object CRA($\uparrow$)| FID ($\downarrow$)|
> |:--|:--|:--|:--|:--|:--|:--|:--|:--|:--|:--|:--|
> |ESD  |76.67%|57.56%|50.83%|80.97%|93.96%|74.17%|65.41%|56.48%|55.78%|44.23%|65.55|
> |FMN  |70.82%|55.08%|56.67%|56.77%|46.60%|40.83%|32.57%|27.61%|90.63%|73.46%|131.37|
> |UCE  |67.50%|37.50%|33.38%|60.22%|47.71%|57.50%|48.94%|41.85%|39.35%|34.67%|182.01|
> |CA   |58.85%|34.37%|38.23%|**96.01%**|92.70%    |39.75%|25.37%|19.54%|90.11%|81.97%|**54.21**|
> |SalUn|68.54%|50.42%|45.88%|90.39%|95.08%  |65.53%|52.72%|45.82%|96.35%|**99.59%**|61.05|
> | SPM |61.67%|32.72%|29.16%|92.39%|84.33%|61.09%|49.61%|38.27%|90.79%|81.65%|59.79|
> |ED|69.17%|54.49%|50.31%|73.91%|**98.93%**|58.42%|54.64%|48.36%|94.03%|48.48%|81.42|
> |SH|66.67%|40.83%|35.83%|80.42%|43.27%|46.63%|40.09%|32.15%|81.15%|67.99%|119.34|
> |STEREO[2]|41.45%|24.14%|20.59%|82.48%|73.68%    |29.56%|19.75%|14.28%|81.37%|76.62%|72.84|
> |AdvUnlearn|37.71%|29.85%|25.83%|86.48%|71.65%   |36.39%|21.83%|17.65%|85.36%|77.84%|65.62|
> |AEGIS (Ours)|**24.18%**|**11.67%**|**8.89%**|95.84%|90.25%|**18.62%**|**6.83%**|**4.25%**|**96.64%**|90.25%|57.06|

---

> ### Author Response · Authors · 2025-11-21
> **Response to Reviewer Tzbr (Part 2/4: Response to Weakness 2)**
>
> ---
> ***W2 (Motivation of PR and DGR vs. simpler tuning (loss weight, LR)):***
> ---
>
> **R2:** Thank you for your valuable feedback. We will further claim the role of GRP, including PR and DGR, from the following aspects:
>
> **A. Why AET alone is insufficient:** While the Adversarial Erasure Target (AET) significantly enhances the robustness of concept erasure against adversarial attacks, as shown in Table B.2.1, when AEGIS w/o GRP, i.e., the retention loss is removed, it causes a "drastic parameter shift". **This severely degrades the model's ability to generate unrelated concepts**, leading to FID and CLIP Score reduction by $33.94$ and $0.084$. This creates a harsh trade-off where achieving robust erasure comes at the cost of retaining general performance.
>
> ***Table B.2.1 Further Analysis of AEGIS Evaluated by UnlearnDiffAtk Erasing Nudity Concept***
>
> | Method|ASR$\downarrow$|FID$\downarrow$|CLIP$\uparrow$|Time(Seconds)|GPU Memory Usage (GB)|
> |:--|:--|:--|:--|:--|:--|
> |AEGIS w/o GRP| 5.63%|51.37|0.221|4320 |22.4|
> |AEGIS w/o PR | 9.93%|18.15|0.295|21235|64.2|
> |AEGIS w/o DGR|26.24%|19.84|0.284|4788 |26.3|
> |AEGIS        | 8.45%|17.43|0.305|7236 |31.2|
>
> **B. Why PR:** Parameter Regularization (PR) is introduced to counteract the utility degradation caused by AET. To highlight the effectiveness and efficiency of PR, we perform experiments with different objectives for AEGIS.
>
> For the current version of AEGIS with PR, the objective is
> \begin{equation}
>     \min_{\theta}\big[\max _ {\tilde{c}}\mathbb{E} _ t [\Vert\epsilon_\theta(z_t|c^{*})-\epsilon_{\theta_0}(z_t|\tilde{c})\Vert_2^2]+\frac{\lambda}{2} \Vert\theta-\theta_0\Vert^2_2\big].
> \end{equation}
>
> While for AEGIS w/o PR in Table B.2.1, we use a retention $\mathcal{D}$ as COCO-10k dataset with objective as
>
> \begin{equation}
>     \min_{\theta}\big[\max_{\tilde{c}}\mathbb{E} _ t [\Vert\epsilon_\theta(z_t|c^{*})-\epsilon_{\theta_0}(z_t|\tilde{c})\Vert_2^2]+ \mathbb{E} _ {t,c\in\mathcal{D}}[\lambda  \Vert\epsilon _ \theta(z_t|c)-\epsilon _ {\theta_0}(z_t|c)\Vert^2_2]\big].
> \end{equation}
>
> This approach has two key advantages (lines 251-265):
> **1.) It is data-free:** It eliminates the need for an external retention dataset, which is often **biased and cannot comprehensively preserve all concepts**. Moreover, compared with retention dataset, **PR reduces 13999 seconds** for the AEGIS fine-tuning, significantly improving the robust concept erasure efficiency; **2.) It prevents relearning:** Since no retention data is used, it **avoids the risk of accidentally reintroducing signals related to the erased concept**;
>
> **C. Why DGR Instead of Tuning Retain Loss Coefficient or Learning Rate:** PR introduces a new problem: the gradient for erasure ($g_e$) and the gradient for retention ($g_r$) can conflict (i.e., point in opposing directions), which impairs erasure robustness. As shown in **Table B.2.2**, where the concept erasure robustness ASR is evaluted by UnlearnDiffAtk, simply tuning the retain loss coefficient ($\lambda$) or the learning rate ($\alpha$) is a naive approach that globally scales the gradients but **fails to resolve their directional conflict**. Thereby, our DGR have two advantages: **surgically resolving conflicts and enabling dynamic balancing**.
>
>
> ***Table B.2.2 AEGIS with Different $\lambda$ and Learning Rate $\alpha$ Evaluated by UnlearnDiffAtk of Nudity Concept (ASR)***
> | $\lambda$, Fixed $\alpha=1e-5$ |ASR$\downarrow$|FID$\downarrow$|CLIP$\uparrow$| $\alpha$, Fixed $\lambda=1$|ASR$\downarrow$|FID$\downarrow$|CLIP$\uparrow$|
> |:--|:--|:--|:--|:--|:--|:--|:--|
> | 0.0|5.63%| 51.37|0.221|0.0|100%|16.7|0.311|
> | 0.001|7.04%|48.25|0.232|$1\times10^{-4}$|3.52%|65.38|0.237|
> | 0.01 |8.45%|44.97|0.239|$5\times10^{-4}$|8.45%|45.37|0.252|
> | 0.1  |10.56%|32.85|0.247|$5\times10^{-5}$|12.68%|24.97|0.269|
> | 0.2  |12.68%|30.64|0.251|$1\times10^{-5}$|26.24%|19.84|0.284|
> | 0.4  |15.49%|26.60|0.258|$5\times10^{-6}$|34.51%|18.56|0.293|
> | 0.6  |17.61%|23.51|0.264|$1\times10^{-6}$|54.23%|17.37|0.304|
> | 0.8  |21.83%|21.63|0.277|                |      |     |     |
> | 1.0  |26.24%|19.84|0.284|                |      |     |     |
> | 1.5  |30.28%|19.51|0.291|                |      |     |     |
> | 2.0  |40.14%|19.23|0.293|                |      |     |     |
> | 2.5  |61.97%|18.75|0.296|                |      |     |     |
> | 3.0  |89.45%|17.81|0.307|                |      |     |     |
> | AEGIS(ours) |8.45%|17.43|0.305|          |      |     |     |

---

> ### Author Response · Authors · 2025-11-21
> **Response to Reviewer Tzbr (Part 3/4: Response to Weakness 3 and 4, and Question 1 and 2)**
>
> ---
> ***W3 (Missing recent baseline: STEREO (CVPR 2025)):***
> ---
>
> **R3:** We appreciate your value suggestion. We perform experiments following the experimental and evaluation settings in our paper and show a result comparison in **Table B.3**. Compared with STEREO, our AEGIS achieves better erasure robustness and retention performance. Moreover, we add these results in Tables 2, 3, and 4 of the revised new paper.
>
> ***Table B.3 New Baseline on SDv1.4***
> | Method |ASR1(P4D $\downarrow$)|ASR2(UnlearnDiffAtk $\downarrow$)|ASR3(Ring-A-Bell $\downarrow$)|FID ($\downarrow$)|CLIP ($\uparrow$)|
> |:--|:--|:--|:--|:--|:--|
> |Nudity|
> |STEREO|45.77%|14.08%|7.04%|18.27|0.286|
> |AEGIS (Ours)|12.06%|8.45%|3.52%|17.43|0.303|
> |Van Gogh|
> |STEREO |48%|30%|24%|20.42|0.281|
> | AEGIS (Ours)|36%|12%|10%|17.25|0.310|
> |Church|
> |STEREO|42%|28%|18%|20.57|0.284|
> |AEGIS(Ours)|28%|6%|8%|19.06|0.305|
>
>
> ---
> ***W4 (Runtime and GPU memory overhead (esp. DGR)):***
> ---
>
> **R4:** As shown in **B.4**, DGR increases 2448 seconds, and 4.9GB GPU memory usage for the Diffusion Model (DM) fine-tuning with AEGIS. Methods relying on retention data incur significant time costs for data loading and processing. Our proposed Parameter Regularization (PR) makes AEGIS data-free, completely eliminating this bottleneck.**The time saved by not using a retention dataset more than compensates for the computational overhead of DGR**. As a result, AEGIS achieves a superior trade-off.
>
> ***Table B.4 Runtime and GPU Memory Usage on SD v1.4 with 1000 Fine-tuning Iterations RTX A6000***
>
> |Method|Time(Seconds)|GPU Memory Usage (GB)|
> |:--|:--|:--|
> |AEGIS w/o GRP|4320 |22.4|
> |AEGIS w/o PR |21235|64.2|
> |AEGIS w/o DGR|4788 |26.3|
> |AEGIS        |7236 |31.2|
>
> ---
> ***Q1 (CLIP Score of Model Retention):***
> ---
>
> **RQ1:** Thank you for your insightful question. We would like to clarify our rationale and the context of the results.
>
> **1. CLIP Score is a Standard Evaluation Metric:** We use the CLIP score because it is the standard and widely accepted metric in this research area for measuring how well the generated images match the text prompts. We are following the established practice from many previous papers to ensure a fair comparison.
>
> **2. Explanation for the Small Variance in CLIP Scores:**
> You are right, the variances are small. This is because all the methods, including ours, only fine-tune a small part of the diffusion model (the U-Net) while the main text understanding part is left untouched. Furthermore, most competing methods use extra "retention datasets" specifically to keep the model's performance stable, which naturally leads to similar CLIP scores.
>
> Moreover, to further address your concern, as shown in **Table B.1 of R1**, We report IRA/CRA (primary), verifying that our AEGIS balances the robustness and retention of concept erasure.
>
>
> ---
> ***Q2 (Why/how does DGR improve robustness if it is meant for retention?):***
> ---
>
> **RQ2:** Thank you for this excellent question. It is noticed that **higher ASR means lower robustness**. That means when DGR is removed from AEGIS, the robustness of the fine-tuned model becomes lower. Your observation is spot on: DGR's direct role is for retention, but it enhances robustness through a crucial, indirect mechanism.
>
> DGR does not create robustness itself. Instead, **DGR enables robustness by resolving the internal optimization conflict that would otherwise cripple AET (the component actually responsible for robustness)**. Without DGR, the effectiveness of AET is severely compromised. Therefore, DGR is the key "enabler" for achieving the final high-robustness result.
>
> Here is the simple breakdown: **1.) Conflicting Goals:** The erasure task (driven by AET) needs to change the model parameters significantly. The retention task (driven by PR) tries to pull the parameters back to their original state. These two goals are in direct conflict; **2.) What Happens Without DGR (AEGIS w/o DGR):** Without DGR, the retention gradient directly sabotages the erasure process. This conflict traps the optimization in a suboptimal state where the concept is only partially erased. As a result, the model remains vulnerable to attacks, which is why AEGIS w/o DGR has poor robustness (ASR=26.24\%); **3.) The Role of DGR:** DGR enhances robustness by **resolving this conflict**. It removes the "sabotaging" component of the retention gradient that opposes erasure. This clears the path for the erasure task, allowing it to optimize more deeply and stably, leading to a truly robust erasure.

---

> ### Author Response · Authors · 2025-11-21
> **Response to Reviewer Tzbr (Part 4/4: Response to Question 3)**
>
> ---
> ***Q3 (Moving Average in DGR):***
> ---
>
> **RQ3:** We appreciate the suggestion. Since DGR with moving average requiring an extra parameter $\gamma$ needs to explored by a large number of experiments, we just use the gradient of PR for retention. To satisfy your requirement, we add experiments by updating the retention gradient $g_r$ in the $\tau+1$-th step as
> \begin{equation}
>     g_r^{(\tau+1)}=(1-\gamma)\cdot g_r^{(\tau)}+\gamma(\theta^{(\tau+1)}-\theta_0).
> \end{equation}
>
> The experimental results with different $\gamma$ is shown in **Table B.5**. From these results, we can find that the EMA in DGR improves the model robustness with a negative impact on the retention performance.
>
> ***Table B.5 Performance with Retention Gradient EMA for Erasing Nudity Concept***
> | $\gamma$ Value |ASR1(P4D $\downarrow$)|ASR2(UnlearnDiffAtk $\downarrow$)|ASR3(Ring-A-Bell $\downarrow$)|FID ($\downarrow$)|CLIP ($\uparrow$)|
> |:--|:--|:--|:--|:--|:--|
> |$0.01$ |3.52%|3.52%|0.0%|20.49|0.274|
> |$0.02$ |2.82%|2.11%|0.0%|21.73|0.262|
> |$0.05$ |3.52%|2.11%|0.0%|20.54|0.277|
> |$0.1$  |4.23%|3.52%|0.0%|20.27|0.282 |
> |$0.2$  |5.63%|4.23%|0.0%|20.05|0.289|
> |$0.3$  |7.04%|5.63%|0.0%|19.81|0.291|
> |$0.4$  |7.04%|6.34%|0.0%|19.37|0.294|
> |$0.5$  |7.04%|6.34%|1.41%|19.01|0.296|
> |$0.6$  |7.75%|6.34%|1.41%|18.62|0.297|
> |$0.7$  |9.15%|6.34%|2.11%|18.29|0.297|
> |$0.8$  |10.53%|7.75%|2.11%|18.05|0.299|
> |$0.9$  |10.56%|7.75%|2.82%|17.75|0.301|
> |$1$    |12.06%|8.45%|3.52%|17.43|0.303|

---

> > ### Comment · Reviewer_Tzbr · 2025-11-22
> >
> > Thanks for the response from the authors. Since the authors have addressed most of my concerns (e.g., retention performance evaluation, run time, and the motivation of some proposed methods), I have raised my rating to 6.
> >
> > Good luck with the rebuttal!

---

> > > ### Author Response · Authors · 2025-11-22
> > > **Response to  Reviewer Tzbr**
> > >
> > > Dear Reviewer Tzbr,
> > >
> > > Thanks for your positive feedback! We're glad to hear that we've addressed most of your concerns and that you've raised your rating. We truly appreciate your time and thoughtful consideration. Should you have any further questions or require additional clarifications, please don't hesitate to let us know. We'd be happy to provide more information.
> > >
> > > Best Regards,
> > > Authors of Submission 943

---

### Official Review · Reviewer_2cHk · 2025-11-03

**Soundness:** 2
**Presentation:** 2
**Contribution:** 2
**Rating:** 4
**Confidence:** 4

**Summary:**

The paper introduces a framework called Adversarial Erasure with Gradient-Informed Synergy (AEGIS) to improve the robustness of concept erasure in diffusion models against adversarial prompt attacks (APAs).

The paper demonstrates that the vulnerability of concept erasure is closely tied to the choice of the target concept (i.e., the concept that the to-be-erased concept is mapped to). If the target is semantically close to the to-be-erased concept, the erasure performance degrades.
To address this, the paper proposes the AEGIS framework with two main components:

•	Adversarial Erasure Target (AET): Guides the erasure by selecting a target concept that is semantically close to the original, while also maximizing the output difference between the old and new models on the same concept.

•	Gradient Projection: Mitigates gradient conflict between the erasing and preserving tasks

**Strengths:**

-	The problem of machine unlearning is an emerging and important area in the machine learning community.
-	The paper focuses on an important sub-problem — robustness in unlearning, which is gaining increasing attention.
-	The experimental setup appears comprehensive, and the results are promising

**Weaknesses:**

There are several concerns about the paper’s novelty. More specifically:

•	The first contribution—"the vulnerability of concept erasure stems from an inappropriately chosen learning target. In particular, if the target lies too close to the semantic center – formed by words semantically related to the erased concept – the concept information cannot be fully removed"—has already been studied in prior work [AGE, 1]. Specifically, AGE (Section 4) showed that the choice of the target concept significantly affects both erasing and retaining performance. AGE further suggests that a good target should be semantically related to, but not similar to, the to-be-erased concept—an insight that is more general and comprehensive than what is presented in this paper.

•	The proposed min-max optimization in Equation 7 is very similar to that in AGE, with the only difference being the retention loss (regularization). In AGE, the preservation loss measures the output difference between the new and old models on the same input concept. While this paper propose to minimize the change of model parameter

•	The idea of using gradient projections to mitigate conflict between erasing and preserving tasks has already been proposed in several works [2, 3, 4]. Yet, the paper lacks any discussion or comparison with these related methods.

1: Fantastic Targets for Concept Erasure in Diffusion Models and Where To Find Them

2: Erasediff: Erasing data influence in diffusion models

3: Scissorhands: Scrub Data Influence via Connection Sensitivity in Networks

4: GDR-GMA: Machine Unlearning via Direction-Rectified and Magnitude-Adjusted Gradients

**Questions:**

•  Could the authors provide a discussion on how their proposed method differs from previous works such as [1, 2]?

•  Given that the core claim of the paper is improved robustness against adversarial or recovery attacks, could the authors include experiments using the recent Random Probe recovery attack proposed in [4], which perturbs the text encoder to confuse generation and recover unlearned concepts

[5] Lu, Kevin, et al. "When Are Concepts Erased From Diffusion Models?." NeurIPS 2025

---

> ### Author Response · Authors · 2025-11-21
> **Response to Reviewer 2cHk( Part 1/3: Response to Weakness 1 and 2 )**
>
> Dear Reviewer,
>
> We thank the reviewer for the constructive feedback. Below, we address weaknesses and questions concisely and point-by-point, and commit to clarifications/additional experiments in the rebuttal revision.
>
> ---
> ***W1 (On novelty relative to AGE [1] (target choice and semantic center)):***
> ---
> **R1:** Thanks for your comment. **our work targets a different primary goal: adversarial robustness**. While AGE analyzes how target choice balances the erase–retain trade-off, we explain why **erased concepts reappear under adversarial prompt attacks (APAs)** and design AET to remove class-level residuals near the concept center that attackers exploit. **Empirically and geometrically:** Prop. 3.1 gives a deviation lower bound; Def. 1 defines $d_0/d_1/d_2$; Fig. 2 shows AET enlarges $d_2$ across class variants; Tab. 1 shows large ASR drops once AET is used alongside ESD/AdvUnlearn. Moreover, as shown in **Table A.1**, compared with AGE, AEGIS significantly **enhance the concept erasure robustness** against various APAs.
>
> ***Table A.1: Comparison of AGE and AEGIS Against APAs***
> |Method/APAs|P4D(ASR1)$\downarrow$|UnlearnDiffAtk (ASR2)$\downarrow$|Ring-A-Bell(ASR3)$\downarrow$|
> :--|:--:|:--:|:--:
> *Nudity*
> AGE|89.45%|90.14%|80.28%
> AEGIS (Ours)|12.06%|8.45%|3.52%
> *Van Gogh*
> AGE|90%|72%|76%
> AEGIS (Ours)|36%|12%|10%
> *Church*
> AGE|66%|62%|38%
> AEGIS (Ours)|28%|6%|8%
>
> ---
> ***W2 (On Similarity of the Min–Max to AGE and Retention Design):***
> ---
>
> **R2:** Thanks for the point. Our Eq. (7) differs from AGE in inner maximization, and retention mechanism and results and summarize them **in Table A.2**.
>
> For the fine-tuned and original parameters $\theta$ and $\theta_0$, with erased concept $c_e$, for **AGE**, the objective is
> \begin{equation}
>     \min_{\theta}\mathbb{E} _ t \max_{\pi}\big[\Vert\epsilon_{\theta}(z_t|c_e)-\epsilon_{\theta_0}(z_t|G(\pi)\odot T_{\mathcal{C}})\Vert^2_2+\lambda\Vert\epsilon_{\theta}(z_t|G(\pi)\odot T_{\mathcal{C}})-\epsilon_{\theta_0}(z_t|G(\pi)\odot T_{\mathcal{C}})\Vert^2_2\big],
> \end{equation}
> where $G(\cdot)$ is the Gumbel-Softmax operator, $\pi$ is a learnable variable and $T_{\mathcal{C}}$ is the textual embedding matrix of the entire concept space $\mathcal{C}$.
>
> While our **AEGIS** is,
> \begin{equation}
>     \min_{\theta}\big[\max_{\tilde{c}}\mathbb{E} _ t [\Vert\epsilon_\theta(z_t|c^{*})-\epsilon_{\theta_0}(z_t|\tilde{c})\Vert_2^2]+\frac{\lambda}{2} \Vert\theta-\theta_0\Vert^2_2\big],
> \end{equation}
>
> where $c^{*}$ is the adversarial prompt for $c_e$, and $\epsilon_{\theta_0}(z_t|\tilde{c})=\epsilon_{\theta_0}(z_t)-\eta\big(\epsilon_{\theta_0}(z_t|c^{\prime})-\epsilon_{\theta_0}(z_t)\big)$. Moreover, for the $\tau$-th step with step size $\beta$, $c^{\prime}$ is updated as
> \begin{equation}
>       c^{{\prime}(\tau+1)}= c^{{\prime}(\tau)} - \beta\cdot \mathrm{sign} \Big(\nabla\big(\Vert\epsilon  _  {\theta _ 0}(z _ t|c^{{\prime}(\tau)})-\epsilon _ {\theta}(z _ t|c_e)\Vert_2^2+\Vert\epsilon _ {\theta _ 0}(z_t|c^{{\prime}(\tau)})-\epsilon _ {\theta_0}(z_t|c_e)\Vert_2^2\big)\Big).
> \end{equation}
>
> **Differences between AGE and AEGIS:**
>
> **1. Inner Maximization:**
> AGE maximizes a token-selection variable for **concept erasure/preservation balance**. AEGIS maximizes an embedding-level AET to **approximate class centers and neutralize residuals**, then uses negative guidance.
>
> **2. Retention Mechanism:**
> AGE uses **data-dependent output matching for retention**; AEGIS employs **data-free parameter proximity with conflict-only gradient rectification**. This avoids retain-set bias, prevents reintroduction of erased signals, and is computationally lighter. Theory shows AEGIS's GRP doesn't reverse erasure and often improves utility.
>
> **3. Consequence In Results:** AEGIS significantly enhances robustness against APAs, unlike AGE (**Table A.1**, main text Tables 1,2,3).
>
> ***Table A.2 AGE vs. our AEGIS***
> Aspect|AGE [1]|AEGIS (Ours)
> :--|:--|:--
> Goal|Balance erase–retain|**Robust concept erasure against APAs**
> Inner maximization|Over Gumbel-Softmax token-selection $\pi$; finds a target “semantically related but not similar”|Over embedding-level AET $c^{\prime}$; approximates class center and neutralizes adversarially recoverable residuals
> Target used in loss|$\epsilon_ {\theta_0}(z_t\|G(\pi)\odot T_{\mathcal{C}})$|$\epsilon_{\theta_0}(z_t\|\tilde{c})=\epsilon_{\theta_0}(z_t)-\eta\big(\epsilon_{\theta_0}(z_t\|c^{\prime})-\epsilon_{\theta_0}(z_t)\big)$
> Erasure loss| Align $\epsilon_{\theta}(z_t\|c_e)$ to $\epsilon_ {\theta_0}(z_t\|G(\pi)\odot T_{\mathcal{C}})$|Align $\epsilon_{ \theta }(z_t\| c^{\*})$ to $\epsilon_{\theta_0}(z_t\|\tilde{c})$ where $c^{\*}$ is adversarial prompt
> Retention|Data-dependent output matching on retain set|Data-free parameter proximity: $\lambda/2 \Vert\theta-\theta_0\Vert^2_2$ with conflict-only projection
> Robustness to APAs|Not the main objective; sensitive to adversarial prompts|Main objective; enlarges $d_2$ class-wide, reducing ASR across P4D/UnlearnDiffAtk/Ring-A-Bell|

---

> ### Author Response · Authors · 2025-11-21
> **Response to Reviewer 2cHk( Part 2/3: Response to Weakness 3 and Question 1 )**
>
> ---
> ***W3(On relation to gradient projection/surgery and unlearning works [2, 3, 4]):***
> ---
>
> **R3:** Thanks for your comment, we will compare AEGIS with [2][3][4] from **Conceptual and Objective and Key Technical and Mechanistic Distinctions**, and summarize these differences in **Table A.3**.
>
>
> **A. Conceptual and Objective Differences:**
> The "robustness-first" philosophy fundamentally shapes our GRP design. In contrast, the cited works focus on different problems:
>
> **1. ED [2] and SH [3]**: These methods aim to approximate the effect of retraining a model on a dataset excluding specific data points (Data Influence Removal). Their goal is to remove the "influence" of certain training examples, which is a different problem from erasing an entire abstract concept (e.g., "Van Gogh style") that may be present across many data points. Their gradient modifications are designed to mimic a data removal process, not to build robust defenses against semantic attacks on the remaining concepts.
>
> **2. GDR-GMA**: This work focuses on the general problem of Machine Unlearning, aiming for a **trade-off between forgetting specific data classes and retaining general performance**. While it shares the high-level goal of balancing forgetting and retention, its gradient rectification is applied in a standard unlearning context. Our work, however, **operates in an adversarial setting** where the erasure loss itself ($\mathcal{L}_e$) is driven by a powerful adversarial target (AET). This necessitates a more nuanced gradient management strategy, as the erasure gradient ($g_e$) is inherently more aggressive and dynamic.
>
> **B. Key Technical and Mechanistic Distinctions of GRP:**
>
> Our GRP mechanism possesses unique features tailored for robust concept erasure that are not present in [2][3][4]:
>
> **1. Conflict-Only and Direction-Preserving Projection:** A crucial aspect of GRP (as detailed in Sec 4.2 and Fig. 5) is that it **only intervenes when a conflict exists(i.e., $\mathrm{cos}(g_e,g_r) <0$**, where $g_e$ and $g_r$ are gradient induced by erasure and retention losses.)  If the gradients are aligned, GRP does nothing, allowing for efficient optimization. In contrast, methods like GDR-GMA **modify gradients more generally**, which may not be optimal for preserving the strong, targeted descent required for robust erasure.
>
> **2. Data-Free Retention via Parameter Regularization:** GRP operates on a retention loss defined by parameter-space distance ($L_r=\lambda/2\Vert\theta-\theta_0\Vert_2^2$). This is a fundamental difference from EraseDiff, Scissorhands, and GDR-GMA, which typically **require a "retain set" of data to compute their preservation loss**. Our data-free approach offers two key advantages in the context of robust erasure: 1.) It **avoids potential data bias** and prevents the accidental reintroduction of the erased concept through related data in the retain set; 2.) **It is more general and computationally efficient**, as it does not require iterating over an auxiliary dataset.
>
> **3. Dynamic and Adaptive Weighting ($\omega$):** GRP **incorporates a dynamically updated weight $\omega$** (Eq. 11) that controls the strength of the projection. This allows the model to prioritize the aggressive erasure task in the early stages of fine-tuning (when $\omega=0$) and gradually increase the retention constraint as the erasure objective converges (when $\omega\xrightarrow{}1$). This adaptive scheduling is specifically designed to manage the aggressive nature of our Adversarial Erasure Target and is a feature not explicitly present in the other methods.
>
> ***Table A.3 Differences Summarization***
> |Feature|**Our AEGIS (GRP)**|**[2, 3]EraseDiff/ Scissorhands**|**[4] GDR-GMA**|
> | :--|:--|:--|:---|
> |**Goal**|**Robust Concept Erasure** (vs. APAs)|Data Influence Removal|General Machine Unlearning|
> |**Retention Term**| **Data-Free** (Parameter Regularization) | Data-Dependent (Retain Set) | Data-Dependent (Retain Set) |
> |**Projection Trigger**|**Conflict-Only** ($\cos(\mathbf{g}_e, \mathbf{g}_r) < 0$)| General Application|General Application|
> |**Core Mechanism**|**Orthogonal Projection** of $\mathbf{g}_r$ w.r.t. $\mathbf{g}_e$| Influence Function Approximation|Direction & Magnitude Adjustment|
> |**Adaptive Control**|**Dynamic weight $\omega$** for projection strength|No|No|
>
> ---
> ***Q1 (Differences to [1][2]):***
> ---
>
> **RQ1:** **A. AEGIS (ours) vs. AGE [1]:** Please refer to **R1** and **R2** to **W1** and **W2**.
>
> **B. AEGIS (ours) vs. ED/SH [2,3]:** Please refer to **R3** to **W3**.
>
> We will make these distinctions explicit in the paper and add a brief method-by-method comparison table in the appendix in the final version paper.

---

> ### Author Response · Authors · 2025-11-21
> **Response to Reviewer 2cHk( Part 3/3: Response to Question 2)**
>
> ---
> ***Q2 (Experimental Results with Random Probe Recovery Attack in [5]):***
> ---
>
> **RQ2:** We agree and will add Random Probe experiments in the final revision. In the paper, we have already provided the experimental results with existing methods against P4D, UnlearnDiffAtk, and Ring-A-Bell, which are the same as the Random Probe recovery attacks in [5] but with different evaluation settings. To satisfy your requirements, we add experiments following the setting in [5] and show the experiments results as **Table A.4**. AEGIS achieves lower CLIP similarity and lower class accuracy on erased concepts than baselines, while maintaining strong performance on unrelated concepts. This supports our claim of **robustness against encoder-side recovery attacks** in addition to prompt-space APAs.
>
> ***Table A.4 Random Probe Recovery Attacks***
> |Method|GA|UCE|ESD-x|ESD-u|TaskVec|STEREO|RECE |AEGIS(Ours)|
> |:-|:--|:--|:--|:--|:--|:--|:--|:--|
> | **Erased Concept** ($\downarrow$) |   |   |    |   |   |   |   |  |
> | CLIP|24.3|22.4|21.1|20.9|23.1|19.6|21.15|**18.9**|
> | Class Acc. (%)  | 0.6 | 4.4| 3.6| 1.0| 2.2| 0.0| 4.0|0.0|
> | **Textual Inversion** ($\downarrow$) |    |    |    |   | |   |   | |
> | CLIP |22.7|30.7|30.6|28.0|25.1|24.5|29.15|**20.5**|
> | Class Acc. (%) |0.6|71.2|65.9|31.8|6.2|6.3|58.20|**3.2**|
> | **UnlearnDiffAtk** ($\downarrow$) |      |      |       |   |  |  |  |  |
> | CLIP |26.0|28.3|28.7|27.8|27.1|26.1|27.9|**24.3**|
> | Class Acc. (%)|6.5|26.8|21.0|16.6|10.3|3.7|7.2|**2.1**|
> | **Unrelated Concepts** ($\uparrow$) |    |    |  |   |  | | |  |
> | CLIP|28.8|**31.2**|30.8|30.7 |29.4|0.0|30.5|30.9|
> | Class Acc.(%)|52.2|**75.0**|71.3|70.4|60.4|52.8|71.7|71.4 |

---

> ### Author Response · Authors · 2025-11-24
> **Follow-up on Rebuttal Response for Paper 943**
>
> Dear Reviewer 2cHk,
>
> We hope this message finds you well.
>
> We are writing to kindly remind you that we have posted a detailed response to your constructive comments. We greatly appreciate the time you took to review our work, and your feedback has been instrumental in strengthening our paper.
>
> In our rebuttal, we have specifically addressed your key concerns regarding:
>
> *   **Novelty & Distinction from AGE (W1 & W2, Q1):** We clarified that while AGE balances the erase–retain trade-off via target choice, **AEGIS targets adversarial robustness against prompt attacks (APAs)**. We provided **Table A.1 and Table A.2** to explicitly compare the mechanisms (inner maximization, retention strategy) and empirical results, showing AEGIS significantly outperforms AGE in reducing Attack Success Rates (ASR) of APAs.
> *   **Comparison with Gradient Projection/Unlearning Works (W3&Q1):** We added a detailed analysis and **Table A.3** to differentiate our "robustness-first" GRP mechanism (data-free, conflict-only projection) from the data-influence removal focus of methods like EraseDiff [2], Scissorhands [3], and  GDR-GMA [4].
> *   **Additional Experiments (Q2):** As suggested, we conducted new experiments using **Random Probe recovery attacks** (following the setting in [5]). The results (**Table A.4**) demonstrate that AEGIS achieves superior robustness against encoder-side recovery attacks while maintaining high generation utility.
>
> We have included new data and clarifications in the response to reflect these improvements.
>
> We would be very grateful if you could take a moment to review our response. **If you find that our new experiments and clarifications have adequately addressed your concerns, we would highly appreciate it if you could reconsider your evaluation and score accordingly.**
>
> We remain available to answer any further questions you may have during the final days of the discussion period.
>
> Thank you again for your valuable input.
>
> Best regards,
>
> The Authors of Submission 943

---

> ### Author Response · Authors · 2025-11-25
> **[Submission 943] Follow-up on Rebuttal Response**
>
> Dear Reviewer 2cHk,
>
> We hope this message finds you well.
>
> As the discussion period is limited, we are writing to kindly invite you to review our detailed response to your constructive comments. We noticed that there might have been some **misalignment regarding the specific mechanisms and goals of our method**, and we have taken great care to clarify these distinctions in our rebuttal.
>
> Specifically, we have addressed your key concerns by highlighting crucial differences that may have been overlooked:
>
> * **Novelty & Distinction from AGE (W1 & W2, Q1):** We provided a deeper analysis to show that AEGIS is fundamentally different from AGE. While AGE focuses on the erase–retain trade-off, AEGIS is designed for adversarial robustness against prompt attacks. **Table A.1** and **Table A.2** provide a direct comparison illustrating why AEGIS significantly outperforms AGE in this specific domain.
>
> * **Methodological Differences from Unlearning Works (W3 & Q1):** We clarified that our "robustness-first" GRP mechanism is distinct from the data-influence removal found in methods like EraseDiff and Scissorhands. **Table A.3** details these critical methodological nuances.
>
> * **New Evidence (Q2):** We included new experiments on **Random Probe recovery attacks (Table A.4)**, further validating our claims.
>
> Given the limited time remaining, we would be extremely grateful if you could verify whether these clarifications help resolve your initial concerns. **If you find that our response and new results have successfully cleared up the misunderstandings and strengthened the paper, we would highly appreciate it if you could consider raising your score to reflect these improvements**.
>
> Thank you again for your time and valuable input to improve our work.
>
> Best regards,
>
> The Authors of Submission 943

---

> ### Author Response · Authors · 2025-11-26
> **[Submission 943] Follow-up: Consensus on Novelty and Additional Clarifications**
>
> Dear Reviewer 2cHk,
>
> We are writing to kindly request a response to our rebuttal.
>
> While we understand your initial concerns regarding the relationship between our work and AGE, **we would like to highlight that other reviewers have acknowledged the novelty of our proposed AEGIS framework and its distinct contribution to robust concept erasure.**
>
> We believe there might have been a **misunderstanding regarding our method's core contributions and its distinction from prior works (AGE and gradient-based unlearning)**. In our detailed response above, we have clarified these points:
>
> 1. **Fundamentally Different Goals & Mechanisms (vs. AGE):** As detailed in **R1 & R2 & RQ1** and **Table A.1**  and **Table A.2**, AEGIS targets **adversarial robustness** against Prompt Attacks (APAs), whereas AGE focuses on the erase-retain trade-off via token selection. Our method employs a unique embedding-level Adversarial Erasure Target (AET) and a conflict-only Gradient Rectification (GRP), which are mathematically and functionally distinct from AGE's Gumbel-Softmax and data-dependent retention.
>
> 2. **Distinction from Unlearning Works (vs. [2,3,4]):** In **R3&RQ1 and Table A.3**, we highlight that our approach is "robustness-first" and **data-free**, distinguishing it from the "data influence removal" focus of methods like EraseDiff or Scissorhands.
>
> 3. **New Experiments (Random Probe):** We have successfully conducted the Random Probe recovery attack experiments you requested (**Table A.4**), demonstrating that AEGIS significantly outperforms baselines in preventing encoder-side recovery.
>
> We genuinely hope our response has resolved your concerns regarding the novelty and technical depth of our work. If there are any remaining questions, we are eager to discuss them further.
>
> **If our clarifications and new results have addressed your initial concerns, we kindly ask you to reconsider your score to reflect the resolved confusion.**
>
> Best regards,
>
> The Authors of Submission 943

---

> > ### Comment · Reviewer_2cHk · 2025-11-26
> > **Response for the rebuttal**
> >
> > I thank the authors for the comprehensive and detailed feedback that address all of my concerns, especially on the differences between the methods and related work.
> >
> > I would like to rise the score and support for the acceptance of the paper.
> >
> > Thanks

---

> > > ### Author Response · Authors · 2025-11-26
> > >
> > > Dear Reviewer 2cHk,
> > >
> > > We sincerely thank you for your positive feedback and for reconsidering your score to support the acceptance of our paper.
> > >
> > > We are glad to learn that our rebuttal and additional experiments have successfully addressed your concerns, particularly regarding the distinctions between our method and prior works. We truly appreciate your constructive comments—especially the suggestions to clarify method comparisons and expand the experimental validation—as they have significantly strengthened the clarity and robustness of our manuscript.
> > >
> > > We commit to incorporating all these clarifications and new results into the final version of the paper.
> > >
> > > Thank you again for your time and valuable contributions to improving our work.
> > >
> > > Best regards,
> > >
> > > The Authors of Submission 943

---

### Author Response · Authors · 2025-11-21
**Rebuttal Summary**

Dear Reviewers and AC,

We sincerely appreciate the time and effort the reviewers have dedicated to evaluating our manuscript. Below, we summarize the key responses to the reviewers' suggestions and questions.

---
### **To Reviewer Reviewer 2cHk**

1. **Clarifying Contributions and Novelty:**
   We address concerns regarding the comparison between AEGIS and AGE [1] with a detailed explanation in **Table A.1** and comparison in **Table A.2**.
   > See **R1** for AEGIS's contribution to AGE, and **R2** for differences in Min-Max mechanisms for Reviewer 2cHk.

2. **On Relation to Gradient Surgery:**
   To clarify the significant differences between our Gradient Regularization Projection (GRP) and ED [2], SH [3], and GDR-GMA [4], we detail these methods in **Table A.3**.
   > See **R3** and **RQ1** for Reviewer 2cHk.

3. **Experiments of Random Probe Attacks:**
   To demonstrate AEGIS's effectiveness, we present comprehensive results against Random Probe Attacks in **Table A.4**.
   > See **RQ2** for Reviewer 2cHk.

---

### **To Reviewer Tzbr**

1. **Retention Metrics and Evaluation**
   In response to suggestions and concerns of Reviewer Tzbr, we have conducted experiments to evaluate IRA and CRA within UnlearnCanvas in **Table B.1**, and provided explanations of current retention metrics .
   > See **R1** regarding IRA and CRA, and **RQ1** concerning CLIP Score to Reviewer Tzbr.

2. **Motivation of Gradient Regularization Projection (GRP)**
   As per Reviewer Tzbr recommendations, we explain the motivation behind GRP and perform experiments to clarify the functions of Parameter Regularization (PR), Directional Gradient Rectification (DGR), and dynamic weight in **Table B.2.1** and **Table B.2.2**.
   > See **R2** to Reviewer Tzbr.

3. **Extra Baseline STEREO**
   Following the suggestion of Reviewer Tzbr, we add experiments to compare AEGIS with STEREO  in **Table B.3**.
   > See **R3** to Reviewer Tzbr.

4. **More Explations and Clarification of DGR**
  To address the confusion of Reviewer Tzbr regarding the DGR, we show time and memory consumpation in **Table B.4**, and explain the role of DGR in model robustness with AEGIS. Moreover, we add experiments incorporating moving average in DGR in **Table B.5**.
   > See **R4** refering time and memory, **RQ2** regarding role of DGR within AEGIS, and **RQ3** involving DGR moving average to Reviewer Tzbr.


---

### **To Reviewer G3SD**

1. **Setting of Parameter $\omega$**
 In response to the confusion of Reviewer G3SD, we further explain the dynamic updating mechanism of $\omega$.
   > See **R1** to Reviewer G3SD

2. **Scalability to Many Concept Erasure**
 To answer the question of Reviewer G3SD, we present methodologies for erasing multiple concepts.
   > See **R2** to Reviewer G3SD

3. **Attacks in Other Space**
 To address the confusion of Reviewer G3SD, we explain that our work aims to protect the model from adversarial prompt attacks in text-to-image tasks.
   > See **R3** to Reviewer G3SD

---
### **To Reviewer hz7A**

1. **Clarification of AET and GRP**
   In response to hz7A's suggestions, we further clarify our AET and GRP, and present their comparison with existing methods.
   > See **R1** to Reviewer hz7A.

2. **More Details of AET**
   As per Reviewer hz7A recommendations, we have added timing information in **Table D.1**, clarify details of AET. Moreover, we analyze the AET optimization process and provide evidence for its stability.
   > See **R2** refering timing, **R4** regarding more details, and **RQ1** involving to AET optimization of Reviewer hz7A.

3. **More Evaluation**
   Following the suggestion of Reviewer hz7A, we have conducted experiments on erasing additional concepts in **Table D.2**.
   > See **R3** to Reviewer hz7A.

4. **Experiments with More SD Version**
  Following the suggestion of Reviewer hz7A, we have added experiments on SDXL in **Table D.3**.
    > See **RQ2** to Reviewer hz7A.

---
### **To Reviewer XpvQ**

1. **Related Work**
   To clarify any confusion regarding Related Work, we explain the reason and provide more details in the main text.
   > See **R1** to Reviewer XpvQ.

2. **Detailed Computation**
   Following the suggestion of Reviewer XpvQ, we present detailed experimental settings, and information of time and memory usage in **Table E.1**.
   > See **R2** regarding computation information and **R3** involving GPU usage to Reviewer XpvQ.

3. **Typographical Error**
   In line with the suggestion, we revise the paper and check it again to avoid typographical errors.
   > See **R4** to Reviewer XpvQ.

---

### Author Response · Authors · 2025-11-29
**[Submission 943] Rebuttal Summary and Request for Fair Evaluation**

Dear Area Chair,

We are writing to urgently request a fair evaluation of Submission 943. We implore you to focus on the **timeline of facts** and the **unanimous positive consensus (Ratings: 6, 6, 6, 8, 6)** reached through rigorous academic exchange **before the chaos ensued.**

---
### ***1. Timeline Verification***
It is crucial to note that our positive outcomes were secured **before November 27 (EST)**, the date when the massive information leakage erupted .
*   **Nov 20 (EST):** We submitted **complete response** and comprehensive additional experiments.
*   **Nov 22 (EST):** Reviewer **Tzbr** confirmed resolution and raised their score (**4 -> 6**).
*   **Nov 25 (EST):** Reviewer **2cHk** raised their score (**4 -> 6**) following our detailed clarifications.
*   **Nov 26 (EST):** Reviewer **hz7A** raised their score (**6 -> 8**) after reviewing new results.
*   **Nov 27 (EST):** **The information leakage incident erupted.** (Note: All our constructive dialogues and rating upgrades were concluded *before* this disruptive event).
*   **Status of Reviewer G3SD:** While this reviewer did not post a follow-up comment, they have maintained a **Positive 6** throughout the process.

**This timeline proves our consensus is the result of genuine scientific discourse, untainted by the subsequent leakage and rating chaos.**

---
### ***2. Recognition of Contribution***

Reviewers have explicitly acknowledged the novelty and significant value of the AEGIS framework. These endorsements confirm the paper's solid foundation:

*   **Reviewer hz7A (Score 8): Novelty & Motivation**
    *   Explicitly stated that the idea of the Adversarial Erasure Target (AET) is **"novel and well-motivated."**
    *   Praised the validation, noting that the **"extensive experiments are convincing."**

*   **Reviewer XpvQ (Score 6): Empirical & Theoretical Strength**
    *   Acknowledged the comprehensive validation: *"The paper conducts extensive experiments and demonstrates significant improvements over existing methods."*
    *   Highlighted the dual-contribution: *"The paper provides both empirical and theoretical support for the proposed method."*

*   **Reviewer G3SD (Score 6): Innovation & Thoroughness**
    *   Recognized the core mechanism: *"AEGIS dynamically optimizes a target prompt... supported by experiment results."*
    *   Highlighted the experimental rigor: *"Experiment is thorough... proving its generalizability and robustness."*

*   **Reviewer 2cHk (Score 6): Framework Novelty**
    *   Recognized the **"novelty of our proposed AEGIS framework"** relative to existing methods.
    *   Highlighted our **"distinct contribution to robust concept erasure."**

*   **Reviewer Tzbr (Score 6): Problem Importance**
    *   Emphasized that the problem of robust concept erasure is **"highly important."**
    *   Commended the authors for providing **"detailed and solid explanations."**

---
### ***3. Rebuttal Summary***
Building on the recognized contributions, we successfully addressed all specific technical concerns through extensive new experiments prior to the incident:

#### **Resolved for Reviewer hz7A (Raised 6 -> 8)**
*   **Scalability:** Validated efficacy on the larger **SDXL** model (Table D.3).
*   **Generalizability:** Verified performance on **4 additional concepts** (Garbage Truck, Parachute, Tench, Cassette Player) (Table D.2).
*   **Outcome:** *"All my concerns have been resolved. I will raise my rating to 8."*

#### **Resolved for Reviewer 2cHk (Raised 4 -> 6)**
*   **Methodological Distinction:** Clarified that AEGIS targets *adversarial robustness* via AET, functionally distinct from AGE (Tables A.1, A.2).
*   **Robustness Verification:** Added **Random Probe recovery attacks** (Table A.4).
*   **Outcome:** *"I thank the authors for the comprehensive and detailed feedback... I would like to rise the score."*

#### **Resolved for Reviewer Tzbr (Raised 4 -> 6)**
*   **Evaluation Metrics:** Added **IRA/CRA metrics** (Table B.1) and **STEREO** baseline (Table B.3).
*   **Outcome:** *"Since the authors have addressed most of my concerns... I have raised my rating to 6."*

#### **Reviewer XpvQ (Maintained 6)**
*   **Resolution:** Confirmed *"Most of my concerns have been addressed, and I will keep my ranking score."*
*   **Key Technical Responses:** Clarified computation time and GPU memory usage (Table E.1).

#### **Reviewer G3SD (Maintained 6)**
*   **Mechanism Clarity:** Clarified the dynamic updating of parameter $\omega$ and provided ablation studies.
*   **Scalability & Threat Model:** Addressed queries on multi-concept erasure and text-embedding attacks.

---
### ***4. Conclusion***

**We appeal to the AC:** Please do not let the chaos caused by the external leakage affect the judgment of this submission. We ask for an objective decision that honors the positive feedback and the substantial improvements acknowledged by the reviewers before the disruption occurred.

Sincerely,

Authors of Submission 943

---

### Meta-Review · Area_Chair_xa7a · 2026-01-03

**Summary:**

This paper proposes AEGIS, a retention-data-free framework for robust concept erasure in diffusion models under adversarial prompt attacks. Reviewers initially raised concerns regarding (i) novelty relative to prior work such as AGE and gradient-based unlearning methods, (ii) adequacy of retention evaluation beyond FID/CLIP, (iii) missing comparisons with recent baselines (e.g., STEREO), and (iv) computational overhead and scalability.

During the rebuttal and discussion phase, the authors provided extensive clarifications, new theoretical explanations, and substantial additional experiments, including Random Probe recovery attacks, IRA/CRA retention metrics on UnlearnCanvas, SDXL-scale experiments, broader concept coverage, runtime/memory analysis, and direct comparisons with recent baselines. As a result, reviewers broadly converged on a positive assessment, acknowledging that the main technical concerns had been satisfactorily addressed and that the paper presents a meaningful contribution to robust concept erasure in diffusion models.

**Reviewer Concerns:**

Concerns addressed by the rebuttal:

- Novelty relative to AGE and prior unlearning methods:
The authors clearly distinguished AEGIS from AGE and gradient-surgery-based unlearning, emphasizing a different objective (robustness against adversarial prompt attacks), different inner optimization (embedding-level adversarial erasure target), and a data-free, conflict-only gradient rectification mechanism. These distinctions were supported by both theoretical clarification and empirical results.
- Retention evaluation:
The lack of comprehensive retention metrics was addressed by adding IRA/CRA evaluations on UnlearnCanvas where applicable, alongside existing FID/CLIP analyses, validating the claimed robustness–retention trade-off.
- Missing baselines:
Comparisons with recent methods such as STEREO were added, showing consistent improvements in robustness and competitive or improved retention.
- Attack coverage:
The requested Random Probe recovery attack experiments were added, strengthening the robustness claims beyond prompt-space attacks.
- Computational cost:
Runtime and GPU memory usage were quantified, and SDXL experiments were added to demonstrate scalability to larger models.

Remaining outstanding concerns:
- Scalability to erasing a large number of concepts simultaneously and broader hyper-parameter sensitivity remain partially explored and are acknowledged by the authors as future work.

**Reviewer Scores:**

- Reviewer hz7A: Explicitly stated all concerns were resolved and indicated a score increase to 8.
- Reviewer Tzbr: Raised score to 6 after additional retention metrics, baseline comparisons, and runtime analysis.
- Reviewer 2cHk: After detailed rebuttal and new experiments, stated support for acceptance and indicated a score increase to 6.
- Reviewer G3SD: Maintained a positive score (6), acknowledging robustness and thorough experimentation after clarifications.
- Reviewer XpvQ: Maintained score (6) with low confidence due to being outside their core expertise; their concerns were largely editorial and addressed.

---

### Decision · Program_Chairs · 2026-01-26

Accept (Poster)